# COLEP: Certifiably Robust Learning-Reasoning Conformal Prediction via Probabilistic Circuits

**Mintong Kang**
UIUC
mintong2@illinois.edu

**Nezihe Merve Gürel**
TU Delft
n.m.gurel@tudelft.nl

**Linyi Li**
UIUC
linyi2@illinois.edu

**Bo Li**
UChicago & UIUC
bol@uchicago.edu

## Abstract

Conformal prediction has shown spurring performance in constructing statistically rigorous prediction sets for arbitrary black-box machine learning models, assuming the data is exchangeable. However, even small adversarial perturbations during the inference can violate the exchangeability assumption, challenge the coverage guarantees, and result in a subsequent decline in prediction coverage. In this work, we propose the first certifiably robust learning-reasoning conformal prediction framework (COLEP) via probabilistic circuits, which comprises a data-driven *learning component* that trains statistical models to learn different semantic concepts, and a *reasoning component* that encodes knowledge and characterizes the relationships among the statistical knowledge models for logic reasoning. To achieve exact and efficient reasoning, we employ probabilistic circuits (PCs) to construct the reasoning component. Theoretically, we provide end-to-end certification of prediction coverage for COLEP under $\ell_2$ bounded adversarial perturbations. We also provide certified coverage considering the finite size of the calibration set. Furthermore, we prove that COLEP achieves higher prediction coverage and accuracy over a single model as long as the utilities of knowledge models are non-trivial. Empirically, we show the validity and tightness of our certified coverage, demonstrating the robust conformal prediction of COLEP on various datasets.

## 1 Introduction

Deep neural networks (DNNs) have demonstrated impressive achievements in different domains (He et al., 2016; Vaswani et al., 2017; Li et al., 2022b; Pan et al., 2019; Chen et al., 2018). However, as DNNs are increasingly employed in real-world applications, particularly those safety-critical ones such as autonomous driving (Xu et al., 2022a) and medical diagnosis (Kang et al., 2023a), concerns regarding their trustworthiness and reliability have emerged (Szegedy et al., 2014; Eykholt et al., 2018; Cao et al., 2021; Li et al., 2017; Wang et al., 2023). To address these concerns, it is crucial to provide the *worst-case certification* for the model predictions (i.e., certified robustness) (Wong & Kolter, 2018; Cohen et al., 2019; Li et al., 2023), and develop techniques that enable users to statistically evaluate the *uncertainty* linked to the model predictions (i.e., conformal prediction) (Lei et al., 2018; Solari & Djordjilović, 2022).

In particular, considering potential adversarial manipulations during test time, where a small perturbation could mislead the models to make incorrect predictions (Szegedy et al., 2014; Madry et al., 2018; Xiao et al., 2018; Cao et al., 2021; Kang et al., 2024b), different robustness certification approaches have been explored to provide the worst-case prediction guarantees for neural networks. On the other hand, conformal prediction has been studied as an effective tool for generating prediction sets that reflect the prediction uncertainty of a given black-box model, assuming that the data is exchangeable (Romano et al., 2020; Vovk et al., 2005). Such prediction *uncertainty* has provided a probabilistic guarantee in addition to the *worst-case certification*. However, it is unclear how such *uncertainty* certification would perform in the worst-case adversarial environments.

Although the worst-case certification for conformal prediction is promising, it is challenging for purely data-driven models to achieve high **certified prediction coverage (i.e., worst-case coverage with bounded input perturbations)**. Recent studies on the learning-reasoning framework, which integrates extrinsic domain knowledge and reasoning capabilities into data-driven models, have demonstrated great success in improving the model worst-case certifications (Yang et al., 2022; Zhang et al., 2023). In this paper, we aim to bridge the *worst-case robustness certification* and *uncertainty* certification, and explore whether such knowledge-enabled logical reasoning could help improve the certified prediction coverage for conformal prediction. We ask: *How to efficiently integrate knowledge and reasoning capabilities into DNNs for conformal prediction? Can we prove that such knowledge and logical reasoning enabled framework would indeed achieve higher certified prediction coverage and accuracy than that of a single DNN?*

In this work, we first propose a certifiably robust learning-reasoning conformal prediction framework (COLEP) via probabilistic circuits (PCs). In particular, we train different data-driven DNNs in the "learning" component, whose logical relationships (e.g., a stop sign should be of octagon shape) are encoded in the "reasoning" component, which is constructed with probabilistic circuits (Darwiche, 1999; 2001) for logical reasoning as shown in Fig 1. Theoretically, we first derive the *certified coverage* for COLEP under $\ell_2$ bounded perturbations. We also provide *certified coverage* considering the finite size of the calibration set. We then prove that COLEP achieves higher certified coverage and accuracy than that of a single model as long as the knowledge models have non-trivial utilities. To our best knowledge, this is the first knowledge-enabled learning framework for conformal prediction and with provably higher certified coverage in adversarial settings.

We have conducted extensive experiments on GTSRB, CIFAR-10, and AwA2 datasets to demonstrate the effectiveness and tightness of the certified coverage for COLEP. We show that the certified prediction coverage of COLEP is significantly higher compared with the SOTA baselines, and COLEP has weakened the tradeoff between prediction coverage and prediction set size. We perform a range of ablation studies to show the impacts of different types of knowledge.

**Related work** Conformal prediction is a statistical tool to construct the prediction set with guaranteed prediction coverage (Vovk et al., 1999; 2005; Shafer & Vovk, 2008; Lei et al., 2013; Yang & Kuchibhotla, 2021; Solari & Djordjilović, 2022; Jin et al., 2023), assuming that the data is exchangeable. Several works have explored the scenarios where the exchangeability assumption is violated (Jin et al., 2023; Barber et al., 2022; Ghosh et al., 2023; Kang et al., 2023c; 2024a). However, it is unclear how to provide the worst-case certification for such prediction coverage considering adversarial setting. Recently, Gendler et al. (Gendler et al., 2022) provided a certified method for conformal prediction by applying randomized smoothing to certify the non-conformity scores. However, such certified coverage could be loose due to the generic nature of randomized smoothing. We propose COLEP with a sensing-reasoning framework to achieve much stronger certified coverage by leveraging the knowledge-enabled logic reasoning capabilities. We provide a comprehensive literature review on certified robustness and logical reasoning in appendix B.

## 2 PRELIMINARIES

Suppose that we have $n$ data samples $\{(X_i, Y_i)\}_{i=1}^n$ with features $X_i \in \mathbb{R}^d$ and labels $Y_i \in \mathcal{Y} := \{1, 2, ..., N_c\}$. Assume that the data samples are drawn exchangeably from some unknown distribution $P_{XY}$. Given a desired coverage $1 - \alpha \in (0, 1)$, conformal prediction methods construct a prediction set $\hat{C}_{n,\alpha} \subseteq \mathcal{Y}$ for a new data sample $(X_{n+1}, Y_{n+1}) \sim P_{XY}$ with the guarantee of *marginal prediction coverage*: $\mathbb{P}[Y_{n+1} \in \hat{C}_{n,\alpha}(X_{n+1})] \geq 1 - \alpha$.

In this work, we focus on the split conformal prediction setting (Lei et al., 2018; Solari & Djordjilović, 2022), where the data samples are randomly partitioned into two disjoint sets: a training set $\mathcal{I}_{\text{tr}}$ and a calibration set $\mathcal{I}_{\text{cal}} = [n] \backslash \mathcal{I}_{\text{tr}}$. In here and what follows, $[n] := \{1, \cdots, n\}$. We fit a classifier $h(x) : \mathbb{R}^d \mapsto \mathcal{Y}$ to the training set $\mathcal{I}_{\text{tr}}$ to estimate the **conditional class probability** $\pi_y(x) := \mathbb{P}[Y = y | X = x]$ of $Y$ given $X$. Using the estimated probabilities that we denote by $\hat{\pi}_y(x)$, we then compute a non-conformity score $S_{\hat{\pi}_y}(X_i, Y_i)$ for each sample in the calibration set $\mathcal{I}_{\text{cal}}$ to measure how much non-conformity each validation sample has with respect to its ground truth label. A commonly used non-conformity score for valid and adaptive coverage is introduced by (Romano et al., 2020) as: $S_{\hat{\pi}_y}(x, y) = \sum_{j \in \mathcal{Y}} \hat{\pi}_j(x) \mathbb{I}_{[\hat{\pi}_j(x) > \hat{\pi}_y(x)]} + \hat{\pi}_y(x) u$, where $\mathbb{I}_{[\cdot]}$ is the indicator function and $u$ is uniformly sampled over the interval $[0, 1]$. Given a desired coverage $1 - \alpha$, the prediction set of a new data point $X_{n+1}$ is formulated as:

$$\hat{C}_{n,\alpha}(X_{n+1}) = \{y \in \mathcal{Y} : S_{\hat{\pi}_y}(X_{n+1}, y) \leq Q_{1-\alpha}(\{S_{\hat{\pi}_y}(X_i, Y_i)\}_{i \in \mathcal{I}_{\text{cal}}})\} \quad (1)$$

where $Q_{1-\alpha}(\{S_{\hat{\pi}_y}(X_i, Y_i)\}_{i \in \mathcal{I}_{\text{cal}}})$ is the $\lceil(1-\alpha)(1+|\mathcal{I}_{\text{cal}}|)\rceil$-th largest value of the set $\{S_{\hat{\pi}_y}(X_i, Y_i)\}_{i \in \mathcal{I}_{\text{cal}}}$. The prediction set $\hat{C}_{n,\alpha}(X_{n+1})$ includes all the labels with a smaller non-conformity score than the $(1 - \alpha)$-quantile of scores in the calibration set. Since we assume the data samples are exchangeable, the marginal coverage of the prediction set $\hat{C}_{n,\alpha}(X_{n+1})$ is no less than $1 - \alpha$.

## 3 LEARNING-REASONING PIPELINE FOR CONFORMAL PREDICTION

To achieve high certified prediction coverage against adversarial perturbations, we introduce the certifiably robust learning-reasoning conformal prediction framework (COLEP). For better readability, we provide structured tables for used notations through the analysis in Appendix K.

### 3.1 OVERVIEW OF COLEP

The COLEP is comprised of a data-driven *learning component* and a logic-driven *reasoning component*. The learning component is equipped with a main model to perform the main task of classification and $L$ knowledge models, each learns different concepts from data. Following the learning component is the reasoning component, which consists of $R$ subcomponents (e.g., PCs) responsible for encoding diverse domain knowledge and logical reasoning by characterizing the logical relationships among the learning

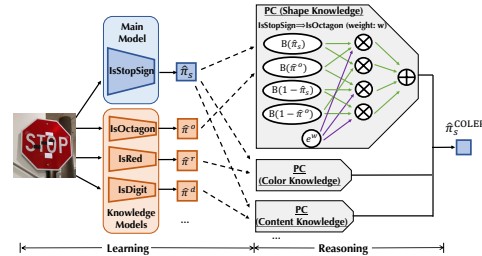

Figure 1: Overview of COLEP.

models. The COLEP framework is depicted in Fig 1. Concretely, consider an input $x \in \mathbb{R}^d$ (e.g., the stop sign image) and an untrusted main model (e.g., stop sign classification model) within the learning component that predicts the label of $x$ along with estimates of conditional class probabilities

$$\hat{\pi}_j(x) := \mathbb{P}[\hat{y} = j | X = x], \tag{2}$$

where $j \in \mathcal{Y} := [N_c]$. Note that the main model as a $N_c$-class classification model can be viewed as $N_c$ binary classification models, each with success probability $\hat{\pi}_{j \in [N_c]}(\cdot)$. In Fig 1, we particularly illustrate COLEP using a binary stop sign classifier $\hat{\pi}_s(\cdot)$ as the main model for simplicity. Each knowledge model (e.g., octagon shape classification model $\hat{\pi}^o(\cdot)$) learns a concept from $x$ (e.g., octagon shape) and outputs its prediction denoted by $\hat{y}^{(l)} \in \{0, 1\}$ where $l \in [L]$ indexes a particular knowledge model. There can be only two possible outcomes: whether the $l$-th knowledge model detects the concept $\hat{y}^{(l)} = 1$ or not $\hat{y}^{(l)} = 0$. Consistently with the notation of the main model, we denote the conditional concept probability of the $l$-th knowledge model by

$$\hat{\pi}^{(l)}(x) := \mathbb{P}[\hat{y}^{(l)} = 1 | X = x] \tag{3}$$

for $l \in [L]$ Once the learning models output their predictions along with estimated probabilities, the reasoning component accepts these estimates as input and passes them through $R$ distinct reasoning subcomponents in parallel. It then combines the probabilities at the outputs of the reasoning subcomponents to calculate the corrected estimates of conditional class probabilities by COLEP, which we denote as $\hat{\pi}_j^{\text{COLEP}}(x)$ for $j \in [N_c]$ (e.g., corrected probability of $x$ having a stop sign or not: $\hat{\pi}_s^{\text{COLEP}}(x)$ or $1\text{-}\hat{\pi}_s^{\text{COLEP}}(x)$) and form the conformal prediction set with them.

### 3.2 REASONING VIA PROBABILISTIC CIRCUITS

We encode two types of knowledge in the reasoning component: (1) *preventive knowledge* of the form "$A \implies B$" (e.g., "*IsStopSign $\implies$ IsOctagon*"), and (2) *permissive knowledge* of the form "$B \implies A$" (e.g., "*HasContentStop $\implies$ IsStopSign*"), where $A$ represents the main task corresponding to the main model (e.g., "*IsStopSign*"), $B$ as one of the $L$ knowledge models (e.g., "*IsOctagon*"), and $\implies$ denotes logical implication with a *conditional variable* as the left operand and a *consequence variable* on the right. In this part, we consider one particular reasoning subcomponent (i.e., one PC) and encode $H$ preventive and permissive knowledge rules $\{K_h\}_{h=1}^H$ in it. The $h$-th *knowledge rule* $K_h$ is assigned with weight $w_h \in \mathbb{R}^+$.

To enable exact and efficient reasoning, we need to seek for appropriate probabilistic models as the reasoning component. Recent work (Gürel et al., 2021; Yang et al., 2022) encode propositional logic rules with Markov Logic Networks (MLNs) (Richardson & Domingos, 2006), but the inference complexity is exponential regarding the number of knowledge models. Variational inference is efficient for reaonsing (Zhang et al., 2023), but induces error during reasoning. Thus, we use probabilistic circuits (PCs) to achieve better trade-offs between the model expressiveness and reasoning efficiency.

**Encode preventive and permissive knowledge rules with PCs**. PCs define a joint distribution over a set of random variables. PCs take an assignment of the random variables as input and output the joint probability of the assignment. Specifically, the input of PCs can be represented by an $N_c + L$ sized vector of Bernoulli random variables (e.g., [*IsStopSign*, *IsOctagon*, *IsRed*, *IsDigit*]) with the corresponding vector of success probabilities (e.g., $\hat{\pi} = [\hat{\pi}_s, \hat{\pi}^o, \hat{\pi}^r, \hat{\pi}^d]$) denoted as:

$$\hat{\pi} := \left[[\hat{\pi}_j]_{j \in [N_c]}, [\hat{\pi}^{(l)}]_{l \in [L]}\right]. \tag{4}$$

Note that $\hat{\pi}$ is a concatenated vector of the estimated conditional class and concept probabilities of the main model and knowledge models, respectively (e.g., $\hat{\pi} = [\hat{\pi}_s, \hat{\pi}^o, \hat{\pi}^r, \hat{\pi}^d]$ with $N_c = 1, L = 2$). Since conformal prediction is our main focus, the conditional probabilities $\hat{\pi}$ in eq. (4) will be frequently referenced throughout the paper. To distinguish the index representing each class label $j \in [N_c]$, we use $j_\forall \in [N_c + L]$ to address entries of $N_c + L$ sized vectors.

The computation of PCs is based on an acyclic directed graph $\mathcal{G}$. Each leaf node $O_{\text{leaf}}$ in $\mathcal{G}$ represents a univariate distribution (e.g., $B(\hat{\pi}_s(x))$: Bernoulli distribution over *IsStopSign* with success rate $\hat{\pi}_s(x)$). The output of a PC is typically the root node $O_{\text{root}}$ representing the joint probability of an assignment (e.g., $p(IsStopSign = 1, IsOctagon = 1|x)$). The graph $\mathcal{G}$ contains internal sum nodes computing a convex combination of children nodes, and product nodes computing the product of them.

Formally, we let the leaf node $O_{\text{leaf}}^{(j_\forall)}$ in the computational graph $\mathcal{G}$ be a Bernoulli distribution with success probability $\hat{\pi}_{j_\forall}(x)$. Let $\mu \in M := \{0, 1\}^{N_c+L}$ be the index variable (not an observation) of a possible assignment for $N_c$ classes in the interest of the main model and $L$ concepts in the interest of $L$ knowledge models. We also define the factor function $F : \{0, 1\}^{N_c+L} \mapsto \mathbb{R}^+$ which takes as input a possible assignment $\mu$ and outputs the factor value of the assignment:

$$F(\mu) = \exp\left\{\sum_{h=1}^{H} w_h \mathbb{I}_{[\mu \sim K_h]}\right\}, \tag{5}$$

where $\mathbb{I}_{[\mu \sim K_h]} = 1$ indicates that the assignment $\mu$ follows the given logic rule $K_h$, and otherwise $\mathbb{I}_{[\mu \sim K_h]} = 0$ (e.g., in the shape PC in fig. 1, $F([1, 1]) = e^w$, $\mathbb{I}_{[[1,1] \sim K]} = 1$, where $K$ is the rule "*IsStopSign* $\implies$ *IsOctagon*" with weight $w$). $F(\mu)$ essentially measures how well the assignment $\mu$ conforms to the specified knowledge rules. Following (Darwiche, 2002; 2003), we let sum nodes compute the logical "or" operation, and product nodes compute the logical "and" operation. We associate each assignment $\mu \in M$ with the factor value $F(\mu)$ at the level of leaf nodes as fig. 1.

Given a feasible assignment of random variables, the output of PC $O_{\text{root}}$ computes the likelihood of the assignment (e.g., $p(IsStopSign = 1, IsOctogon = 1|x)$). When considering the likelihood of the $j$-th class label, we can marginalize class labels other than $j$ and formulate the class probability conditioned on input $x$ as:

$$\hat{\pi}_j^{\text{COLEP}}(x) = \frac{\sum_{\mu \in M, \, \mu_j=1} O_{\text{root}}(\mu)}{\sum_{\mu \in M} O_{\text{root}}(\mu)} = \frac{\sum_{\mu \in M, \, \mu_j=1} \exp\left\{\sum_{j_\forall=1}^{N_c+L} T(\hat{\pi}_{j_\forall}(x), \mu_{j_\forall})\right\} F(\mu)}{\sum_{\mu \in M} \exp\left\{\sum_{j_\forall=1}^{N_c+L} T(\hat{\pi}_{j_\forall}(x), \mu_{j_\forall})\right\} F(\mu)}, \tag{6}$$

where $T(a, b) := \log(ab + (1-a)(1-b))$ for simplicity of notations and $\mu_j$ denotes the $j$-th element of vector $\mu$. Note that the numerator and denominator of eq. (6) can be exactly and efficiently computed with a single forwarding pass of PCs (Hitzler & Sarker, 2022; Choi & Darwiche, 2017; Rooshenas & Lowd, 2014). We can also improve the expressiveness of the reasoning component by linearly combining outputs of $R$ PCs $\hat{\pi}_j^{(r)}$ with coefficients $\beta_r$ for the $r$-th PC as $\hat{\pi}_j^{\text{COLEP}}(x) = \sum_{r \in [R]} \beta_r \hat{\pi}_j^{(r)}(x)$. The core of this formulation is the mixture model involving a latent variable $r_{\text{PC}}$ representing the PCs. In short, we can write $\hat{\pi}_j^{\text{COLEP}}(x)$ as the marginalized probability over the latent variable $r_{\text{PC}}$ as $\hat{\pi}_j^{\text{COLEP}}(x) = \sum_{r \in [R]} \mathbb{P}[r_{\text{PC}} = r] \cdot \hat{\pi}_j^{(r)}(x)$. Hence, the coefficient $\beta_r$ for the $r$-th PC are determined by $\mathbb{P}[r_{\text{PC}} = r]$. Although we lack direct knowledge of this probability, we can estimate it using the data by examining how frequently each PC $r$ correctly predicts the outcome across the given examples, similarly as in the estimation of prior class probabilities for Naive Bayes classifiers.

**Illustrative example**. We open up the stop sign classification example in fig. 1 further as follows. The main model predicts that the image $x$ has a stop sign with probability $\hat{\pi}_s(x)$. In parallel to it, the knowledge model that detects octagon shape predicts that the image $x$ has an object with octagon shape with probability $\hat{\pi}_o(x)$. Then we can encode the logic rule "*IsStopSign* $\implies$ *IsOctagon*" with weight $w$ as fig. 1. The reasoning component can correct the output probability with encoded knowledge rules and improve the robustness of the prediction. Consider the following concrete example. Suppose that the adversarial example of a speed limit sign misleads the main classifier

to detect it as a stop sign with a large probability (e.g., $\hat{\pi}_s = 0.9$), but the octagon classifier is not misled and correctly predicts that the speed limit sign has a square shape (e.g., $\hat{\pi}_o = 0.0$). Then according the computation as eq. (6), the corrected probability of detecting a stop sign given the attacked speed limit image is $\hat{\pi}_s^{\text{COLEP}} = 0.9/(0.1e^w + 0.9)$, which is down-weighted from the original wrong prediction $\hat{\pi}_s = 0.9$ as we have a positive logical rule weight $w > 0$. Therefore, the reasoning component can correct the wrong output probabilities with the knowledge rules and lead to a more robust prediction framework.

## 3.3 CONFORMAL PREDICTION WITH COLEP

After obtaining the corrected conditional class probabilities using COLEP in eq. (6), we move on to conformal prediction with COLEP. To begin with, we construct prediction sets for each class label within the learning component. Using eq. (1), the prediction set for the $j$-th class, where $j \in [N_c]$, can be formulated as:

$$\hat{C}_{n,\alpha_j}^{\text{COLEP}j}(X_{n+1}) = \{q^y \in \{0,1\} : S_{\hat{\pi}_j^{\text{COLEP}}}(X_{n+1}, q^y) \leq Q_{1-\alpha_j}(\{S_{\hat{\pi}_j^{\text{COLEP}}}(X_i, \mathbb{I}_{[Y_i=j]})\}_{i \in \mathcal{I}_{\text{cal}}})\} \quad (7)$$

where $1 - \alpha_j$ is the user-defined coverage level, similar to $\alpha$ in the standard conformal prediction. The fact that element 1 is present in $\hat{C}_{n,\alpha_j}^{\text{COLEP}j}(X_{n+1})$ signifies that the class label $j$ is a possible ground truth label for $X_{n+1}$ and can be included in the final prediction set. That is:

$$\hat{C}_{n,\alpha}^{\text{COLEP}}(X_{n+1}) = \{j \in [N_c] : 1 \in \hat{C}_{n,\alpha_j}^{\text{COLEP}j}(X_{n+1})\} \quad (8)$$

where $\alpha = \max_{j \in [N_c]}\{\alpha_j\}$. It is worth noting that if $\alpha_j$ is set to a fixed value for all $j \in [N_c]$, $\alpha$ in the standard conformal prediction setting is recovered. In this work, we adopt a fixed $\alpha$ for each $\alpha_j$ for simplicity, but note that our framework allows setting different miscoverage levels for each class $j$, which is advantageous in addressing imbalanced classes. We defer more discussions on the flexibility of defining class-wise coverage levels to Appendix D.1.

## 4 CERTIFIABLY ROBUST LEARNING-REASONING CONFORMAL PREDICTION

In this section, we provide robustness certification of COLEP for conformal prediction. We start by providing the certification of the reasoning component in Section 4.1, and then move to end-to-end certified coverage as well as finite-sample certified coverage in Section 4.2. We defer all the proofs to Appendix. The overview of the certification framework is provided in fig. 4 in Appendix E.

## 4.1 CERTIFICATION FRAMEWORK AND CERTIFICATION OF THE REASONING COMPONENT

The inference time adversaries violate the data exchangeability assumption, compromising the guaranteed coverage. With COLEP, we aim to investigate (1) whether the guaranteed coverage can be preserved for COLEP using a prediction set that takes the perturbation bound into account, and (2) whether we can provide a worst-case coverage (a lower bound) using the standard prediction set as before in eq. (8). We provide certification for goal (1) and (2) in Thms. 2 and 3, respectively.

The certification procedure can be decomposed into several subsequent stages. First, we **certify learning robustness** by deriving the bounds for $\hat{\pi}$ within the learning component given the input perturbation bound $\delta$. There are several existing approaches to certify the robustness of models within this component (Cohen et al., 2019; Zhang et al., 2018; Gowal et al., 2018). In this work, we use randomized smoothing (Cohen et al., 2019) (details in Appendix C.3), and get the bound of $\hat{\pi}$ under the perturbation. Next, we **certify reasoning robustness** by deriving the reasoning-corrected bounds using the bounds of the learning component (Thm. 1). Finally, we **certify the robustness of conformal prediction** considering the reasoning-corrected bounds (Thm. 2 and Thm. 3).

As robustness of the learning component is performed using randomized smoothing, we directly focus on the robustness certification of reasoning component.

**Theorem 1** (Bounds for Conditional Class Probabilities $\hat{\pi}_j^{\text{COLEP}}(x)$ within the Reasoning Component). *Given any input $x$ and perturbation bound $\delta$, we let $[\underline{\hat{\pi}_{j_\forall}}(x), \overline{\hat{\pi}_{j_\forall}}(x)]$ be bounds for the estimated conditional class and concept probabilities by all models with $j_\forall \in [N_c + L]$ (for example, achieved via randomized smoothing). Let $V_d^j$ be the set of index of conditional variables in the PC except for $j \in [N_c]$ and $V_s^j$ be that of consequence variables. Then the bound for COLEP-corrected estimate of the conditional class probability $\hat{\pi}_j^{\text{COLEP}}$ is given by:*

$$\mathbb{U}[\hat{\pi}_j^{COLEP}(x)] = \left\{ \frac{(1 - \overline{\hat{\pi}_j}(x)) \sum_{\mu_j=0} \exp\left\{ \sum_{j_\forall \in V_d^j} T(\overline{\hat{\pi}_{j_\forall}}(x), \mu_{j_\forall}) + \sum_{j_\forall \in V_s^j} T(\underline{\hat{\pi}_{j_\forall}}(x), \mu_{j_\forall}) \right\} F(\mu)}{\overline{\hat{\pi}_j}(x) \sum_{\mu_j=1} \exp\left\{ \sum_{j_\forall \in V_d^j} T(\hat{\pi}_{j_\forall}(x), \mu_{j_\forall}) + \sum_{j_\forall \in V_s^j} T(\overline{\hat{\pi}_{j_\forall}}(x), \mu_{j_\forall}) \right\} F(\mu)} + 1 \right\}^{-1} \quad (9)$$

*where $T(a, b) = \log(ab + (1-a)(1-b))$. We similarly give the lower bound $\mathbb{L}[\hat{\pi}_j^{COLEP}]$ in Appendix F.1.*

*Remarks.* Thm. 1 establishes a certification connection from the probability bound of the learning component to that of the reasoning component. Applying robustness certification methods to DNNs, we can get the probability bound of the learning component $[\underline{\hat{\pi}}, \overline{\hat{\pi}}]$ within bounded input perturbations. Thm. 1 shows that we can compute the probability bound after the reasoning component $[\mathbb{L}[\hat{\pi}_j^{COLEP}], \mathbb{U}[\hat{\pi}_j^{COLEP}]]$ following eq. (9). Since Equation (9) presents an analytical formulation of the bound after the reasoning component as a function of the bounds before the reasoning component, we can directly and efficiently compute $[\mathbb{L}[\hat{\pi}_j^{COLEP}], \mathbb{U}[\hat{\pi}_j^{COLEP}]]$ in evaluations.

## 4.2 CERTIFIABLY ROBUST CONFORMAL PREDICTION

In this section, we leverage the reasoning component bounds in Thm. 1 to perform calibration with a worst-case non-conformity score function for adversarial perturbations. This way, we perform certifiably robust conformal prediction in the adversary setting during the inference time. We formulate the procedure to achieve this in Thm. 2.

**Theorem 2** (Certifiably Robust Conformal Prediction of COLEP). *Consider a new test sample $X_{n+1}$ drawn from $P_{XY}$. For any bounded perturbation $\|\epsilon\|_2 \leq \delta$ in the input space and the adversarial sample $\tilde{X}_{n+1} := X_{n+1} + \epsilon$, we have the following guaranteed marginal coverage:*

$$\mathbb{P}[Y_{n+1} \in \hat{C}_{n,\alpha}^{COLEP\delta}(\tilde{X}_{n+1})] \geq 1 - \alpha \quad (10)$$

*if we construct the certified prediction set of COLEP where*

$$\hat{C}_{n,\alpha}^{COLEP\delta}(\tilde{X}_{n+1}) = \left\{ j \in [N_c] : S_{\hat{\pi}_j^{COLEP}}(\tilde{X}_{n+1}, 1) \leq Q_{1-\alpha}(\{S_{\hat{\pi}_j^{COLEP\delta}}(X_i, \mathbb{I}_{[Y_i=j]})\}_{i \in \mathcal{I}_{cal}}) \right\} \quad (11)$$

*and $S_{\hat{\pi}_j^{COLEP\delta}}(\cdot, \cdot)$ is a function of worst-case non-conformity score considering perturbation radius $\delta$:*

$$S_{\hat{\pi}_j^{COLEP\delta}}(X_i, \mathbb{I}_{[Y_i=j]}) = \begin{cases} \mathbb{U}_\delta[\hat{\pi}_j^{COLEP}(X_i)] + u(1 - \mathbb{U}_\delta[\hat{\pi}_j^{COLEP}(X_i)]), & Y_i \neq j \\ 1 - \mathbb{L}_\delta[\hat{\pi}_j^{COLEP}(X_i)] + u\mathbb{L}_\delta[\hat{\pi}_j^{COLEP}(X_i)], & Y_i = j \end{cases} \quad (12)$$

*with $\mathbb{U}_\delta[\hat{\pi}_j^{COLEP}(x)] = \max_{|\eta|_2 \leq \delta} \hat{\pi}_j^{COLEP}(x + \eta)$ and $\mathbb{L}_\delta[\hat{\pi}_j^{COLEP}(x)] = \min_{|\eta|_2 \leq \delta} \hat{\pi}_j^{COLEP}(x + \eta)$.*

*Remarks.* Thm. 2 shows that the coverage guarantee of COLEP in the adversary setting is still valid (eq. (10)) if we construct the prediction set by considering the worst-case perturbation as in eq. (11). That is, the prediction set of COLEP in eq. (11) covers the ground truth of an adversarial sample with nominal level $1 - \alpha$. To achieve that, we use a worst-case non-conformity score as in eq. (12) during calibration to counter the influence of adversarial sample during inference. The bound of output probability $(\mathbb{U}_\delta[\hat{\pi}_j^{COLEP}], \mathbb{L}_\delta[\hat{\pi}_j^{COLEP}])$ in eq. (12) can be computed by Thm. 1 to achieve end-to-end robustness certification of COLEP, which generally holds for any DNN and probabilistic circuits. We also consider the finite-sample errors of using randomized smoothing for the learning component certification as Anonymous (2023) and provided the theoretical statement and proofs in thm. 6 in Appendix F.3.

**Theorem 3** (Certified (Worst-Case) Coverage of COLEP). *Consider the new sample $X_{n+1}$ drawn from $P_{XY}$ and adversarial sample $\tilde{X}_{n+1} := X_{n+1} + \epsilon$ with any perturbation $\|\epsilon\|_2 \leq \delta$ in the input space. We have:*

$$\mathbb{P}[Y_{n+1} \in \hat{C}_{n,\alpha}^{COLEP}(\tilde{X}_{n+1})] \geq \tau^{COLEP_{cer}} := \min_{j \in [N_c]} \left\{ \tau_j^{COLEP_{cer}} \right\}, \quad (13)$$

*where the certified (worst-case) coverage of the $j$-th class label $\tau_j^{COLEP_{cer}}$ is formulated as:*

$$\tau_j^{COLEP_{cer}} = \max \left\{ \tau : Q_\tau(\{S_{\hat{\pi}_j^{COLEP\delta}}(X_i, \mathbb{I}_{[Y_i=j]})\}_{i \in \mathcal{I}_{cal}}) \leq Q_{1-\alpha}(\{S_{\hat{\pi}_j^{COLEP}}(X_i, \mathbb{I}_{[Y_i=j]})\}_{i \in \mathcal{I}_{cal}}) \right\}. \quad (14)$$

*Remarks.* Thm. 2 constructs a certified prediction set with $1 - \alpha$ coverage as eq. (11) by considering the worst-case perturbation during calibration to counter the influence of test-time adversaries. In contrast, Thm. 3 targets the case when we still adopt a standard calibration process without the consideration of test-time perturbations, which would lead to a lower coverage than $1 - \alpha$ and Thm. 3 concretely shows the formulation of the worst-case coverage. Thm. 3 presents the formulations of the lower bound of the coverage $(\tau_j^{COLEP_{cer}})$ with the standard prediction set as eq. (8) in the adversary setting. In addition to the certified coverage, we consider finite calibration set size and certified coverage with finite sample errors in Appendix G.

# 5 THEORETICAL ANALYSIS OF COLEP

In Section 5.3, we theoretically show that COLEP achieves better coverage and prediction accuracy than a single main model as long as the utility of knowledge models and rules (characterized in Section 5.1) is non-trivial. To achieve it, we provide a connection between those utilities and the effectiveness of PCs in Section 5.2. The certification overview is provided in fig. 4 in Appendix E.

## 5.1 CHARACTERIZATION OF MODELS AND KNOWLEDGE RULES

We consider a mixture of benign distribution $\mathcal{D}_b$ and adversarial distribution $\mathcal{D}_a$ denoted by $\mathcal{D}_m := p_{\mathcal{D}_b}\mathcal{D}_b + p_{\mathcal{D}_a}\mathcal{D}_a$ with $p_{\mathcal{D}_b} + p_{\mathcal{D}_a} = 1$. We map a set of implication rules encoded in the $r$-th PC to an undirected graph $\mathcal{G}_r = (\mathcal{V}_r, \mathcal{E}_r)$, where $\mathcal{V}_r = [N_c + L]$ corresponds to $N_c$ (binary) class labels and $L$ (binary) concept labels. There exists an edge between node $j_1 \in \mathcal{V}_r$ and node $j_2 \in \mathcal{V}_r$ iff they are connected by an implication rule $j_1 \implies j_2$ or $j_2 \implies j_1$.

Let $\mathcal{A}_r(j_\forall)$ be the set of nodes in the connected component of node $j_\forall$ in graph $\mathcal{G}_r$. Let $\mathcal{A}_{r,d}(j_\forall) \subseteq \mathcal{A}_r(j_\forall)$ be the set of conditional variables and $\mathcal{A}_{r,s}(j_\forall) \subseteq \mathcal{A}_r(j_\forall)$ be the set of consequence variables. For any sample $x$, let $\nu(x) \in [0,1]^{N_c+L}$ be the Boolean vector of ground truth class and concepts such that $\nu(x) := \left[[\mathbb{I}_{[y=j]}]_{j \in [N_c]}, [\mathbb{I}_{[y^{(l)}=1]}]_{l \in [L]}\right]$. We denote $\nu(x)$ as $\nu$ when sample $x$ is clear in the context. For simplicity, we assume a fixed weight $w \in \mathbb{R}^+$ for all knowledge rules in the analysis. In what follows, we characterize the prediction capability of each model (main or knowledge) and establish a relation between these characteristics and the prediction performance of COLEP.

For each class and concept probability estimated by learning models $\hat{\pi}_{j_\forall \in [N_c+L]} : \mathbb{R}^d \mapsto [0,1]$ (recall eq. (4)), we have the following definition:

$$\mathbb{P}_\mathcal{D}\left[\hat{\pi}_{j_\forall}(x) \leq 1 - t_{j_\forall,\mathcal{D}} \,|\, \nu_{j_\forall} = 0\right] \geq z_{j_\forall,\mathcal{D}}, \quad \mathbb{P}_\mathcal{D}\left[\hat{\pi}_{j_\forall}(x) \geq t_{j_\forall,\mathcal{D}} \,|\, \nu_{j_\forall} = 1\right] \geq z_{j_\forall,\mathcal{D}} \tag{15}$$

where $t_{j_\forall,\mathcal{D}}$, $z_{j_\forall,\mathcal{D}}$ are parameters quantifying quality of the main model and knowledge models in predicting correct class and concept, respectively. In particular, $t_{j_\forall,\mathcal{D}}$ characterizes the threshold of confidence whereas $z_{j_\forall,\mathcal{D}}$ quantifies the probability matching this threshold over data distribution $\mathcal{D}$. Specifically, models with a large $t_{j_\forall,\mathcal{D}}, z_{j_\forall,\mathcal{D}}$ imply that the model outputs a small class probability when the ground truth of input is 0 and a large class probability when the ground truth of input is 1, indicating the model performs well on prediction. We define the combined utility of models within $r$-th PC by using individual model qualities. For $j$-th class, we have:

$$T_{j,\mathcal{D}}^{(r)} = \Pi_{j_\forall \in \mathcal{A}_r(j)} t_{j_\forall,\mathcal{D}}, \quad Z_{j,\mathcal{D}}^{(r)} = \Pi_{j_\forall \in \mathcal{A}_r(j)} z_{j_\forall,\mathcal{D}} \tag{16}$$

and the utility of associated knowledge rules within $r$-th PC for the $j$-th class as:

$$U_j^{(r)} = \begin{cases} \mathbb{P}\left[\nu_{j_\forall} = 0, \exists j_\forall \in \mathcal{A}_{r,s}(j) \,|\, \nu_j = 0\right], & r \in \mathcal{P}_d(j) \\ \mathbb{P}\left[\nu_{j_\forall} = 1, \exists j_\forall \in \mathcal{A}_{r,d}(j) \,|\, \nu_j = 1\right], & r \in \mathcal{P}_s(j) \end{cases} \tag{17}$$

where $\mathcal{P}_d(j)$ and $\mathcal{P}_s(j)$ are PC index sets where $j$ appears as conditional and consequence variables. $U_j^{(r)}$ quantifies the utility of knowledge rules for class $j$. Consider the case where $j$ corresponds to a conditional variable in the $r$-th PC. Small $U_j^{(r)}$ indicates that $\mathbb{P}\left[\nu_{j_\forall} = 1, \forall j_\forall \in \mathcal{A}_{r,s}(j) \,|\, \nu_j = 0\right]$ is high, hence implication rules for $j_\forall$ are universal and of low utility.

## 5.2 EFFECTIVENESS OF THE REASONING COMPONENT WITH PCS

Here we theoretically build the connection between the utility of knowledge models $T_{j_\forall,\mathcal{D}}^{(r)}, Z_{j_\forall,\mathcal{D}}^{(r)}$ and the utility of implication rules $U_{j_\forall}^{(r)}$ with the effectiveness of reasoning components. We quantify the effectiveness of the reasoning component as the relative confidence in ground truth class probabilities. Specifically, we expect $\mathbb{E}\left[\hat{\pi}_j^{\text{COLEP}}(X) - \hat{\pi}_j(X) \,|\, Y = j\right]$ to be large for $j \in [N_c]$ as it enables COLEP to correct the prediction of the main model, especially in the adversary setting.

**Lemma 5.1** (Effectiveness of the Reasoning Component). *For any class probability within models $j \in [N_c]$ and data sample $(X, Y)$ drawn from the mixture distribution $\mathcal{D}_m$, we can prove that:*

$$\mathbb{E}\left[\hat{\pi}_j^{COLEP}(X) \,|\, Y \neq j\right] \leq \mathbb{E}[\hat{\pi}_j(X) - \epsilon_{j,0} \,|\, Y \neq j], \mathbb{E}\left[\hat{\pi}_j^{COLEP}(X) \,|\, Y = j\right] \geq \mathbb{E}\left[\hat{\pi}_j(X) + \epsilon_{j,1} \,|\, Y = j\right] \tag{18}$$

$$\epsilon_{j,0} = \sum_{\mathcal{D} \in \{\mathcal{D}_a, \mathcal{D}_b\}} p_\mathcal{D} \left[\sum_{r \in \mathcal{P}_d(j)} \beta_r U_j^{(r)} Z_{j,\mathcal{D}}^{(r)} \left(\hat{\pi}_j - \frac{\hat{\pi}_j}{\hat{\pi}_j + (1 - \hat{\pi}_j)/\lambda_{j,\mathcal{D}}^{(r)}}\right) + \sum_{r \in \mathcal{P}_s(j)} \beta_r \left(\hat{\pi}_j - \frac{\hat{\pi}_j}{\hat{\pi}_j + (1 - \hat{\pi}_j)T_{j,\mathcal{D}}^{(r)}}\right)\right]$$

$$\epsilon_{j,1} = \sum_{\mathcal{D} \in \{\mathcal{D}_a, \mathcal{D}_b\}} p_\mathcal{D} \left[\sum_{r \in \mathcal{P}_d(j)} \beta_r \left(\frac{\hat{\pi}_j}{\hat{\pi}_j + (1 - \hat{\pi}_j)/T_{j,\mathcal{D}}^{(r)}} - \hat{\pi}_j\right) + \sum_{r \in \mathcal{P}_s(j)} \beta_r U_j^{(r)} Z_{j,\mathcal{D}}^{(r)} \left(\frac{\hat{\pi}_j}{\hat{\pi}_j + (1 - \hat{\pi}_j)\lambda_{j,\mathcal{D}}^{(r)}} - \hat{\pi}_j\right)\right]$$

$$\tag{19}$$

*where $\lambda_{j,\mathcal{D}}^{(r)} = (1/T_{j,\mathcal{D}}^{(r)} + e^{-w} - 1)$ and we shorthand $\hat{\pi}_j(X)$ as $\hat{\pi}_j$.*

*Remarks.* Lemma 5.1 analyzes the relative confidence of the ground truth class probabilities after corrections with the reasoning component. Towards that, we consider different cases based on the utility of the models and knowledge rules described earlier and combine conditional upper bounds with probabilities. We can quantify the effectiveness of the reasoning component with $\epsilon_{j,0}, \epsilon_{j,1}$ since non-trivial $\epsilon_{j,0}, \epsilon_{j,1}$ can correct the confidence of ground truth class in the expected direction and benefit conformal prediction by inducing a smaller non-conformity score for an adversarial sample.

## 5.3 COMPARISON BETWEEN COLEP AND SINGLE MAIN MODEL

Here we reveal the superiority of COLEP over a single main model in terms of the certified prediction coverage and prediction accuracy. Let $\epsilon_{j,0,\mathcal{D}} = \mathbb{E}_{(X,Y) \sim \mathcal{D}}[\epsilon_{j,0}|\nu_j = 0]$ and $\epsilon_{j,1,\mathcal{D}} = \mathbb{E}_{(X,Y) \sim \mathcal{D}}[\epsilon_{j,1}|\nu_j = 1]$. In Section 5.2, we show that the utility of models and knowledge rules is crucial in forming an understanding of COLEP and $\epsilon_{j,0} > 0, \epsilon_{j,1} > 0$ holds. In similar spirit, $\epsilon_{j,0,\mathcal{D}} > 0$ and $\epsilon_{j,1,\mathcal{D}} > 0$ for $j \in [N_c]$ and $\mathcal{D} \in \{\mathcal{D}_a, \mathcal{D}_b\}$ in the subsequent analysis.

**Theorem 4** (Comparison of Marginal Coverage of COLEP and Main Model). *Consider the adversary setting that the calibration set $\mathcal{I}_{cal}$ consists of $n_{\mathcal{D}_b}$ samples drawn from the benign distribution $\mathcal{D}_b$, while the new sample $(X_{n+1}, Y_{n+1})$ is drawn $n_{\mathcal{D}_a}$ times from the adversarial distribution $\mathcal{D}_a$. Assume that $A(\hat{\pi}_j, \mathcal{D}_a) < 0.5 < A(\hat{\pi}_j, \mathcal{D}_b)$ for $j \in [N_c]$, where $A(\hat{\pi}_j, \mathcal{D})$ is the expectation of prediction accuracy of $\hat{\pi}_j$ on $\mathcal{D}$. Then we have:*

$$\mathbb{P}[Y_{n+1} \in \hat{C}_{n,\alpha}^{COLEP}(\tilde{X}_{n+1})] > \mathbb{P}[Y_{n+1} \in \hat{C}_{n,\alpha}(\tilde{X}_{n+1})], \quad w.p.$$

$$1 - \max_{j \in [N_c]} \{\exp\{-2n_{\mathcal{D}_a}(0.5 - A(\hat{\pi}_j, \mathcal{D}_a))^2 \epsilon_{j,1,\mathcal{D}_a}^2\} + n_{\mathcal{D}_b}\exp\{-2n_{\mathcal{D}_b}((A(\hat{\pi}_j, \mathcal{D}_b) - 0.5)\sum_{c \in \{0,1\}} p_{jc}\epsilon_{j,c,\mathcal{D}_b})^2\}\}$$

*where $p_{j0} = \mathbb{P}_{\mathcal{D}_b}[\mathbb{I}_{[Y \neq j]}]$ and $p_{j1} = \mathbb{P}_{\mathcal{D}_b}[\mathbb{I}_{[Y=j]}]$ are class probabilities on benign distribution.* (20)

*Remarks.* Thm. 4 shows that COLEP can achieve better marginal coverage than a single model with a high probability exponentially approaching 1. The probability increases in particular with a higher quality of models represented by $\epsilon_{j,1,\mathcal{D}_a}, \epsilon_{j,c,\mathcal{D}_b}, A(\hat{\pi}_j, \mathcal{D}_b)$. $\epsilon_{j,1,\mathcal{D}_a}, \epsilon_{j,c,\mathcal{D}_b}$ quantifies the effectiveness of the correction of the reasoning component and is positively correlated with the utility of models $T_{j,\mathcal{D}}^{(r)}, Z_{j,\mathcal{D}}^{(r)}$ and the utility of knowledge rules $U_j^{(r)}$ as shown in Lemma 5.1, indicating that better knowledge models and rules benefits a higher prediction coverage with COLEP. The probability also increases with lower accuracy on the adversarial distribution $A(\hat{\pi}_j, \mathcal{D}_a)$, indicating COLEP improves marginal coverage more likely in a stronger adversary setting.

**Theorem 5** (Comparison of Prediction Accuracy of COLEP and Main Model). *Suppose that we evaluate the expected prediction accuracy of $\hat{\pi}_j^{COLEP}(\cdot)$ and $\hat{\pi}_j(\cdot)$ on $n$ samples drawn from $\mathcal{D}_m$ and denote the prediction accuracy as $A(\hat{\pi}_j^{COLEP}(\cdot), \mathcal{D}_m)$ and $A(\hat{\pi}_j(\cdot), \mathcal{D}_m)$. Then we have:*

$$A(\hat{\pi}_j^{COLEP}(\cdot), \mathcal{D}_m) \geq A(\hat{\pi}_j(\cdot), \mathcal{D}_m), \quad w.p. \ 1 - \sum_{\mathcal{D} \in \{\mathcal{D}_a, \mathcal{D}_b\}} p_{\mathcal{D}} \sum_{c \in \{0,1\}} \mathbb{P}_{\mathcal{D}}[Y = j] \exp\{-2n(\epsilon_{j,c,\mathcal{D}})^2\}. \ (21)$$

*Remarks.* Thm. 5 shows that COLEP achieves better prediction accuracy than the main model with a high probability exponentially approaching 1. The probability increases with more effectiveness of the reasoning component quantified by a large $\epsilon_{j,c,\mathcal{D}}$, which positively correlates to the utility of models $T_{j,\mathcal{D}}^{(r)}, Z_{j,\mathcal{D}}^{(r)}$ and the utility of knowledge rules $U_j^{(r)}$ as shown in Lemma 5.1, indicating that better knowledge models and rules benefits a higher prediction accuracy with COLEP. In Appendix I, we further show that COLEP achieves higher prediction accuracy with more useful knowledge rules.

## 6 EXPERIMENTS

We evaluate COLEP on certified conformal prediction in the adversarial setting on various datasets, including GTSRB (Stallkamp et al., 2012), CIFAR-10, and AwA2 (Xian et al., 2018). For fair comparisons, we use the same model architecture and parameters in COLEP and baselines CP (Romano et al., 2020) and RSCP (Gendler et al., 2022). The desired coverage is set $0.9$ across evaluations. Note that we leverage randomized smoothing for learning component certification (100k Monte-Carlo sampling) and consider the finite-sample errors of RSCP following Anonymous (2023) and that of COLEP following Thm. 6. We fix weights $w$ as $1.5$ and provide more details in Appendix J.

**Construction of the reasoning component**. In each PC, we encode a type of implication knowledge with disjoint attributes. In GTSRB, we have a PC of shape knowledge (e.g.,"octagon", "square"), a PC of boundary color knowledge (e.g., "red boundary", "black boundary"), and a PC of content knowledge (e.g., "digit 50", "turn left"). Specifically, in the PC of shape knowledge, we can encode the implication rules such as IsStopSign $\implies$ IsOctagon (stop signs are octagon), IsSpeedLimit $\implies$ IsSquare (speed limit signs are square). In summary, we have 3 PCs and 28 knowledge rules in

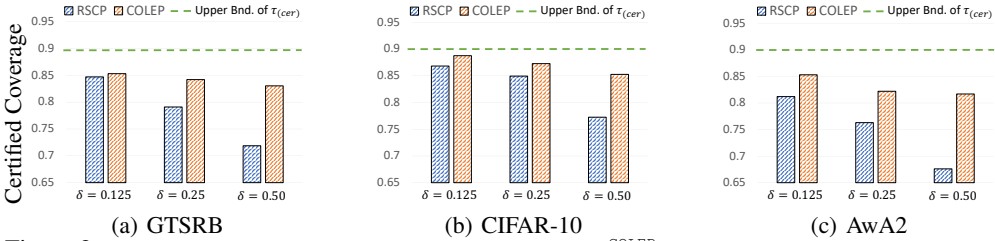

Figure 2: Comparison of certified coverage between COLEP ($\tau^{\text{COLEP}_{\text{cer}}}$) and RSCP under bounded perturbations $\delta = 0.125, 0.25, 0.50$ on GTSRB, CIFAR-10, and AwA2. The upper bound of certified coverage $\tau_{(\text{cer})}$ is 0.9.

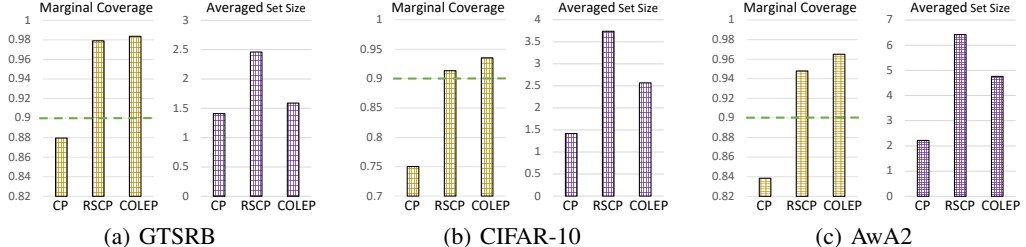

Figure 3: Comparison of the marginal coverage and averaged set size for CP, RSCP, and COLEP under PGD attack ($\delta = 0.25$) on GTSRB, CIFAR-10, and AwA2. The nominal coverage level (green line) is 0.9.

GTSRB, 3 PCs and 30 knowledge rules in CIFAR-10, and 4 PCs and 187 knowledge rules in AwA2. We also provide detailed steps of PC construction in appendix J.1.

**Certified Coverage under Bounded Perturbations.** We evaluate the certified coverage of COLEP given a new test sample $X_{n+1}$ with bounded perturbation $\epsilon$ ($|\epsilon|_2 < \delta$) based on our certification in Thm. 3. Certified coverage indicates the worst-case coverage during inference in the adversary setting, hence a higher certified coverage indicates a more robust conformal prediction and tigher certification. We compare the certified coverage of COLEP with the SOTA baseline RSCP (Theorem 2 in (Gendler et al., 2022)). The comparison of certified coverage between COLEP and RSCP is provided in Figure 2. The results indicate that COLEP consistently achieves higher certified coverage under different $\ell_2$ norm bounded perturbations, and COLEP outperforms RSCP by a large margin under a large perturbation radius $\delta = 0.5$. Note that an obvious upper bound of the certified coverage in the adversary setting is the guaranteed coverage without an adversary, which is 0.9 in the evaluation. The closeness between the certified coverage of COLEP $\tau_{(\text{cer})}^{\text{COLEP}}$ and the upper bound of coverage guarantee $\tau_{(\text{cer})}$ shows the robustness of COLEP and tightness of our certification.

**Prediction Coverage and Prediction Set Size under Adversarial Attacks.** We also evaluate the marginal coverage and averaged set size of COLEP and baselines under adversarial attacks. COLEP constructs the certifiably robust prediction set following Thm. 2. We compare the results with the standard conformal prediction (CP) and the SOTA conformal prediction with randomized smoothing (RSCP). For fair comparisons, we apply PGD attack (Madry et al., 2018) with the same parameters on CP, RSCP, and COLEP. For COLEP, we consider the adaptive PGD attack against the complete learning-reasoning pipeline. The comparison of the marginal coverage and averaged set size for CP, RSCP, and COLEP under PGD attack ($\delta = 0.25$) is provided in Figure 3. The results indicate that the marginal coverage of CP is below the nominal coverage level 0.9 under PGD attacks as data exchangeability is violated, while COLEP still achieves higher marginal coverage than the nominal level, validating the robustness of COLEP for conformal prediction in the adversary setting. Compared with RSCP, COLEP achieves both larger marginal coverage and smaller set size, demonstrating that COLEP maintains the guaranteed coverage with less inflation of the prediction set. The observation validates our theoretical analysis in Thm. 4 that COLEP can achieve better coverage than a single model with the power of kwowledge-enabled logical reasoning. We provide more evaluations of different coverage levels $1 - \alpha$, different perturbation bound $\delta$, different conformal prediction baselines, and contributions of different knowledge rules in Appendix J.2.

**Conclusion** In this paper, we present COLEP, a certifiably robust conformal prediction framework via knowledge-enabled logical reasoning. We leverage PCs for efficient reasoning and provide robustness certification. We also provide end-to-end certification for finite-sample certified coverage in the presence of adversaries and theoretically prove the advantage of COLEP over a single model. We included the discussions of limitations, broader impact, and future work in Appendix A.

ACKNOWLEDGMENTS

This work is partially supported by the National Science Foundation under grant No. 1910100, No. 2046726, No. 2229876, DARPA GARD, the National Aeronautics and Space Administration (NASA) under grant No. 80NSSC20M0229, the Alfred P. Sloan Fellowship, the Amazon research award, and the eBay research award.

ETHICS STATEMENT

We do not see potential ethical issues about `COLEP`. In contrast, with the power of logical reasoning, `COLEP` is a certifiably robust conformal prediction framework against adversaries during the inference time.

REPRODUCIBILITY STATEMENT

The reproducibility of `COLEP` span theoretical and experimental perspectives. We provide complete proofs of all the theoretical results in appendices. We provide the source codes for implementing `COLEP` at `https://github.com/kangmintong/COLEP`.

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

# Appendix

**Contents**

# A    DISCUSSIONS, BROADER IMPACT, LIMITATIONS, AND FUTURE WORK

**Broader Impact.** In this paper, we propose a certifiably robust conformal prediction framework COLEP via knowledge-enabled logic reasoning. We prove that for any adversarial test sample with $\ell_2$-norm bounded perturbations, the prediction set of COLEP captures the ground truth label with a guaranteed coverage level. Considering distribution shifts during inference time and weakening the assumption of data exchangeability has been an active area of conformal prediction. COLEP demonstrates both theoretical and empirical improvements in the adversarial setting and can inspire future work to explore various types of distribution shifts with the power of logical reasoning and the certification framework of COLEP. Furthermore, robustness certification for conformal prediction can be viewed as the generalization of certified prediction robustness, which justifies a certifiably robust prediction rather than a prediction set. Considering the challenges of providing tight bounds for certified prediction robustness, we expect that COLEP can motivate the certification of prediction robustness to benefit from a statistical perspective.

**Limitations.** A possible limitation may lie in the computational costs induced by pretraining knowledge models. The problem can be partially relieved by training knowledge models with parallelism. Moreover, we only need to pretrain them once and then can encode different types of knowledge rules based on domain knowledge. The training cost is worthy since we theoretically and empirically show that more knowledge models benefit the robustness of COLEP a lot for conformal prediction. Another possible limitation may lie in the access to knowledge rules. For the datasets that release hierarchical information (e.g., ImageNet, CIFAR-100, Visual Genome), we can easily adapt the information to implication rules used in COLEP. We can also seek knowledge graphs (Chen et al., 2020b) with related label semantics for rule design. These approaches may be effective in assisting practitioners in designing knowledge rules, but the process is not fully automatic, and human efforts are still needed. Overall, for any knowledge system, one naturally needs domain experts to design the knowledge rules specific to that application. There is probably no universal strategy on how to aggregate knowledge for any arbitrary application, and rather application-specific constructions are needed. This is where the power as well as limitation of our framework comes from.

**Future work.** The following directions are potentially interesting future work based on the framework of COLEP: 1) exploring different structures of knowledge representations and their effectiveness on conformal prediction performance, 2) proposing a more systematic way of designing logic rules based on domain knowledge, and 3) initiating a better understanding of the importance of different types of knowledge rules and motivating better design of rule weights accordingly.

**Discussions.** We will consider the cases where the knowledge graph is contaminated by adversarial attackers or contains noises and misinformation in this part. The robustness of COLEP mainly originates from the knowledge rules via factor function $F(\cdot)$ denoting how well each assignment conforms to the specified knowledge rules. Since the knowledge rules and weights are specified by practitioners and fixed during inferences, the PCs encoding the knowledge rules together with the main model and knowledge models (learning + reasoning component in Figure 1) can be viewed as a complete model. According to the adversarial attack settings in the literature Goodfellow et al. (2014); Gendler et al. (2022), the model weights can be well protected in practical cases and the attackers can only manipulate the inputs but not the model, and thus, the PC graph (i.e., the source of side information) which is a part of the augmented model can not be manipulated by adversarial attacks. On the other hand, it is an interesting question to discuss the case when the knowledge rules may be contaminated and contain misinformation due to the neglect of designers. We can view the problem of checking knowledge rules and correcting misinformation as a pre-processing step before the employment of COLEP. The parallel module of knowledge checking and correction is well-explored by a line of research Melo & Paulheim (2017); Caminhas (2019); Chen et al. (2020a), which basically detects and corrects false logical relations in the semantic embedding space via consistency-checking techniques. It is also interesting to leverage more advanced large language models to check and correct human-designed knowledge rules. We deem that the process of knowledge-checking is independent and parallel to COLEP, which may take additional efforts before the deployment of COLEP but significantly benefits in achieving a much more robust system based on our analysis and evaluations.

**Alternative certification**. There are two general certification directions for certifiably robust conformal prediction: (1) plugging in the standard score for for test example, and the quantile is computed

on the worst-case scores for each calibration point as in our paper; (2) plugging in the worst-case score for the test example and the standard score for each calibration point. We can also adapt the results with the second line based on the symmetry between $x$ and $x'$ is indeed an interesting point. Specifically, given perturbed $x'$ during test time, we can compute the bounds of output logits in the $\delta$-ball, which will also cover the clean $x$. Since the logit bounds also hold for clean $x$, we can compute the worst-case scores based on the bounds and finally construct the prediction set with the clean quantile value. The two lines of certification are essentially parallel. It is fundamentally due to the exchangeability of clean samples, which leads to consideration of worst-case bounds either during calibration or inference. One practical advantage of the first line of certification might be that the computation of worst-case bounds requires more runtime, and thus, computing it during the offline calibration process might be more desirable than during the online inference process.

**Discussions of selection of rule weights**. If a knowledge rule is true with a lower probability, then the associated weight rule should be smaller, and vice versa. For example, there might be a case that a knowledge rule only holds with probability 70%. In the example, the probability is with respect to the data distribution, which means that 70% of data satisfies the rule "IsStopSign→IsOctogan". Besides, other factors such as the portions of class labels also affect the importance of corresponding rules. If one class label is pretty rare in the data distribution, then the knowledge rules associated with it clearly should be assigned a small weight, and vice versa. In summary, the importance of knowledge rules depends on the data distribution, no matter whether we select it via grid search or optimize it in a more subtle way. Therefore, we deem that a general guideline for an effective and efficient selection of rule weights is to perform a grid search on a held-out validation set, which is also essential for standard model training. However, we do not think the effectiveness of COLEP is quite sensitive to the selection of weights. From the theoretical analysis and evaluations, we can expect benefits from the knowledge rules as long as we have a positive weight $w$ for the rules.

# B More related work

**Certified Prediction Robustness** To provide the worst-case robustness certification, a rich body of work has been proposed, including convex relaxations (Zhang et al., 2018; Li et al., 2019; Kang et al., 2023b), Lipschitz-bounded model layers (Tsuzuku et al., 2018; Singla et al., 2022; Xu et al., 2022b), and randomized smoothing (Cohen et al., 2019; Salman et al., 2019; Li et al., 2021; 2022a). However, existing work mainly focuses on certifying the *prediction robustness* rather than the *prediction coverage*, which is demonstrated to provide essential uncertainty quantification in conformal prediction (Jin et al., 2023). In this work, we aim to bridge worst-cast robustness certification and prediction coverage, and design learning frameworks to improve the uncertainty certification.

**Knowledged-Enabled Logical Reasoning** Neural-symbolic methods (Ahmed et al., 2022; Garcez et al., 2022; Yi et al., 2018) use symbols to represent objects, encode relationships with logical rules, and perform logical inference on them. DeepProbLog (Manhaeve et al., 2018) and Semantic loss (Xu et al., 2018) are representatives that leverage logic circuits Brown (2003) with weighted mode counting (WMC) to integrate symbolic knowledge. *Probabilistic circuits (PCs)* (Darwiche, 2002; 2003; Kisa et al., 2014) are another family of probabilistic models which allow exact and efficient inference. PCs permit various types of inference to be performed in linear time of the size of circuits (Hitzler & Sarker, 2022; Choi & Darwiche, 2017; Rooshenas & Lowd, 2014).

**Worst-case Certification**. In particular, considering potential adversarial manipulations during test time, where a small perturbation could mislead the models to make incorrect predictions (Szegedy et al., 2014; Madry et al., 2018; Xiao et al., 2018; Cao et al., 2021), different robustness certification approaches have been explored to provide the worst-case prediction guarantees for neural networks. For instance, a range of deterministic and statistical *worst-case certifications* have been proposed for a single model (Wong & Kolter, 2018; Zhang et al., 2018; Cohen et al., 2019; Lecuyer et al., 2019), reinforcement learning (Wu et al., 2022a; Kumar et al., 2022; Wu et al., 2022b), federated learning paradigms (Xie et al., 2021; 2022), and fair learning (Kang et al., 2022; Ray Chaudhury et al., 2022), under constrained perturbations (Balunovic et al., 2019; Li et al., 2021; 2022a).

## C PRELIMINARIES

### C.1 PROBABILISTIC CIRCUITS

**Definition 1** (Probabilistic Circuit). A probabilistic circuit (PC) $\mathcal{P}$ defines a joint distribution over the set of random variables $\mathbf{X}$ and can be defined as a tuple $(\mathcal{G}, \psi)$, where $\mathcal{G}$ represents an acyclic directed graph $(\mathcal{V}, \mathcal{E})$, and $\psi : \mathcal{V} \mapsto 2^{\mathbf{X}}$ the scope function that maps each node in $V$ to a subset of $\mathbf{X}$ (i.e., scope). This scope function maps the nodes in the graph to specific subsets of random variables $\mathbf{X}$. A leaf node $O_{\text{leaf}}$ of $\mathcal{G}$ computes a probability density over its scope $\psi(O_{\text{leaf}})$. All internal nodes of $\mathcal{G}$ are either sum nodes ($S$) or product nodes ($P$). A sum node $S$ computes a convex combination of its children: $S = \sum_{N \in \mathbf{ch}(S)} w_{S,N} N$ where $\mathbf{ch}(S)$ denotes the children of node $S$. A product node $P$ computes a product of its children: $P = \prod_{N \in \mathbf{ch}(P)} N$.

The output of PC is typically the value of the root node $O_{\text{root}}$. We mainly focus on the marginal probability computation inside PCs (e.g., $\mathbb{P}[x_j = 1] = \int_{x \in \mathbf{X}, x_j = 1} O_{\text{root}}(x) / \int_{x \in \mathbf{X}} O_{\text{root}}(x) dx$). The inference of marginal probability is tractable by imposing structural constraints of *smoothness* and *decomposability* : (1) *decomposability*: for any product node $P \in \mathcal{V}$, $\psi(v_1) \cap \psi(v_2) = \emptyset$, for $v_1, v_2 \in \mathbf{ch}(P), v_1 \neq v_2$, and 2) *smoothness*: for any sum node $S \in \mathcal{V}$, $\psi(v_1) = \psi(v_2)$, for $v_1, v_2 \in \mathbf{ch}(S)$. With the constraint of decomposability, the integrals on the product node $P$ can be factorized into integrals of children nodes $\mathbf{ch}(P)$, which can be formulated as follows:

$$
\int_{x \in \mathbf{X}} P(x) dx = \int_{x \in \mathbf{X}} \prod_{N \in \mathbf{ch}(P)} N(x) dx
$$
$$
= \prod_{N \in \mathbf{ch}(P)} \left[ \int_{x \in \mathbf{X}} N(x) dx \right]
$$
(22)

With the constraint of smoothness, the integrals on the sum node $S$ can be decomposed linearly to integrals on the children nodes $\mathbf{ch}(S)$, which can be formulated as:

$$
\int_{x \in \mathbf{X}} S(x) dx = \int_{x \in \mathbf{X}} \sum_{N \in \mathbf{ch}(S)} w_{S,N} N(x) dx
$$
$$
= \sum_{N \in \mathbf{ch}(S)} w_{S,N} \left[ \int_{x \in \mathbf{X}} N(x) dx \right]
$$
(23)

We can see that the smoothness and decomposability of PCs enable the integral computation on parent nodes to be pushed down to that on children nodes, and thus, the integral $\int_{x \in \mathbf{X}, x_j = 1} O_{\text{root}}(x)$ or $\int_{x \in \mathbf{X}} O_{\text{root}}(x)$ can be computed by a single forwarding pass of the graph $\mathcal{G}$. In total, we only need two forwarding passes in the graph $\mathcal{G}$ to compute the marginal probability $\mathbb{P}[x_j = 1] = \int_{x \in \mathbf{X}, x_j = 1} O_{\text{root}}(x) / \int_{x \in \mathbf{X}} O_{\text{root}}(x) dx$.

**Knowledge rules in PCs.** For the implication rule $A \implies B$, we define $A$ as conditional variables and $B$ as consequence variables. Suppose that the $r$-th PC encodes a set of implication rules defined over the set of Boolean variables $V_r$. The PC is called a homogeneous logic PC if there exists a bipartition $V_{rd}$ and $V_{rs}$ ($V_{rd} \cup V_{rs} = V$, $V_{rd} \cap V_{rs} = \emptyset$) such that all variables in $V_{rd}$ only appear as conditional variables and all variables in $V_{rs}$ only appears as consequence variables. We construct homogeneous logic PCs to follow the structural properties of PCs (i.e., decomposability and smoothness) and can show that such construction is intuitive in real-world applications. For example, in the road sign recognition task, we construct a PC with the knowledge of shapes. We encode a set of implication rules $A \implies B$ where $A$ is the road sign, and $B$ is the corresponding shape (e.g., a road sign can imply an octagon shape). We can verify that such a PC is a homogeneous PC and follow the structural constraints in the computation graph $\mathcal{G}$, as shown in Figure 1.

### C.2 MULTIPLE PCS WITH COEFFICIENTS $\beta_r$ IN THE COMBINATION OF PCS

Let us consider the case with $R$ PCs. The conditional class probability of the $j$-th class corrected by $r$-th PC can be formulated as:

$$\hat{\pi}_j^{(r)}(x) = \frac{\sum_{\mu \in M, \, \mu_j = 1} O_{\text{root}}^{(r)}(\mu)}{\sum_{\mu \in M} O_{\text{root}}^{(r)}(\mu)} = \frac{\sum_{\mu \in M, \, \mu_j = 1} \exp\left\{\sum_{j_\forall = 1}^{N_c + L} T(\hat{\pi}_{j_\forall}(x), \mu_{j_\forall})\right\} F_r(\mu)}{\sum_{\mu \in M} \exp\left\{\sum_{j_\forall = 1}^{N_c + L} T(\hat{\pi}_{j_\forall}(x), \mu_{j_\forall})\right\} F_r(\mu)} \quad (24)$$

where $\mu_j$ denotes the $j$-th element of vector $\mu$, and $T(a, b) = \log(ab + (1 - a)(1 - b))$, and $F_r(\mu) = \exp\left\{\sum_{h=1}^{H_r} w_{hr} \mathbb{I}[\mu \sim K_{hr}]\right\}$. $H_r, K_{hr}, w_{hr}$ are the number of knowledge rules, the $h$-th logic rule, and the weight of the $h$-th logic rule for the $r$-th PC. Note that the numerator and denominator of eq. (24) can be exactly and efficiently computed with a single forwarding pass of PCs (Hitzler & Sarker, 2022; Choi & Darwiche, 2017; Rooshenas & Lowd, 2014). Specifically, the inference time complexity of one PC defined on graph $(\mathcal{V}, \mathcal{E})$ with node set $\mathcal{V}$ and edge set $\mathcal{E}$ is $\mathcal{O}(|\mathcal{V}|)$.

We improve the expressiveness of the reasoning component by combining $R$ PCs with coefficients $\beta_r$ for the $r$-th PC. Formally, given data input $x$, the corrected conditional class probability $\hat{\pi}_j^{\text{COLEP}}$ can be expressed as:

$$\hat{\pi}_j^{\text{COLEP}}(x) = \sum_{r \in [R]} \beta_r \hat{\pi}_j^{(r)}(x) \quad (25)$$

We also note that the coefficients $\beta_r$ correspond to the accuracy of $r$-th PC normalized over all PC accuracies. The core of this formulation is the mixture model involving a latent variable $r_{\text{PC}}$ representing the PCs. In short, we can write $\hat{\pi}_j^{(r)}(x)$ as $\hat{\pi}_j^{(r)}(x) = \mathbb{P}[Y = y | X = x, r_{\text{PC}} = r]$, and $\hat{\pi}_j^{\text{COLEP}}(x)$ as the marginalized probability over the latent variable $r_{\text{PC}}$ such as:

$$\hat{\pi}_j^{\text{COLEP}}(x) = \mathbb{P}[Y = y | X = x] = \sum_{r \in [R]} \mathbb{P}[r_{\text{PC}} = r] \cdot \mathbb{P}[Y = y | X = x, r_{\text{PC}} = r]$$
$$= \sum_{r \in [R]} \mathbb{P}[r_{\text{PC}} = r] \cdot \hat{\pi}_j^{(r)}(x). \quad (26)$$

Hence, the coefficient $\beta_r$ for the $r$-th PC are determined by $\mathbb{P}[r_{\text{PC}} = r]$. Although we lack direct knowledge of this probability, we can estimate it using the data by examining how frequently each PC $r$ correctly predicts the outcome across the given examples, similarly as in the estimation of prior class probabilities for Naive Bayes classifiers.

### C.3 Randomized Smoothing

**Lemma C.1** (Randomized Smoothing (Cohen et al., 2019)). *Considering the class conditional probability $\hat{\pi}_j : \mathbb{R}^d \mapsto [0, 1]$, we construct a smoothed model $g_j(x; \sigma) = \mathbb{E}_{\eta \sim \mathcal{N}(0, \sigma^2)}[\hat{\pi}_j(x + \eta)]$. With input perturbation $\|\epsilon\|_2 \leq \delta$, the smoothed model satisfies:*

$$\Phi(\Phi^{-1}(g_j(x; \sigma)) - \delta/\sigma) \leq g_j(x + \epsilon; \sigma) \leq \Phi(\Phi^{-1}(g_j(x; \sigma)) + \delta/\sigma), \quad (27)$$

*where $\Phi$ is the Gaussian cumulative density function (CDF) and $\Phi^{-1}$ is its inverse.*

In practice, we can estimate the expectation in $g_j(x; \sigma)$ by taking the average over many i.i.d. realizations of Gaussian noises following (Cohen et al., 2019). Thus, the output probability of the smoothed model can be bounded given input perturbations. Note that the specific ways of certifying the learning component are orthogonal to certifying the reasoning component and the conformal prediction procedure, and one can plug in different certification strategies for the learning component.

## D Conformal Prediction with COLEP

### D.1 Flexibility of user-defined coverage level

In this part, we illustrate the flexibility of COLEP to use different miscoverage levels $\alpha_j$ for different label classes $j \in [N_c]$ and prove the guaranteed coverage. Recall that in the standard conformal prediction setting, we are given a desired coverage $1 - \alpha$ and construct the prediction set of a new test sample $\hat{C}_{n,\alpha}(X_{n+1})$ with the guarantee of marginal coverage: $\mathbb{P}[Y_{n+1} \subseteq \hat{C}_{n,\alpha}(X_{n+1})] \geq 1 - \alpha$. In COLEP, we construct the prediction set $\hat{C}_{n,\alpha_j}^{\text{COLEP}_j}(X_{n+1})$ for the $j$-th class ($j \in [N_c]$) as in Equation (7)

and construct the final prediction set $\hat{C}_{n,\alpha}^{\text{COLEP}}(X_{n+1})$ as in Equation (8). Then we have the following:

$$\mathbb{P}\left[Y_{n+1} \notin \hat{C}_{n,\alpha}^{\text{COLEP}}(X_{n+1})\right] \tag{28}$$

$$=\mathbb{P}\left[1 \notin \hat{C}_{n,\alpha_j}^{\text{COLEP}^{Y_{n+1}}}(X_{n+1})\right] \tag{29}$$

$$=\mathbb{P}\left[S_{\hat{\pi}_{Y_{n+1}}^{\text{COLEP}}}(X_{n+1},1) > Q_{1-\alpha_{Y_{n+1}}}\left(S_{\hat{\pi}_{Y_{n+1}}^{\text{COLEP}}}\left(\{X_i, \mathbb{I}_{[Y_i=Y_{n+1}]}\}_{i\in\mathcal{I}_{\text{cal}}}\right)\right)\right] \tag{30}$$

$$\leq \max_{j\in[N_c]} \mathbb{P}\left[S_{\hat{\pi}_j^{\text{COLEP}}}(X_{n+1}, \mathbb{I}_{[Y_{n+1}=j]}) > Q_{1-\alpha_j}\left(S_{\hat{\pi}_j^{\text{COLEP}}}\left(\{X_i, \mathbb{I}_{[Y_i=j]}\}_{i\in\mathcal{I}_{\text{cal}}}\right)\right)\right] \tag{31}$$

$$= \max_{j\in[N_c]} \alpha_j \tag{32}$$

Therefore, we conclude that as long as we set $\max_{j\in[N_c]}\{\alpha_j\} \leq \alpha$, we have the guarantee of marginal coverage: $\mathbb{P}\left[Y_{n+1} \in \hat{C}_{n,\alpha}^{\text{COLEP}}(X_{n+1})\right] \geq 1-\alpha$. The flexibility of COLEP to set various coverage levels for different label classes is advantageous in addressing problems related to the imbalance of classes. For example, when samples of class $j$ are extremely vulnerable to be attacked by the adversary, we can set a higher miscoverage level $\alpha_j$ for class $j$ to consider the class-specific vulnerability instead of setting a higher $\alpha$ for all classes which may affect the performance on other classes.

### D.2 ILLUSTRATIVE EXAMPLE

Let us consider the example of classifying whether the given image has a stop sign in Figure 1. The main model generates an estimate $\hat{\pi}(x)$, meaning that the image $x$ has a stop sign with probability $\hat{\pi}(x)$. Suppose that we have a knowledge model to detect whether the image has an object of octagon shape with an estimate $\hat{\pi}_o(x)$, meaning that the image $x$ has an object of octagon shape with probability $\hat{\pi}_o(x)$. Then we can encode the logic rule "IsStopSign $\implies$ IsOctagon" with weight $w$ in the PC, as shown in Figure 1. The PC defines a joint distribution over the Bernoulli random variables IsStopSign and IsStopSign and computes the likelihood of an instantiation of the random variables at its output (e.g., $p(\text{IsStopSign} = 1, \text{IsOctagon} = 0)$). The represented joint distribution in the PC is determined by the encoded logic rule with the pattern in the top PC of Figure 1. We can see that only the assignment (IsStopSign $= 1$, IsOctagon $= 0$) is not multiplied by the weight $w$ due to the contradiction of the rule. Therefore, the probability of instantiations of random variables with such contradictions will be down-weighted after the correction of the PC. In the simple PC, we formally compute the class probability as:

$$
\begin{aligned}
p(\text{IsStopSign} = 1) = &\{p(\text{IsStopSign} = 1, \text{IsOctagon} = 0) + p(\text{IsStopSign} = 1, \text{IsOctagon} = 1)\}/ \\
&\{p(\text{IsStopSign} = 1, p(\text{IsOctagon} = 0) + p(\text{IsStopSign} = 1, \text{IsOctagon} = 1) \\
&+ p(\text{IsStopSign} = 0, \text{IsOctagon} = 0) + p(\text{IsStopSign} = 0, \text{IsOctagon} = 1)\}.
\end{aligned}
\tag{33}
$$

Note that the nominator and denominator can be tractably computed by a forwarding pass of the PC graph. COLEP can enlarge the probability estimate $p(\text{IsStopSign} = 1)$ when $x$ is a stop sign and reduce it when $x$ is not a stop sign by leveraging additional concept information $p(\text{IsStopSign}$, as rigorously analyzed in Lemma 5.1.

## E OVERVIEW OF CERTIFICATION

We provide the overview of certification flow in Figure 4, for a better understanding the connections and differences of certifications in COLEP.

## F OMITTED PROOFS IN SECTION 4

### F.1 PROOF OF THM. 1

**Theorem 1** (Generalization of Thm. 1 with multiple PCs). *Given any input $x$ and perturbation bound $\delta$, we let $[\underline{\hat{\pi}_{j_\forall}}(x), \overline{\hat{\pi}_{j_\forall}}(x)]$ be bounds for the estimated conditional class and concept probabilities by*

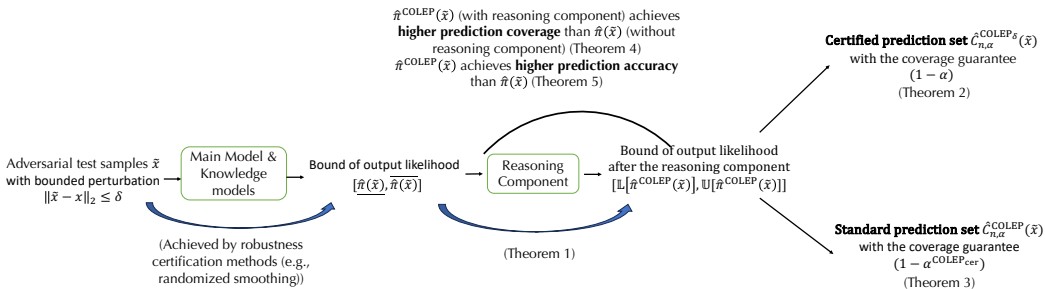

Figure 4: Overview of prediction certification of COLEP. The certification setting is that the inference time adversaries can violate the data exchangeability assumption, compromising the guaranteed coverage. Therefore, the certification generally achieves the following three goals. (1) We can preserve the guaranteed coverage using a prediction set that takes the perturbation bound into account (Thm. 2), achieved by computing the probability bound of models before the reasoning component (by randomized smoothing) and the bound after the reasoning component (by Thm. 1). (2) We prove the worst-case coverage (a lower bound) if we use the standard prediction set as before (Thm. 3). (3) We theoretically show that COLEP can achieve a better prediction coverage (Thm. 4) and prediction accuracy (Thm. 5) than a data-driven model without the reasoning component.

the models with $j_\forall \in [N_c + L]$. Let $V_{rd}^j$ be the set of index of conditional variables in the $r$-th PC except for $j \in [N_c]$ and $V_{rs}^j$ be that of consequence variables. Then the bound for COLEP-corrected estimate of the conditional class probability $\hat{\pi}_j^{COLEP}$ is given by:

$$
\mathbb{U}[\hat{\pi}_j^{COLEP}(x)] = \sum_{r \in [R]} \beta_r \left\{ \frac{(1 - \overline{\hat{\pi}_j}(x)) \sum_{\mu_j = 0} \exp\left\{ \sum_{j_\forall \in V_{rd}^j} T(\overline{\hat{\pi}_{j_\forall}}(x), \mu_{j_\forall}) + \sum_{j_\forall \in V_{rs}^j} T(\underline{\hat{\pi}_{j_\forall}}(x), \mu_{j_\forall}) \right\} F_r(\mu)}{\overline{\hat{\pi}_j}(x) \sum_{\mu_j = 1} \exp\left\{ \sum_{j_\forall \in V_{rd}^j} T(\underline{\hat{\pi}_{j_\forall}}(x), \mu_{j_\forall}) + \sum_{j_\forall \in V_{rs}^j} T(\overline{\hat{\pi}_{j_\forall}}(x), \mu_{j_\forall}) \right\} F_r(\mu)} + 1 \right\}^{-1}
$$

$$
\mathbb{L}[\hat{\pi}_j^{COLEP}(x)] = \sum_{r \in [R]} \beta_r \left\{ \frac{(1 - \underline{\hat{\pi}_j}(x)) \sum_{\mu_j = 0} \exp\left\{ \sum_{j_\forall \in V_{rd}^j} T(\underline{\hat{\pi}_{j_\forall}}(x), \mu_{j_\forall}) + \sum_{j_\forall \in V_{rs}^j} T(\overline{\hat{\pi}_{j_\forall}}(x), \mu_{j_\forall}) \right\} F_r(\mu)}{\underline{\hat{\pi}_j}(x) \sum_{\mu_j = 1} \exp\left\{ \sum_{j_\forall \in V_{rd}^j} T(\overline{\hat{\pi}_{j_\forall}}(x), \mu_{j_\forall}) + \sum_{j_\forall \in V_{rs}^j} T(\underline{\hat{\pi}_{j_\forall}}(x), \mu_{j_\forall}) \right\} F_r(\mu)} + 1 \right\}^{-1}
$$

(34)

where $T(a, b) = \log(ab + (1 - a)(1 - b))$.

*Proof of Thm. 1.* We first provide a lemma that proves the monotonicity of the complex function with respect to the estimated condition probabilities of conditional variables and consequence variables within PCs as follows.

**Lemma F.1** (Function Monotonicity within PCs). *We define the target function as $G(q, d) = \sum_{\mu \in M_{jd}} \exp\left\{ \sum_{j_\forall \in [N_c + L] \setminus \{j\}} T(\hat{\pi}_{j_\forall}(x), \mu_{j_\forall}) \right\} F_r(\mu)$, where $q \in [N_c + L] \setminus \{j\}$ and $M_{jd} = \{\mu \in M : \mu_j = d\}$ ($d \in \{0, 1\}$). Then we have $\partial G / \partial \hat{\pi}_q(x) \leq 0$ if class label $q$ corresponds to a conditional variable in the PC and $\partial G / \partial \hat{\pi}_q(x) \geq 0$ if class label $q$ corresponds to a consequence variable in the PC.*

*Proof of Lemma F.1.* Note that the function $G(q, d)$ is linear with respect to $\hat{\pi}_q(x)$. To explore the monotonicity, we only need to determine the sign of the coefficients of $\hat{\pi}_q(x)$. We first consider the case where class label $q$ corresponds to a conditional variable. We can rearrange the expression of $G(q, d)$ as follows:

$$G(q,d) = \sum_{\mu \in M_{jd}} \exp\left\{ \sum_{j_\forall \in [N_c+L]\backslash\{j\}} T(\hat{\pi}_{j_\forall}(x), \mu_{j_\forall}) \right\} F_r(\mu)$$

$$= \sum_{\mu \in M_{jd}, \mu_q=1} \exp\left\{ \sum_{j_\forall \in [N_c+L]\backslash\{j\}} T(\hat{\pi}_{j_\forall}(x), \mu_{j_\forall}) \right\} F_r(\mu) + \sum_{\mu \in M_{jd}, \mu_q=0} \exp\left\{ \sum_{j_\forall \in [N_c+L]\backslash\{j\}} T(\hat{\pi}_{j_\forall}(x), \mu_{j_\forall}) \right\} F_r(\mu)$$

$$= \hat{\pi}_q(x) \sum_{\mu \in M_{jd}, \mu_q=1} \exp\left\{ \sum_{j_\forall \in [N_c+L]\backslash\{j,q\}} T(\hat{\pi}_{j_\forall}(x), \mu_{j_\forall}) \right\} F_r(\mu) + (1-\hat{\pi}_q(x)) \sum_{\mu \in M_{jd}, \mu_q=0} \exp\left\{ \sum_{j_\forall \in [N_c+L]\backslash\{j,q\}} T(\hat{\pi}_{j_\forall}(x), \mu_{j_\forall}) \right\} F_r(\mu)$$

$$= \hat{\pi}_q(x) \left[ \sum_{\mu \in M_{jd}, \mu_q=1} \exp\left\{ \sum_{j_\forall \in [N_c+L]\backslash\{j,q\}} T(\hat{\pi}_{j_\forall}(x), \mu_{j_\forall}) \right\} F_r(\mu) - \sum_{\mu \in M_{jd}, \mu_q=0} \exp\left\{ \sum_{j_\forall \in [N_c+L]\backslash\{j,q\}} T(\hat{\pi}_{j_\forall}(x), \mu_{j_\forall}) \right\} F_r(\mu) \right]$$

$$+ \left[ \sum_{\mu \in M_{jd}, \mu_q=0} \exp\left\{ \sum_{j_\forall \in [N_c+L]\backslash\{j,q\}} T(\hat{\pi}_{j_\forall}(x), \mu_{j_\forall}) \right\} F_r(\mu) \right]$$

(35)

Then we rearrange the coefficients of $\hat{\pi}_q(x)$ as follows:

$$\sum_{\mu \in M_{jd}, \mu_q=1} \exp\left\{ \sum_{j_\forall \in [N_c+L]\backslash\{j,q\}} T(\hat{\pi}_{j_\forall}(x), \mu_{j_\forall}) \right\} F_r(\mu) - \sum_{\mu \in M_{jd}, \mu_q=0} \exp\left\{ \sum_{j_\forall \in [N_c+L]\backslash\{j,q\}} T(\hat{\pi}_{j_\forall}(x), \mu_{j_\forall}) \right\} F_r(\mu)$$

$$= \sum_{\mu \in M_{jd}, \mu_q=0} \exp\left\{ \sum_{j_\forall \in [N_c+L]\backslash\{j,q\}} T(\hat{\pi}_{j_\forall}(x), \mu_{j_\forall}) \right\} \left[ F_r(R(\mu,q)) - F_r(\mu) \right],$$

(36)

where $R(\mu,q) : \{0,1\}^{N_c+L} \times \mathbb{Z}^+ \mapsto \{0,1\}^{N_c+L}$ rewrites the $q$-th dimension of vector $\mu$ to 1. Note that when $q$ is the index of a conditional variable, given any implication logic rule $K_{hr}$, we have $\mathbb{I}[R(\mu,q) \sim K_{hr}] \leq \mathbb{I}[\mu \sim K_{hr}]$. Therefore, $F_r(R(\mu,q)) - F_r(\mu) \leq 0$ holds and $\partial G/\partial \hat{\pi}_q(x) \leq 0$.

Similarly, when $q$ is the index of a consequence variable, given any implication logic rule $K_{hr}$, we have $\mathbb{I}[R(\mu,q) \sim K_{hr}] \geq \mathbb{I}[\mu \sim K_{hr}]$. Therefore, $F_r(R(\mu,q)) - F_r(\mu) \geq 0$ holds and $\partial G/\partial \hat{\pi}_q(x) \geq 0$.

$\square$

Consider the $r$-th PC. We first rearrange the formulation of the class conditional probability $\hat{\pi}_j^{(r)}(x)$ to the pattern of function $G(\cdot)$ and then leverage Lemma F.1 to derive the upper and lower bound. The reformulation of $\hat{\pi}_j^{(r)}(x)$ is shown as the following:

$$\hat{\pi}_j^{(r)}(x) = \frac{\sum_{\mu \in M, \, \mu_j=1} \exp\left\{ \sum_{j_\forall=1}^{N_c+L} T(\hat{\pi}_{j_\forall}(x), \mu_{j_\forall}) \right\} F_r(\mu)}{\sum_{\mu \in M} \exp\left\{ \sum_{j_\forall=1}^{N_c+L} T(\hat{\pi}_{j_\forall}(x), \mu_{j_\forall}) \right\} F_r(\mu)}$$

(37)

$$= \frac{1}{1 + \dfrac{\sum_{\mu \in M, \, \mu_j=0} \exp\left\{ \sum_{j_\forall=1}^{N_c+L} T(\hat{\pi}_{j_\forall}(x), \mu_{j_\forall}) \right\} F_r(\mu)}{\sum_{\mu \in M, \, \mu_j=1} \exp\left\{ \sum_{j_\forall=1}^{N_c+L} T(\hat{\pi}_{j_\forall}(x), \mu_{j_\forall}) \right\} F_r(\mu)}}$$

(38)

$$= \frac{1}{1 + \dfrac{(1-\hat{\pi}_j(x)) \sum_{\mu \in M, \, \mu_j=0} \exp\left\{ \sum_{j_\forall \in [N_c+L]\backslash\{j\}} T(\hat{\pi}_{j_\forall}(x), \mu_{j_\forall}) \right\} F_r(\mu)}{\hat{\pi}_j(x) \sum_{\mu \in M, \, \mu_j=1} \exp\left\{ \sum_{j_\forall \in [N_c+L]\backslash\{j\}} T(\hat{\pi}_{j_\forall}(x), \mu_{j_\forall}) \right\} F_r(\mu)}}$$

(39)

Next, we focus on deriving the lower bound $\mathbb{L}[\hat{\pi}_j^{(r)}(x)]$, and the upper bound can be derived similarly.

$$\hat{\pi}_j^{(r)}(x) \geq \left\{ 1 + \frac{(1 - \hat{\pi}_j(x)) \sum_{\mu \in M, \ \mu_j=0} \exp \left\{ \sum_{j_\forall \in [N_c+L] \setminus \{j\}} T(\hat{\pi}_{j_\forall}(x), \mu_{j_\forall}) \right\} F_r(\mu)}{\underline{\hat{\pi}_j}(x) \sum_{\mu \in M, \ \mu_j=1} \exp \left\{ \sum_{j_\forall \in [N_c+L] \setminus \{j\}} T(\hat{\pi}_{j_\forall}(x), \mu_{j_\forall}) \right\} F_r(\mu)} \right\}^{-1}$$

(40)

$$\geq \left\{ 1 + \frac{(1 - \underline{\hat{\pi}_j}(x)) \cdot \mathbb{U} \left[ \sum_{\mu \in M, \ \mu_j=0} \exp \left\{ \sum_{j_\forall \in [N_c+L] \setminus \{j\}} T(\hat{\pi}_{j_\forall}(x), \mu_{j_\forall}) \right\} F_r(\mu) \right]}{\underline{\hat{\pi}_j}(x) \cdot \mathbb{L} \left[ \sum_{\mu \in M, \ \mu_j=1} \exp \left\{ \sum_{j_\forall \in [N_c+L] \setminus \{j\}} T(\hat{\pi}_{j_\forall}(x), \mu_{j_\forall}) \right\} F_r(\mu) \right]} \right\}^{-1}$$

(41)

In Equation (41), we relax the numerator and denominator separately and then will show how to derive the upper bound of the numerator and the lower bound of the denominator in Equation (41). We can apply Lemma F.1 for any $q \in [N_c + L] \setminus \{j\}$ and get the following:

$$\sum_{\mu \in M, \ \mu_j=0} \exp \left\{ \sum_{j_\forall \in [N_c+L] \setminus \{j\}} T(\hat{\pi}_{j_\forall}(x), \mu_{j_\forall}) \right\} F_r(\mu) \leq \sum_{j_\forall \in V_{rd}^j} T(\overline{\hat{\pi}_{j_\forall}}(x), \mu_{j_\forall}) + \sum_{j_\forall \in V_{rs}^j} T(\underline{\hat{\pi}_{j_\forall}}(x), \mu_{j_\forall})$$

$$\sum_{\mu \in M, \ \mu_j=0} \exp \left\{ \sum_{j_\forall \in [N_c+L] \setminus \{j\}} T(\hat{\pi}_{j_\forall}(x), \mu_{j_\forall}) \right\} F_r(\mu) \geq \sum_{j_\forall \in V_{rd}^j} T(\underline{\hat{\pi}_{j_\forall}}(x), \mu_{j_\forall}) + \sum_{j_\forall \in V_{rs}^j} T(\overline{\hat{\pi}_{j_\forall}}(x), \mu_{j_\forall})$$

(42)

where $V_{rd}^j$ is the set of index of conditional variables in the $r$-th PC except for $j \in [N_c]$ and $V_{rs}^j$ is that of consequence variables.

From Equation (41) and Equation (42), we can derive the final bound of $\hat{\pi}_j^{\text{COLEP}}(x)$ by considering all the PCs:

$$\mathbb{U}[\hat{\pi}_j^{\text{COLEP}}(x)] = \sum_{r \in [R]} \beta_r \left\{ \frac{(1 - \overline{\hat{\pi}_j}(x)) \sum_{\mu_j=0} \exp \left\{ \sum_{j_\forall \in V_{rd}^j} T(\overline{\hat{\pi}_{j_\forall}}(x), \mu_{j_\forall}) + \sum_{j_\forall \in V_{rs}^j} T(\underline{\hat{\pi}_{j_\forall}}(x), \mu_{j_\forall}) \right\} F_r(\mu)}{\overline{\hat{\pi}_j}(x) \sum_{\mu_j=1} \exp \left\{ \sum_{j_\forall \in V_{rd}^j} T(\underline{\hat{\pi}_{j_\forall}}(x), \mu_{j_\forall}) + \sum_{j_\forall \in V_{rs}^j} T(\overline{\hat{\pi}_{j_\forall}}(x), \mu_{j_\forall}) \right\} F_r(\mu)} + 1 \right\}^{-1}$$

$$\mathbb{L}[\hat{\pi}_j^{\text{COLEP}}(x)] = \sum_{r \in [R]} \beta_r \left\{ \frac{(1 - \underline{\hat{\pi}_j}(x)) \sum_{\mu_j=0} \exp \left\{ \sum_{j_\forall \in V_{rd}^j} T(\underline{\hat{\pi}_{j_\forall}}(x), \mu_{j_\forall}) + \sum_{j_\forall \in V_{rs}^j} T(\overline{\hat{\pi}_{j_\forall}}(x), \mu_{j_\forall}) \right\} F_r(\mu)}{\underline{\hat{\pi}_j}(x) \sum_{\mu_j=1} \exp \left\{ \sum_{j_\forall \in V_{rd}^j} T(\overline{\hat{\pi}_{j_\forall}}(x), \mu_{j_\forall}) + \sum_{j_\forall \in V_{rs}^j} T(\underline{\hat{\pi}_{j_\forall}}(x), \mu_{j_\forall}) \right\} F_r(\mu)} + 1 \right\}^{-1}$$

(43)

$\square$

## F.2 PROOF OF THM. 2

**Theorem 2** (Recall). Consider a new test sample $X_{n+1}$ drawn from $P_{XY}$. For any bounded perturbation $\|\epsilon\|_2 \leq \delta$ in the input space and the adversarial sample $\tilde{X}_{n+1} := X_{n+1} + \epsilon$, we have the following guaranteed marginal coverage:

$$\mathbb{P}[Y_{n+1} \in \hat{C}_{n,\alpha}^{\text{COLEP}\delta}(\tilde{X}_{n+1})] \geq 1 - \alpha$$

(44)

if we construct the certified prediction set of COLEP where

$$\hat{C}_{n,\alpha}^{\text{COLEP}\delta}(\tilde{X}_{n+1}) = \left\{ j \in [N_c] : S_{\hat{\pi}_j^{\text{COLEP}}}(\tilde{X}_{n+1}, 1) \leq Q_{1-\alpha}(\{S_{\hat{\pi}_j^{\text{COLEP}\delta}}(X_i, \mathbb{I}_{[Y_i=j]})\}_{i \in \mathcal{I}_{\text{cal}}}) \right\}$$

(45)

and $S_{\hat{\pi}_j^{\text{COLEP}\delta}}(\cdot, \cdot)$ is a function of worst-case non-conformity score considering perturbation radius $\delta$:

$$S_{\hat{\pi}_j^{\text{COLEP}\delta}}(X_i, \mathbb{I}_{[Y_i=j]}) = \begin{cases} \mathbb{U}_\delta[\hat{\pi}_j^{\text{COLEP}}(X_i)] + u(1 - \mathbb{U}_\delta[\hat{\pi}_j^{\text{COLEP}}(X_i)]), & Y_i \neq j \\ 1 - \mathbb{L}_\delta[\hat{\pi}_j^{\text{COLEP}}(X_i)] + u\mathbb{L}_\delta[\hat{\pi}_j^{\text{COLEP}}(X_i)], & Y_i = j \end{cases} \tag{46}$$

with $\mathbb{U}_\delta[\hat{\pi}_j^{\text{COLEP}}(x)] = \max_{|\eta|_2 \leq \delta} \hat{\pi}_j^{\text{COLEP}}(x+\eta)$ and $\mathbb{L}_\delta[\hat{\pi}_j^{\text{COLEP}}(x)] = \min_{|\eta|_2 \leq \delta} \hat{\pi}_j^{\text{COLEP}}(x+\eta)$.

*Proof of Thm. 2.* We can upper bound the probability of miscoverage of the main task with the probability of miscoverage of label classes as follows:

$$\mathbb{P}\left[Y_{n+1} \notin \hat{C}_{n,\alpha}^{\text{COLEP}\delta}(\tilde{X}_{n+1})\right] \tag{47}$$

$$\leq \max_{j \in [N_c]} \mathbb{P}\left[\mathbb{I}_{[Y_{n+1}=j]} \notin \hat{C}_{n,\alpha,j}^{\text{COLEP}\delta}(\tilde{X}_{n+1})\right] \tag{48}$$

$$\leq \max_{j \in [N_c]} \mathbb{P}\left[S_{\hat{\pi}_j^{\text{COLEP}}}(\tilde{X}_{n+1}, \mathbb{I}_{[Y_{n+1}=j]}) > Q_{1-\alpha}(S_{\hat{\pi}_j^{\text{COLEP}\delta}}(\{X_i, \mathbb{I}_{[Y_i=j]}\}_{i \in \mathcal{I}_{\text{cal}}})\right] \tag{49}$$

$$\leq \max_{j \in [N_c]} \mathbb{P}\left[S_{\hat{\pi}_j^{\text{COLEP}\delta}}(X_{n+1}, \mathbb{I}_{[Y_{n+1}=j]}) > Q_{1-\alpha}(S_{\hat{\pi}_j^{\text{COLEP}\delta}}(\{X_i, \mathbb{I}_{[Y_i=j]}\}_{i \in \mathcal{I}_{\text{cal}}})\right] \tag{50}$$

$$\leq \alpha \tag{51}$$

where $\hat{C}_{n,\alpha,j}^{\text{COLEP}\delta}(\tilde{X}_{n+1}) := \left\{q^y \in \{0,1\} : S_{\hat{\pi}_j^{\text{COLEP}}}(\tilde{X}_{n+1}, y) \leq Q_{1-\alpha}(S_{\hat{\pi}_j^{\text{COLEP}\delta}}(\{X_i, \mathbb{I}_{[Y_i=j]}\}_{i \in \mathcal{I}_{\text{cal}}})\right\}$ is the certified prediction set for the class label $j$.

Note that Equation (50) holds because we have $S_{\hat{\pi}_j^{\text{COLEP}\delta}}(X_{n+1}, \mathbb{I}_{[Y_{n+1}=j]}) \geq S_{\hat{\pi}_j^{\text{COLEP}}}(\tilde{X}_{n+1}, \mathbb{I}_{[Y_{n+1}=j]})$. Next, we will prove Equation (50) rigorously. We first consider the probability of miscoverage conditioned on $Y_{n+1} = j$.

$$\mathbb{P}\left[S_{\hat{\pi}_j^{\text{COLEP}}}(\tilde{X}_{n+1}, \mathbb{I}_{[Y_{n+1}=j]}) > Q_{1-\alpha}(S_{\hat{\pi}_j^{\text{COLEP}\delta}}(\{X_i, \mathbb{I}_{[Y_i=j]}\}_{i \in \mathcal{I}_{\text{cal}}}) | Y_{n+1} = j\right] \tag{52}$$

$$\leq \mathbb{P}\left[1 - (1-u)\hat{\pi}_j^{\text{COLEP}}(\tilde{X}_{n+1}) > Q_{1-\alpha}(S_{\hat{\pi}_j^{\text{COLEP}\delta}}(\{X_i, \mathbb{I}_{[Y_i=j]}\}_{i \in \mathcal{I}_{\text{cal}}}) | Y_{n+1} = j\right] \tag{53}$$

$$\leq \mathbb{P}\left[1 - (1-u) \cdot \mathbb{L}\left[\hat{\pi}_j^{\text{COLEP}}(\tilde{X}_{n+1})\right] > Q_{1-\alpha}(S_{\hat{\pi}_j^{\text{COLEP}\delta}}(\{X_i, \mathbb{I}_{[Y_i=j]}\}_{i \in \mathcal{I}_{\text{cal}}}) | Y_{n+1} = j\right] \tag{54}$$

$$\leq \mathbb{P}\left[S_{\hat{\pi}_j^{\text{COLEP}\delta}}(X_{n+1}, \mathbb{I}_{[Y_{n+1}=j]}) > Q_{1-\alpha}(S_{\hat{\pi}_j^{\text{COLEP}\delta}}(\{X_i, \mathbb{I}_{[Y_i=j]}\}_{i \in \mathcal{I}_{\text{cal}}}) | Y_{n+1} = j\right] \tag{55}$$

Similarly, we consider the probability of miscoverage conditioned on $Y_{n+1} \neq j$.

$$\mathbb{P}\left[S_{\hat{\pi}_j^{\text{COLEP}}}(\tilde{X}_{n+1}, \mathbb{I}_{[Y_{n+1}=j]}) > Q_{1-\alpha}(S_{\hat{\pi}_j^{\text{COLEP}\delta}}(\{X_i, \mathbb{I}_{[Y_i=j]}\}_{i \in \mathcal{I}_{\text{cal}}}) | Y_{n+1} \neq j\right] \tag{56}$$

$$\leq \mathbb{P}\left[u + (1-u)\hat{\pi}_j^{\text{COLEP}}(\tilde{X}_{n+1}) > Q_{1-\alpha}(S_{\hat{\pi}_j^{\text{COLEP}\delta}}(\{X_i, \mathbb{I}_{[Y_i=j]}\}_{i \in \mathcal{I}_{\text{cal}}}) | Y_{n+1} \neq j\right] \tag{57}$$

$$\leq \mathbb{P}\left[u + (1-u) \cdot \mathbb{U}\left[\hat{\pi}_j^{\text{COLEP}}(\tilde{X}_{n+1})\right] > Q_{1-\alpha}(S_{\hat{\pi}_j^{\text{COLEP}\delta}}(\{X_i, \mathbb{I}_{[Y_i=j]}\}_{i \in \mathcal{I}_{\text{cal}}}) | Y_{n+1} \neq j\right] \tag{58}$$

$$\leq \mathbb{P}\left[S_{\hat{\pi}_j^{\text{COLEP}\delta}}(X_{n+1}, \mathbb{I}_{[Y_{n+1}=j]}) > Q_{1-\alpha}(S_{\hat{\pi}_j^{\text{COLEP}\delta}}(\{X_i, \mathbb{I}_{[Y_i=j]}\}_{i \in \mathcal{I}_{\text{cal}}}) | Y_{n+1} \neq j\right] \tag{59}$$

Combining Equation (55) and Equation (59), we prove that Equation (50) holds. Finally, from Equation (51), we prove that:

$$\mathbb{P}\left[Y_{n+1} \in \hat{C}_{n,\alpha}^{\text{COLEP}\delta}(\tilde{X}_{n+1})\right] \geq 1 - \alpha \tag{60}$$

$\square$

### F.3 THM. 2 WITH FINITE-SAMPLE ERRORS BY RANDOMIZED SMOOTHING

**Theorem 6** (Thm. 2 with finite-sample errors of learning component certification by randomized smoothing)**.** *Consider a new test sample $X_{n+1}$ drawn from $P_{XY}$. For any bounded perturbation*

$\|\epsilon\|_2 \leq \delta$ in the input space and the adversarial sample $\tilde{X}_{n+1} := X_{n+1} + \epsilon$, we have the following guaranteed marginal coverage:

$$\mathbb{P}[Y_{n+1} \in \hat{C}_{n,\alpha}^{COLEP\delta}(\tilde{X}_{n+1})] \geq 1 - \alpha \tag{61}$$

if we construct the certified prediction set of $COLEP$ where

$$\hat{C}_{n,\alpha}^{COLEP\delta}(\tilde{X}_{n+1}) = \left\{ j \in [N_c] : S_{\hat{\pi}_j^{COLEP}}(\tilde{X}_{n+1}, 1) \leq Q_{1-\alpha+2\beta}(\{S_{\hat{\pi}_j^{COLEP\delta}}(X_i, \mathbb{I}_{[Y_i=j]})\}_{i \in \mathcal{I}_{cal}}) \right\} \tag{62}$$

and $S_{\hat{\pi}_j^{COLEP\delta}}(\cdot, \cdot)$ is a function of worst-case non-conformity score considering perturbation radius $\delta$:

$$S_{\hat{\pi}_j^{COLEP\delta}}(X_i, \mathbb{I}_{[Y_i=j]}) = \begin{cases} \mathbb{U}_{\beta,\delta}[\hat{\pi}_j^{COLEP}(X_i)] + u(1 - \mathbb{U}_{\beta,\delta}[\hat{\pi}_j^{COLEP}(X_i)]), & Y_i \neq j \\ 1 - \mathbb{L}_{\beta,\delta}[\hat{\pi}_j^{COLEP}(X_i)] + u\mathbb{L}_{\beta,\delta}[\hat{\pi}_j^{COLEP}(X_i)], & Y_i = j \end{cases} \tag{63}$$

where $\mathbb{U}_{\beta,\delta}[\hat{\pi}_j^{COLEP}(x)]$ and $\mathbb{L}_{\beta,\delta}[\hat{\pi}_j^{COLEP}(x)]$ are computed by plugging in the $\beta$-confidence probability bound $[\underline{\hat{\pi}_{j_\forall}}(x;\beta), \overline{\hat{\pi}_{j_\forall}}(x;\beta)]$ into Thm. 1. The high-confidence bound $[\underline{\hat{\pi}_{j_\forall}}(x;\beta), \overline{\hat{\pi}_{j_\forall}}(x;\beta)]$ can be obtained by randomized smoothing as the following:

$$\underline{\hat{\pi}_{j_\forall}}(x;\beta) = \Phi(\Phi^{-1}(\hat{\mathbb{E}}[\hat{\pi}_{j_\forall}(x+\sigma) - b_{Hoef}(\beta)]) - \delta/\sigma) - b_{Bern}(\beta) \tag{64}$$

$$\overline{\hat{\pi}_{j_\forall}}(x;\beta) = \Phi(\Phi^{-1}(\hat{\mathbb{E}}[\hat{\pi}_{j_\forall}(x+\sigma) + b_{Hoef}(\beta)]) - \delta/\sigma) + b_{Bern}(\beta) \tag{65}$$

where $\hat{\mathbb{E}}$ is the empirical mean of $N_{MC}$ Monte-Carlo samples during randomized smoothing and $b_{Hoef}(\beta) = \sqrt{\dfrac{\ln 1/\beta}{2N_{MC}}}$ is the Hoeffding's finite-sample error and $b_{Bern}(\beta) = \sqrt{\dfrac{2V \ln 2/\beta}{N_{MC}}} + \dfrac{7 \ln 2/\beta}{3(N_{MC}-1)}$ is the Berstein's finite-sample error.

*Proof of Thm. 6.* Bounding the finite-sample error of randomized smoothing similarly as Theorem 1 in Anonymous (2023), we get that for a given $j_\forall \in [N_c + L]$, we have the following:

$$\mathbb{P}\left[\mathbb{L}_{\beta,\delta}\left[\hat{\pi}_j^{COLEP}(\tilde{X}_{n+1})\right] \leq \hat{\pi}_j^{COLEP}(\tilde{X}_{n+1}) \leq \mathbb{U}_{\beta,\delta}\left[\hat{\pi}_j^{COLEP}(\tilde{X}_{n+1})\right]\right] \geq 1 - 2\beta \tag{66}$$

We can upper bound the probability of miscoverage of the main task with the probability of miscoverage of label classes as follows:

$$\mathbb{P}\left[Y_{n+1} \notin \hat{C}_{n,\alpha}^{COLEP\delta}(\tilde{X}_{n+1})\right] \tag{67}$$

$$\leq \max_{j \in [N_c]} \mathbb{P}\left[\mathbb{I}_{[Y_{n+1}=j]} \notin \hat{C}_{n,\alpha,j}^{COLEP\delta}(\tilde{X}_{n+1})\right] \tag{68}$$

$$\leq \max_{j \in [N_c]} \mathbb{P}\left[S_{\hat{\pi}_j^{COLEP}}(\tilde{X}_{n+1}, \mathbb{I}_{[Y_{n+1}=j]}) > Q_{1-\alpha+2\beta}(S_{\hat{\pi}_j^{COLEP\delta}}(\{X_i, \mathbb{I}_{[Y_i=j]}\}_{i \in \mathcal{I}_{cal}})\right] \tag{69}$$

$$\leq \max_{j \in [N_c]} \mathbb{P}\left[S_{\hat{\pi}_j^{COLEP\delta}}(X_{n+1}, \mathbb{I}_{[Y_{n+1}=j]}) > Q_{1-\alpha+2\beta}(S_{\hat{\pi}_j^{COLEP\delta}}(\{X_i, \mathbb{I}_{[Y_i=j]}\}_{i \in \mathcal{I}_{cal}})\right] + 2\beta \tag{70}$$

$$\leq \alpha - 2\beta + 2\beta \tag{71}$$

$$\leq \alpha \tag{72}$$

where $\hat{C}_{n,\alpha,j}^{COLEP\delta}(\tilde{X}_{n+1}) := \left\{ q^y \in \{0,1\} : S_{\hat{\pi}_j^{COLEP}}(\tilde{X}_{n+1}, y) \leq Q_{1-\alpha+2\beta}(S_{\hat{\pi}_j^{COLEP\delta}}(\{X_i, \mathbb{I}_{[Y_i=j]}\}_{i \in \mathcal{I}_{cal}}) \right\}$ is the certified prediction set for the class label $j$.

Note that Equation (70) holds because we have $S_{\hat{\pi}_j^{COLEP\delta}}(X_{n+1}, \mathbb{I}_{[Y_{n+1}=j]}) \geq S_{\hat{\pi}_j^{COLEP}}(\tilde{X}_{n+1}, \mathbb{I}_{[Y_{n+1}=j]})$. Next, we will prove Equation (70) rigorously. We first consider

the probability of miscoverage conditioned on $Y_{n+1} = j$.

$$\mathbb{P}\left[S_{\hat{\pi}_j^{\text{COLEP}}}(\tilde{X}_{n+1}, \mathbb{I}_{[Y_{n+1}=j]}) > Q_{1-\alpha+2\beta}(S_{\hat{\pi}_j^{\text{COLEP}}\delta}(\{X_i, \mathbb{I}_{[Y_i=j]}\}_{i \in \mathcal{I}_{\text{cal}}}) \,|\, Y_{n+1} = j\right] \tag{73}$$

$$\leq \mathbb{P}\left[1 - (1-u)\hat{\pi}_j^{\text{COLEP}}(\tilde{X}_{n+1}) > Q_{1-\alpha+2\beta}(S_{\hat{\pi}_j^{\text{COLEP}}\delta}(\{X_i, \mathbb{I}_{[Y_i=j]}\}_{i \in \mathcal{I}_{\text{cal}}}) \,|\, Y_{n+1} = j\right] \tag{74}$$

$$\leq \mathbb{P}\left[1 - (1-u) \cdot \mathbb{L}_{\beta,\delta}\left[\hat{\pi}_j^{\text{COLEP}}(\tilde{X}_{n+1})\right] > Q_{1-\alpha+2\beta}(S_{\hat{\pi}_j^{\text{COLEP}}\delta}(\{X_i, \mathbb{I}_{[Y_i=j]}\}_{i \in \mathcal{I}_{\text{cal}}}) \,|\, Y_{n+1} = j\right] \tag{75}$$

$$+ \mathbb{P}\left[\mathbb{L}_{\beta,\delta}\left[\hat{\pi}_j^{\text{COLEP}}(\tilde{X}_{n+1})\right] \leq \hat{\pi}_j^{\text{COLEP}}(\tilde{X}_{n+1})\right] \tag{76}$$

$$\leq \mathbb{P}\left[S_{\hat{\pi}_j^{\text{COLEP}}\delta}(X_{n+1}, \mathbb{I}_{[Y_{n+1}=j]}) > Q_{1-\alpha+2\beta}(S_{\hat{\pi}_j^{\text{COLEP}}\delta}(\{X_i, \mathbb{I}_{[Y_i=j]}\}_{i \in \mathcal{I}_{\text{cal}}}) \,|\, Y_{n+1} = j\right] + 2\beta \tag{77}$$

Similarly, we consider the probability of miscoverage conditioned on $Y_{n+1} \neq j$.

$$\mathbb{P}\left[S_{\hat{\pi}_j^{\text{COLEP}}}(\tilde{X}_{n+1}, \mathbb{I}_{[Y_{n+1}=j]}) > Q_{1-\alpha+2\beta}(S_{\hat{\pi}_j^{\text{COLEP}}\delta}(\{X_i, \mathbb{I}_{[Y_i=j]}\}_{i \in \mathcal{I}_{\text{cal}}}) \,|\, Y_{n+1} \neq j\right] \tag{78}$$

$$\leq \mathbb{P}\left[u + (1-u)\hat{\pi}_j^{\text{COLEP}}(\tilde{X}_{n+1}) > Q_{1-\alpha+2\beta}(S_{\hat{\pi}_j^{\text{COLEP}}\delta}(\{X_i, \mathbb{I}_{[Y_i=j]}\}_{i \in \mathcal{I}_{\text{cal}}}) \,|\, Y_{n+1} \neq j\right] \tag{79}$$

$$\leq \mathbb{P}\left[u + (1-u) \cdot \mathbb{U}_{\beta,\delta}\left[\hat{\pi}_j^{\text{COLEP}}(\tilde{X}_{n+1})\right] > Q_{1-\alpha+2\beta}(S_{\hat{\pi}_j^{\text{COLEP}}\delta}(\{X_i, \mathbb{I}_{[Y_i=j]}\}_{i \in \mathcal{I}_{\text{cal}}}) \,|\, Y_{n+1} \neq j\right] \tag{80}$$

$$+ \mathbb{P}\left[\mathbb{U}_{\beta,\delta}\left[\hat{\pi}_j^{\text{COLEP}}(\tilde{X}_{n+1})\right] \geq \hat{\pi}_j^{\text{COLEP}}(\tilde{X}_{n+1})\right] \tag{81}$$

$$\leq \mathbb{P}\left[S_{\hat{\pi}_j^{\text{COLEP}}\delta}(X_{n+1}, \mathbb{I}_{[Y_{n+1}=j]}) > Q_{1-\alpha+2\beta}(S_{\hat{\pi}_j^{\text{COLEP}}\delta}(\{X_i, \mathbb{I}_{[Y_i=j]}\}_{i \in \mathcal{I}_{\text{cal}}}) \,|\, Y_{n+1} \neq j\right] + 2\beta \tag{82}$$

Combining Equation (77) and Equation (82), we prove that Equation (70) holds. Finally, from Equation (72), we prove that:

$$\mathbb{P}\left[Y_{n+1} \in \hat{C}_{n,\alpha}^{\text{COLEP}\delta}(\tilde{X}_{n+1})\right] \geq 1 - \alpha \tag{83}$$

$$\square$$

### F.4 PROOF OF THM. 3

**Theorem 3** (Recall). Consider the new sample $X_{n+1}$ drawn from $P_{XY}$ and adversarial sample $\tilde{X}_{n+1} := X_{n+1} + \epsilon$ with any perturbation $\|\epsilon\|_2 \leq \delta$ in the input space. We have:

$$\mathbb{P}[Y_{n+1} \in \hat{C}_{n,\alpha}^{\text{COLEP}}(\tilde{X}_{n+1})] \geq \tau^{\text{COLEP}_{\text{cer}}} := \min_{j \in [N_c]}\left\{\tau_j^{\text{COLEP}_{\text{cer}}}\right\}, \tag{84}$$

where the certified coverage of the $j$-th class label $\tau_j^{\text{COLEP}_{\text{cer}}}$ is formulated as:

$$\tau_j^{\text{COLEP}_{\text{cer}}} = \max\left\{\tau : Q_\tau(\{S_{\hat{\pi}_j^{\text{COLEP}}\delta}(X_i, \mathbb{I}_{[Y_i=j]})\}_{i \in \mathcal{I}_{\text{cal}}}) \leq Q_{1-\alpha}(\{S_{\hat{\pi}_j^{\text{COLEP}}}(X_i, \mathbb{I}_{[Y_i=j]})\}_{i \in \mathcal{I}_{\text{cal}}})\right\}. \tag{85}$$

*Proof of Thm. 3.* We can bound the probability of coverage of the main task with the probability of coverage of label classes as follows:

$$\mathbb{P}\left[Y_{n+1} \in \hat{C}_{n,\alpha}^{\text{COLEP}}(\tilde{X}_{n+1})\right] \tag{86}$$

$$\geq \min_{j \in [N_c]} \mathbb{P}\left[\mathbb{I}_{[Y_{n+1}=j]} \in \hat{C}_{n,\alpha,j}^{\text{COLEP}}(\tilde{X}_{n+1})\right] \tag{87}$$

$$\geq \min_{j \in [N_c]} \mathbb{P}\left[S_{\hat{\pi}_j^{\text{COLEP}}}(\tilde{X}_{n+1}, \mathbb{I}_{[Y_{n+1}=j]}) \leq Q_{1-\alpha}(S_{\hat{\pi}_j^{\text{COLEP}}}(\{X_i, \mathbb{I}_{[Y_i=j]}\}_{i \in \mathcal{I}_{\text{cal}}}))\right] \tag{88}$$

$$\geq \min_{j \in [N_c]} \mathbb{P}\left[S_{\hat{\pi}_j^{\text{COLEP}}}(\tilde{X}_{n+1}, \mathbb{I}_{[Y_{n+1}=j]}) \leq Q_{\tau_j^{\text{COLEP}_{\text{cer}}}}(S_{\hat{\pi}_j^{\text{COLEP}}\delta}(\{X_i, \mathbb{I}_{[Y_i=j]}\}_{i \in \mathcal{I}_{\text{cal}}}))\right] \tag{89}$$

$$\geq \min_{j \in [N_c]} \mathbb{P}\left[S_{\hat{\pi}_j^{\text{COLEP}}\delta}(X_{n+1}, \mathbb{I}_{[Y_{n+1}=j]}) \leq Q_{\tau_j^{\text{COLEP}_{\text{cer}}}}(S_{\hat{\pi}_j^{\text{COLEP}}\delta}(\{X_i, \mathbb{I}_{[Y_i=j]}\}_{i \in \mathcal{I}_{\text{cal}}}))\right] \tag{90}$$

$$\geq \min_{j \in [N_c]}\{\tau_j^{\text{COLEP}_{\text{cer}}}\} \tag{91}$$

where Equation (90) holds because $S_{\hat{\pi}_j^{\text{COLEP}\delta}}(X_{n+1}, \mathbb{I}_{[Y_{n+1}=j]}) \geq S_{\hat{\pi}_j^{\text{COLEP}}}(\tilde{X}_{n+1}, \mathbb{I}_{[Y_{n+1}=j]})$ and is proved rigorously in the proof of Thm. 2 in Appendix F.2.

$\square$

## G  FINITE-SAMPLE CERTIFIED COVERAGE

In this section, we consider the finite-sample error induced by the finite size of the calibration set $\mathcal{I}_{\text{cal}}$. We provide the finite-sample certified coverage in the adversary setting as follows.

**Corollary 1** (Certified Coverage with Finite Samples). *Consider the new sample $X_{n+1}$ drawn from $P_{XY}$ and adversarial sample $\tilde{X}_{n+1} := X_{n+1} + \epsilon$ with any perturbation $\|\epsilon\|_2 \leq \delta$ in the input space. We have the following finite-sample certified coverage:*

$$\mathbb{P}[Y_{n+1} \in \hat{C}_{n,\alpha}^{COLEP}(\tilde{X}_{n+1})] \geq \left(1 + \frac{1}{|\mathcal{I}_{cal}|}\right) \min_{j \in [N_c]} \left\{\tau_j^{COLEP_{cer}}\right\} - \frac{\sqrt{\log(2)/2} + \sqrt{2}/(4\sqrt{\log(2)} + 8/\pi)}{\sqrt{|\mathcal{I}_{cal}|}}$$
(92)

*where $|\mathcal{I}_{cal}|$ is the size of the calibration set in conformal prediction.*

*Proof sketch and remarks.* Leveraging corollary 1 of (Massart, 1990) and following the sketch in (Yang & Kuchibhotla, 2021), we can prove the finite-sample error bound. Combining the finite-sample error with the certified coverage bound in Thm. 3 concludes the proof. Note that the union bound is applied to multiple models in (Yang & Kuchibhotla, 2021) while we circumvent it by aggregating individual results for class labels subtly, and thus, COLEP achieves an improved sample complexity of conformal prediction compared with ensemble learning given the same tolerance of finite-sample error.

*Proof of Corollary 1.* From Corollary 1 of (Massart, 1990), we have that for all $\epsilon > 0$,

$$\mathbb{P}\left[\sup_{t \in \mathbb{R}} \left| \frac{1}{|\mathcal{I}_{\text{cal}}|} \sum_{i \in \mathcal{I}_{\text{cal}}} \mathbb{I}_{\left[S_{\hat{\pi}_j^{\text{COLEP}\delta}}(X_i, Y_i) \leq t\right]} - \mathbb{P}\left[S_{\hat{\pi}_j^{\text{COLEP}\delta}}(X_{n+1}, Y_{n+1}) \leq t\right] \right| \geq \frac{\epsilon}{\sqrt{|\mathcal{I}_{\text{cal}}|}} \right] \leq 2\exp\left\{-2\epsilon^2\right\}$$
(93)

For ease of notation, we let:

$$W := \sqrt{|\mathcal{I}_{\text{cal}}|} \sup_{t \in \mathbb{R}} \left| \frac{1}{|\mathcal{I}_{\text{cal}}|} \sum_{i \in \mathcal{I}_{\text{cal}}} \mathbb{I}_{\left[S_{\hat{\pi}_j^{\text{COLEP}\delta}}(X_i, Y_i) \leq t\right]} - \mathbb{P}\left[S_{\hat{\pi}_j^{\text{COLEP}\delta}}(X_{n+1}, Y_{n+1}) \leq t\right] \right|.$$
(94)

We let $t = Q_{\tau_j^{\text{COLEP}_{\text{cer}}}}(S_{\hat{\pi}_j^{\text{COLEP}\delta}}(\{X_i, \mathbb{I}_{[Y_i=j]}\}_{i \in \mathcal{I}_{\text{cal}}})$. It implies that:

$$\begin{aligned}
&\mathbb{P}\left[S_{\hat{\pi}_j^{\text{COLEP}\delta}}(X_{n+1}, Y_{n+1}) \leq Q_{\tau_j^{\text{COLEP}_{\text{cer}}}}(S_{\hat{\pi}_j^{\text{COLEP}\delta}}(\{X_i, \mathbb{I}_{[Y_i=j]}\}_{i \in \mathcal{I}_{\text{cal}}}))\right] \\
&\geq \frac{1}{|\mathcal{I}_{\text{cal}}|} \sum_{i \in \mathcal{I}_{\text{cal}}} \mathbb{I}_{\left[S_{\hat{\pi}_j^{\text{COLEP}\delta}}(X_i, Y_i) \leq Q_{\tau_j^{\text{COLEP}_{\text{cer}}}}(S_{\hat{\pi}_j^{\text{COLEP}\delta}}(\{X_i, \mathbb{I}_{[Y_i=j]}\}_{i \in \mathcal{I}_{\text{cal}}}))\right]} - \frac{W}{\sqrt{|\mathcal{I}_{\text{cal}}|}} \\
&\geq \frac{\lceil (1 + |\mathcal{I}_{\text{cal}}|)\tau_j^{\text{COLEP}_{\text{cer}}} \rceil}{|\mathcal{I}_{\text{cal}}|} - \frac{W}{\sqrt{|\mathcal{I}_{\text{cal}}|}} \\
&\geq \left(1 + \frac{1}{|\mathcal{I}_{\text{cal}}|}\right) \tau_j^{\text{COLEP}_{\text{cer}}} - \frac{W}{\sqrt{|\mathcal{I}_{\text{cal}}|}}
\end{aligned}$$
(95)

Then using Equation (93), we can bound $\mathbb{E}[W]$ as follows:

$$\mathbb{E}[W] = \int_0^\infty \mathbb{P}\left[W \geq u\right] du \tag{96}$$

$$\leq \int_0^{\sqrt{\log(2)/2}} \mathbb{P}\left[W \geq u\right] du + \int_{\sqrt{\log(2)/2}}^\infty \mathbb{P}\left[W \geq u\right] du \tag{97}$$

$$\leq \sqrt{\log(2)/2} + 2 \int_{\sqrt{\log(2)/2}}^\infty \exp\{-2u^2\} du \tag{98}$$

$$\overset{(a)}{\leq} \sqrt{\log(2)/2} + \frac{\sqrt{2}/2}{2\sqrt{\log(2)} + 4/\pi} \tag{99}$$

Inequality (a) follows from Formula 7.1.13 of (Abramowitz & Stegun, 1948). Finally, combining the results in Thm. 3, we get:

$$\mathbb{P}[Y_{n+1} \in \hat{C}_{n,\alpha}^{\text{COLEP}}(\tilde{X}_{n+1})] \geq \left(1 + \frac{1}{|\mathcal{I}_{\text{cal}}|}\right) \min_{j \in [N_c]} \left\{\tau_j^{\text{COLEP}_{\text{cer}}}\right\} - \frac{\sqrt{\log(2)/2} + \sqrt{2}/(4\sqrt{\log(2)} + 8/\pi)}{\sqrt{|\mathcal{I}_{\text{cal}}|}} \tag{100}$$

$\square$

# H  OMITTED PROOFS IN SECTION 5

## H.1  PROOF OF LEMMA 5.1

**Lemma 6.1** (Recall). For any class probability within models $j \in [N_c]$ and data sample $(X, Y)$ drawn from the mixture distribution $\mathcal{D}_m$, we can prove that:

$$\mathbb{E}\left[\hat{\pi}_j^{\text{COLEP}}(X)\,|\,Y \neq j\right] \leq \mathbb{E}[\hat{\pi}_j(X) - \epsilon_{j,0}\,|\,Y \neq j], \mathbb{E}\left[\hat{\pi}_j^{\text{COLEP}}(X)\,|\,Y = j\right] \geq \mathbb{E}\left[\hat{\pi}_j(X) + \epsilon_{j,1}\,|\,Y = j\right] \tag{101}$$

$$\epsilon_{j,0} = \sum_{\mathcal{D} \in \{\mathcal{D}_a, \mathcal{D}_b\}} p_{\mathcal{D}} \left[ \sum_{r \in \mathcal{P}_d(j)} \beta_r U_j^{(r)} Z_{j,\mathcal{D}}^{(r)} \left( \hat{\pi}_j - \frac{\hat{\pi}_j}{\hat{\pi}_j + (1 - \hat{\pi}_j)/\lambda_{j,\mathcal{D}}^{(r)}} \right) + \sum_{r \in \mathcal{P}_s(j)} \beta_r \left( \hat{\pi}_j - \frac{\hat{\pi}_j}{\hat{\pi}_j + (1 - \hat{\pi}_j)T_{j,\mathcal{D}}^{(r)}} \right) \right]$$

$$\epsilon_{j,1} = \sum_{\mathcal{D} \in \{\mathcal{D}_a, \mathcal{D}_b\}} p_{\mathcal{D}} \left[ \sum_{r \in \mathcal{P}_d(j)} \beta_r \left( \frac{\hat{\pi}_j}{\hat{\pi}_j + (1 - \hat{\pi}_j)/T_{j,\mathcal{D}}^{(r)}} - \hat{\pi}_j \right) + \sum_{r \in \mathcal{P}_s(j)} \beta_r U_j^{(r)} Z_{j,\mathcal{D}}^{(r)} \left( \frac{\hat{\pi}_j}{\hat{\pi}_j + (1 - \hat{\pi}_j)\lambda_{j,\mathcal{D}}^{(r)}} - \hat{\pi}_j \right) \right] \tag{102}$$

where $\lambda_{j,\mathcal{D}}^{(r)} = (1/T_{j,\mathcal{D}}^{(r)} + e^{-w} - 1)$ and we shorthand $\hat{\pi}_j(X)$ as $\hat{\pi}_j$.

*Proof of Lemma 5.1.* We first consider one PC and then analyze the effectiveness of $R$ linearly combined PCs. Without loss of generality, we focus on the class conditional probability by the $r$-th PC ($r \in [R]$) for the class label $j \in [N_c]$.

Recall that we map a set of implication rules encoded in the $r$-th PC to an undirected graph $\mathcal{G}_r = (\mathcal{V}_r, \mathcal{E}_r)$, where $\mathcal{V}_r = [N_c + L]$ corresponds to the Boolean vector of class and concept predictions in the learning stage. There exists an edge between node $j_1 \in \mathcal{V}_r$ and node $j_2 \in \mathcal{V}_r$ iff they are connected by an implication rule $j_1 \implies j_2$ or $j_2 \implies j_1$. Let $\mathcal{A}_r(j_\forall)$ be the set of nodes in the connected component of node $j_\forall$ except for node $j_\forall$ (i.e., $j_\forall \notin \mathcal{A}_r(j_\forall)$). Let $\mathcal{A}_{r,d}(j_\forall) \subseteq \mathcal{A}_r(j_\forall)$ be the set of conditional variables and $\mathcal{A}_{r,s}(j_\forall) \subseteq \mathcal{A}_r(j_\forall)$ be the set of consequence variables. Denote $M_j$ be the universal set of assignments except for the $j$-th element. Let $\mathcal{P}_d(j)$ and $\mathcal{P}_s(j)$ be the set of PC index where $j$ appears as conditional variables and consequence variables, respectively.

We consider a mixture of benign and adversarial distributions denoted by $\mathcal{D}_m$, which indicates that an input $x$ is either a benign example drawn from $\mathcal{D}_b$ with probability $p_{\mathcal{D}_b} := \mathbb{P}[x \sim \mathcal{D}_b]$ or an adversarial example drawn from $\mathcal{D}_a$ with probability $p_{\mathcal{D}_a} := \mathbb{P}[x \sim \mathcal{D}_a]$, where $p_{\mathcal{D}_b} + p_{\mathcal{D}_a} = 1$. We analyze the effectiveness of PCs in four cases: case (1) $r \in \mathcal{P}_d(j)$ and $Y \neq j$, case (2) $r \in \mathcal{P}_d(j)$ and $Y = j$, case (3) $r \in \mathcal{P}_s(j)$ and $Y = j$, and case (4) $r \in \mathcal{P}_s(j)$ and $Y \neq j$. We leave the probability conditions on $Y = j$ or $Y \neq j$ in each case for simplicity and consider them in the conclusions.

**Case (1)** $r \in \mathcal{P}_d(j)$ and $Y \neq j$. We can derive the class conditional probability by PCs as follows:

$$\hat{\pi}_j^{(r)}(x) = \frac{\sum_{\mu \in M, \, \mu_j = 1} \exp\left\{\sum_{j_\forall = 1}^{N_c + L} T(\hat{\pi}_{j_\forall}(x), \mu_{j_\forall})\right\} F_r(\mu)}{\sum_{\mu \in M} \exp\left\{\sum_{j_\forall = 1}^{N_c + L} T(\hat{\pi}_{j_\forall}(x), \mu_{j_\forall})\right\} F_r(\mu)} \tag{103}$$

$$= \frac{1}{1 + \dfrac{\sum_{\mu \in M, \, \mu_j = 0} \exp\left\{\sum_{j_\forall = 1}^{N_c + L} T(\hat{\pi}_{j_\forall}(x), \mu_{j_\forall})\right\} F_r(\mu)}{\sum_{\mu \in M, \, \mu_j = 1} \exp\left\{\sum_{j_\forall = 1}^{N_c + L} T(\hat{\pi}_{j_\forall}(x), \mu_{j_\forall})\right\} F_r(\mu)}} \tag{104}$$

$$= \left\{1 + \frac{(1 - \hat{\pi}_j(x)) \sum_{\mu \in M, \, \mu_j = 0} \exp\left\{\sum_{j_\forall \in [N_c + L] \setminus \{j\}} T(\hat{\pi}_{j_\forall}(x), \mu_{j_\forall})\right\} F_r(\mu)}{\hat{\pi}_j(x) \sum_{\mu \in M, \, \mu_j = 1} \exp\left\{\sum_{j_\forall \in [N_c + L] \setminus \{j\}} T(\hat{\pi}_{j_\forall}(x), \mu_{j_\forall})\right\} F_r(\mu)}\right\}^{-1} \tag{105}$$

Note that since $r \in \mathcal{P}_s(j)$, we have $\sum_{\mu \in M, \, \mu_j = 0} \exp\left\{\sum_{j_\forall \in [N_c + L] \setminus \{j\}} T(\hat{\pi}_{j_\forall}(x), \mu_{j_\forall})\right\} F_r(\mu) \geq \sum_{\mu \in M, \, \mu_j = 1} \exp\left\{\sum_{j_\forall \in [N_c + L] \setminus \{j\}} T(\hat{\pi}_{j_\forall}(x), \mu_{j_\forall})\right\} F_r(\mu)$, and thus, $\hat{\pi}_j^{(r)}(x) \leq \hat{\pi}_j(x)$ holds. Next, we analyze the gap to get a tight bound based on the distribution of the Boolean vector of the ground truth class and concepts $\nu$.

*Case (1a)*: If $\exists j_\forall \in \mathcal{A}_{r,s}(j)$, $\nu_{j_\forall} = 0$, then we have:

$$\frac{\sum_{\mu \in M, \, \mu_j = 0} \exp\left\{\sum_{j_\forall \in [N_c + L] \setminus \{j\}} T(\hat{\pi}_{j_\forall}(x), \mu_{j_\forall})\right\} F_r(\mu)}{\sum_{\mu \in M, \, \mu_j = 1} \exp\left\{\sum_{j_\forall \in [N_c + L] \setminus \{j\}} T(\hat{\pi}_{j_\forall}(x), \mu_{j_\forall})\right\} F_r(\mu)} \tag{106}$$

$$\geq \frac{\exp\left\{\sum_{j_\forall \in [N_c + L] \setminus \{j\}} T(\hat{\pi}_{j_\forall}(x), \nu_{j_\forall})\right\} F_r(\nu)}{\sum_{\mu \in M, \, \mu_j = 1} \exp\left\{\sum_{j_\forall \in [N_c + L] \setminus \{j\}} T(\hat{\pi}_{j_\forall}(x), \mu_{j_\forall})\right\} F_r(\mu)} \tag{107}$$

$$\geq \frac{1}{e^{-w} + \sum_{\mu_j = 1, \mu \neq \nu} \exp\left\{\sum_{j_\forall \in [N_c + L] \setminus \{j\}} \left(T(\hat{\pi}_{j_\forall}(x), \mu_{j_\forall}) - T(\hat{\pi}_{j_\forall}(x), \nu_{j_\forall})\right)\right\} F_r(\mu)/F_r(\nu)} \tag{108}$$

When we have models such that $\hat{\pi}_{j_\forall}(x) > 1 - t_{j_\forall, \mathcal{D}} \,|\, \nu_{j_\forall} = 0$ or $\hat{\pi}_{j_\forall}(x) < 1 - t_{j_\forall, \mathcal{D}} \,|\, \nu_{j_\forall} = 1$, $\exists j_\forall \in \mathcal{A}_r(j)$ (with probability $1 - Z_{j, \mathcal{D}}^{(r)}$), we leverage the general bound in the case,

$$\mathbb{E}[\hat{\pi}_j^{(r)}(x) | Y \neq j] \leq \mathbb{E}[\hat{\pi}_j(x) | Y \neq j] \tag{109}$$

When we have models of high quality such that $\hat{\pi}_{j_\forall}(x) \leq 1 - t_{j_\forall, \mathcal{D}} \,|\, \nu_{j_\forall} = 0$ and $\hat{\pi}_{j_\forall}(x) \geq 1 - t_{j_\forall, \mathcal{D}} \,|\, \nu_{j_\forall} = 1$, $\forall j_\forall \in \mathcal{A}_r(j)$ (with probability $Z_{j, \mathcal{D}}^{(r)}$), we can derive a tighter bound as follows:

$$\sum_{\mu_j = 1, \mu \neq \nu} \exp\left\{\sum_{j_\forall \in [N_c + L] \setminus \{j\}} \left(T(\hat{\pi}_{j_\forall}(x), \mu_{j_\forall}) - T(\hat{\pi}_{j_\forall}(x), \nu_{j_\forall})\right)\right\} F_r(\mu)/F_r(\nu) \tag{110}$$

$$\leq \sum_{\mu_j = 1, \mu \neq \nu} \exp\left\{\sum_{j_\forall \in [N_c + L] \setminus \{j\}} \left(T(\hat{\pi}_{j_\forall}(x), \mu_{j_\forall}) - T(\hat{\pi}_{j_\forall}(x), \nu_{j_\forall})\right)\right\} \tag{111}$$

$$\overset{(a)}{\leq} \prod_{j_\forall \in \mathcal{A}_r(j)} \left(1 + \frac{1 - t_{j_\forall, \mathcal{D}}}{t_{j_\forall, \mathcal{D}}}\right) - 1 \tag{112}$$

$$\leq \frac{1}{\Pi_{j_\forall \in \mathcal{A}_r(j)} t_{j_\forall, \mathcal{D}}} - 1 := 1/T_{j, \mathcal{D}}^{(r)} - 1 \tag{113}$$

In inequality (a), we plug in the lower bound of confidence of a correct prediction $t_{j_\forall, \mathcal{D}}$ and the upper bound of confidence of a wrong prediction $1 - t_{j_\forall, \mathcal{D}}$ for label $j_\forall$.

Combining Equations (109) and (113), we get the following bound for case (1a):

$$\mathbb{E}\left[\hat{\pi}_j^{(r)}(x) | Y \neq j\right] \leq \mathbb{E}\left[(1 - Z_{j,\mathcal{D}}^{(r)})\hat{\pi}_j(x) + Z_{j,\mathcal{D}}^{(r)} \frac{\hat{\pi}_j(x)}{\hat{\pi}_j(x) + (1 - \hat{\pi}_j(x))/\lambda_{j,\mathcal{D}}^{(r)}} | Y \neq j\right] \quad (114)$$

where $\lambda_{j,\mathcal{D}}^{(r)} = (1/T_{j,\mathcal{D}}^{(r)} + e^{-w} - 1)$.

*Case (1b)*: If $\forall j_\forall \in \mathcal{A}_{r,s}(j)$, $\nu_{j_\forall} = 1$, then we use the general bound in the case:

$$\mathbb{E}[\hat{\pi}_j^{(r)}(x) | Y \neq j] \leq \mathbb{E}[\hat{\pi}_j(x) | Y \neq j] \quad (115)$$

Note that case (1a) is with probability $U_j^{(r)}$ and case (1b) is with probability $1 - U_j^{(r)}$. Combining Equation (114) in *case (1a)* and Equation (115) in *case (1b)*, we have:

$$\mathbb{E}[\hat{\pi}_j^{(r)}(x) | Y \neq j] \leq \mathbb{E}\left[U_j^{(r)}\left[(1 - Z_{j,\mathcal{D}}^{(r)})\hat{\pi}_j(x) + Z_{j,\mathcal{D}}^{(r)} \frac{\hat{\pi}_j(x)}{\hat{\pi}_j(x) + (1 - \hat{\pi}_j(x))/\lambda_{j,\mathcal{D}}^{(r)}}\right] + (1 - U_j^{(r)})\hat{\pi}_j(x) | Y \neq j\right]$$

$$\leq \mathbb{E}\left[\hat{\pi}_j(x) - U_j^{(r)} Z_{j,\mathcal{D}}^{(r)}\left(\hat{\pi}_j(x) - \frac{\hat{\pi}_j(x)}{\hat{\pi}_j(x) + (1 - \hat{\pi}_j(x))/\lambda_{j,\mathcal{D}}^{(r)}}\right) | Y \neq j\right]$$

$$(116)$$

**Case (2)**: $r \in \mathcal{P}_d(j)$ and $Y = j$. Since in the $r$-th PC $j$ is the index of a conditional variable, we have $\forall j_\forall \in \mathcal{A}_{r,s}(j)$, $\nu_{j_\forall} = 1$, where $\nu$ is the ground truth Boolean vector. Then we have the following:

$$\frac{\sum_{\mu \in M, \mu_j = 0} \exp\left\{\sum_{j_\forall \in [N_c + L] \setminus \{j\}} T(\hat{\pi}_{j_\forall}(x), \mu_{j_\forall})\right\} F_r(\mu)}{\sum_{\mu \in M, \mu_j = 1} \exp\left\{\sum_{j_\forall \in [N_c + L] \setminus \{j\}} T(\hat{\pi}_{j_\forall}(x), \mu_{j_\forall})\right\} F_r(\mu)} \quad (117)$$

$$\leq \frac{\sum_{\mu \in M, \mu_j = 0} \exp\left\{\sum_{j_\forall \in [N_c + L] \setminus \{j\}} T(\hat{\pi}_{j_\forall}(x), \mu_{j_\forall})\right\} F_r(\mu)}{\exp\left\{\sum_{j_\forall \in [N_c + L] \setminus \{j\}} T(\hat{\pi}_{j_\forall}(x), \nu_{j_\forall})\right\} F_r(\nu)} \quad (118)$$

$$\leq \prod_{j_\forall \in \mathcal{A}_r(j)} \left(1 + \frac{1 - t_{j_\forall, \mathcal{D}}}{t_{j_\forall, \mathcal{D}}}\right) := 1/T_{j,\mathcal{D}}^{(r)} \quad (119)$$

Then the following holds in case (2):

$$\mathbb{E}\left[\hat{\pi}_j^{(r)}(x) | Y = j\right] \geq \mathbb{E}\left[\frac{\hat{\pi}_j(x)}{\hat{\pi}_j(x) + (1 - \hat{\pi}_j(x))/T_{j,\mathcal{D}}^{(r)}} | Y = j\right] \quad (120)$$

**Case (3)**: $r \in \mathcal{P}_s(j)$ and $Y = j$. As in Case (1), we similarly consider cases based on the distribution of ground truth Boolean vector $\nu$ to get a tighter bound.

*Case (3a)*: $\exists j_\forall \in \mathcal{A}_{r,d}(j)$, $\nu_{j_\forall} = 1$.

When we have models such that $\hat{\pi}_{j_\forall}(x) > 1 - t_{j_\forall, \mathcal{D}} | \nu_{j_\forall} = 0$ or $\hat{\pi}_{j_\forall}(x) < 1 - t_{j_\forall, \mathcal{D}} | \nu_{j_\forall} = 1$, $\exists j_\forall \in \mathcal{A}_r(j)$ (with probability $1 - Z_{j,\mathcal{D}}^{(r)}$), we leverage the general bound in the case,

$$\mathbb{E}[\hat{\pi}_j^{(r)}(x) | Y \neq j] \leq \mathbb{E}[\hat{\pi}_j(x) | Y \neq j] \quad (121)$$

Equation (121) holds because in case (3), $j$ corresponds to a consequence variable in the $r$-th PC, and thus, we always have:

$$\sum_{\mu \in M, \mu_j = 0} \exp\left\{\sum_{j_\forall \in [N_c + L] \setminus \{j\}} T(\hat{\pi}_{j_\forall}(x), \mu_{j_\forall})\right\} F_r(\mu) \leq \sum_{\mu \in M, \mu_j = 1} \exp\left\{\sum_{j_\forall \in [N_c + L] \setminus \{j\}} T(\hat{\pi}_{j_\forall}(x), \mu_{j_\forall})\right\} F_r(\mu)$$

$$(122)$$

When we have models of high quality such that $\hat{\pi}_{j_\forall}(x) \leq 1 - t_{j_\forall,\mathcal{D}} \,|\, \nu_{j_\forall} = 0$ and $\hat{\pi}_{j_\forall}(x) \geq 1 - t_{j_\forall,\mathcal{D}} \,|\, \nu_{j_\forall} = 1$, $\forall j_\forall \in \mathcal{A}_r(j)$ (with probability $Z^{(r)}_{j,\mathcal{D}}$), we can derive a tighter bound as follows:

$$\frac{\sum_{\mu \in M, \, \mu_j=0} \exp\left\{\sum_{j_\forall \in [N_c+L]\setminus\{j\}} T(\hat{\pi}_{j_\forall}(x), \mu_{j_\forall})\right\} F_r(\mu)}{\sum_{\mu \in M, \, \mu_j=1} \exp\left\{\sum_{j_\forall \in [N_c+L]\setminus\{j\}} T(\hat{\pi}_{j_\forall}(x), \mu_{j_\forall})\right\} F_r(\mu)} \tag{123}$$

$$\leq \frac{\sum_{\mu \in M, \, \mu_j=0} \exp\left\{\sum_{j_\forall \in [N_c+L]\setminus\{j\}} T(\hat{\pi}_{j_\forall}(x), \mu_{j_\forall})\right\} F_r(\mu)}{\exp\left\{\sum_{j_\forall \in [N_c+L]\setminus\{j\}} T(\hat{\pi}_{j_\forall}(x), \nu_{j_\forall})\right\} F_r(\nu)} \tag{124}$$

$$\leq \exp\{-w\} + \prod_{j_\forall \in \mathcal{A}_r(j)} \left(1 + \frac{1 - t_{j_\forall,\mathcal{D}}}{t_{j_\forall,\mathcal{D}}}\right) - 1 := \lambda^{(r)}_{j,\mathcal{D}} \tag{125}$$

Therefore, the following holds in case (3a);

$$\mathbb{E}\left[\hat{\pi}^{(r)}_j(x)\,|\,Y=j\right] \geq \mathbb{E}\left[(1 - Z^{(r)}_{j,\mathcal{D}})\hat{\pi}_j(x) + Z^{(r)}_{j,\mathcal{D}} \frac{\hat{\pi}_j(x)}{\hat{\pi}_j(x) + (1 - \hat{\pi}_j(x))\lambda^{(r)}_{j,\mathcal{D}}}\,|\,Y=j\right] \tag{126}$$

*Case (3b)*: If $\forall j_\forall \in \mathcal{A}_{r,d}(j)$, $\nu_{j_\forall} = 0$, then we leverage the general bound:

$$\mathbb{E}\left[\hat{\pi}^{(r)}_j(x)\,|\,Y=j\right] \geq \mathbb{E}\left[\hat{\pi}_j(x)\,|\,Y=j\right] \tag{127}$$

Note that case (3a) is with probability $U^{(r)}_j$ and case (3b) is with probability $1 - U^{(r)}_j$. Combining Equations (126) and (127), the following holds:

$$\mathbb{E}\left[\hat{\pi}^{(r)}_j(x)\,|\,Y=j\right] \geq \mathbb{E}\left[\hat{\pi}_j(x) + U^{(r)}_j Z^{(r)}_{j,\mathcal{D}}\left(\frac{\hat{\pi}_j(x)}{\hat{\pi}_j(x) + (1 - \hat{\pi}_j(x))\lambda^{(r)}_{j,\mathcal{D}}} - \hat{\pi}_j(x)\right)\,|\,Y=j\right] \tag{128}$$

**Case (4)**: $r \in \mathcal{P}_s(j)$ and $Y \neq j$. Since $j$ is the index of a consequence variable, we have $\forall j_\forall \in \mathcal{A}_{r,d}(j)$, $\nu_{j_\forall} = 0$. Then we can lower bound the coefficients as follows:

$$\frac{\sum_{\mu \in M, \, \mu_j=0} \exp\left\{\sum_{j_\forall \in [N_c+L]\setminus\{j\}} T(\hat{\pi}_{j_\forall}(x), \mu_{j_\forall})\right\} F_r(\mu)}{\sum_{\mu \in M, \, \mu_j=1} \exp\left\{\sum_{j_\forall \in [N_c+L]\setminus\{j\}} T(\hat{\pi}_{j_\forall}(x), \mu_{j_\forall})\right\} F_r(\mu)} \tag{129}$$

$$\geq \frac{\exp\left\{\sum_{j_\forall \in [N_c+L]\setminus\{j\}} T(\hat{\pi}_{j_\forall}(x), \nu_{j_\forall})\right\} F_r(\nu)}{\sum_{\mu \in M, \, \mu_j=1} \exp\left\{\sum_{j_\forall \in [N_c+L]\setminus\{j\}} T(\hat{\pi}_{j_\forall}(x), \mu_{j_\forall})\right\} F_r(\mu)} \tag{130}$$

$$\geq \frac{1}{\prod_{j_\forall \in \mathcal{A}_r(j)} \left(1 + \frac{1 - t_{j_\forall,\mathcal{D}}}{t_{j_\forall,\mathcal{D}}}\right)} := T^{(r)}_{j,\mathcal{D}} \tag{131}$$

Thus, in case (4) we have:

$$\mathbb{E}[\hat{\pi}^{(r)}_j(x)\,|\,Y \neq j] \leq \mathbb{E}\left[\frac{\hat{\pi}_j(x)}{\hat{\pi}_j(x) + (1 - \hat{\pi}_j(x))T^{(r)}_{j,\mathcal{D}}}\,|\,Y \neq j\right] \tag{132}$$

Then we consider all $R$ PCs to get the effectiveness of COLEP on the class conditional probability. Combining Equation (116) in case (1) and Equation (132) in case (4), we finally have:

$$\mathbb{E}\left[\hat{\pi}^{\text{COLEP}}_j(X)\,|\,Y \neq j\right] \leq \mathbb{E}[\hat{\pi}_j(X) - \epsilon_{j,0}\,|\,Y \neq j] \tag{133}$$

where

$$\epsilon_{j,0} = \sum_{\mathcal{D} \in \{\mathcal{D}_a, \mathcal{D}_b\}} p_\mathcal{D} \left[\sum_{r \in \mathcal{P}_d(j)} \beta_r U^{(r)}_j Z^{(r)}_{j,\mathcal{D}}\left(\hat{\pi}_j - \frac{\hat{\pi}_j}{\hat{\pi}_j + (1 - \hat{\pi}_j)/\lambda^{(r)}_{j,\mathcal{D}}}\right) + \sum_{r \in \mathcal{P}_s(j)} \beta_r \left(\hat{\pi}_j - \frac{\hat{\pi}_j}{\hat{\pi}_j + (1 - \hat{\pi}_j)T^{(r)}_{j,\mathcal{D}}}\right)\right] \tag{134}$$

Combining Equation (120) in case (2) and Equation (128) in case (3), we finally have:

$$\mathbb{E}\left[\hat{\pi}_j^{\text{COLEP}}(X) \,|\, Y = j\right] \geq \mathbb{E}\left[\hat{\pi}_j(X) + \epsilon_{j,1} \,|\, Y = j\right] \tag{135}$$

where

$$\epsilon_{j,1} = \sum_{\mathcal{D} \in \{\mathcal{D}_a, \mathcal{D}_b\}} p_{\mathcal{D}} \left[ \sum_{r \in \mathcal{P}_d(j)} \beta_r \left( \frac{\hat{\pi}_j}{\hat{\pi}_j + (1-\hat{\pi}_j)/T_{j,\mathcal{D}}^{(r)}} - \hat{\pi}_j \right) + \sum_{r \in \mathcal{P}_s(j)} \beta_r U_j^{(r)} Z_{j,\mathcal{D}}^{(r)} \left( \frac{\hat{\pi}_j}{\hat{\pi}_j + (1-\hat{\pi}_j)\lambda_{j,\mathcal{D}}^{(r)}} - \hat{\pi}_j \right) \right] \tag{136}$$

$\square$

## H.2 PROOF OF THM. 4

**Theorem 4** (Recall). Consider the adversary setting that the calibration set $\mathcal{I}_{\text{cal}}$ consists of $n_{\mathcal{D}_b}$ samples drawn from the benign distribution $\mathcal{D}_b$, while the new sample $(X_{n+1}, Y_{n+1})$ is drawn $n_{\mathcal{D}_a}$ times from the adversarial distribution $\mathcal{D}_a$. Assume that $A(\hat{\pi}_j, \mathcal{D}_a) < 0.5 < A(\hat{\pi}_j, \mathcal{D}_b)$ for $j \in [N_c]$, where $A(\hat{\pi}_j, \mathcal{D})$ is the expectation of prediction accuracy of $\hat{\pi}_j$ on $\mathcal{D}$. Then we have:

$$\mathbb{P}[Y_{n+1} \in \hat{C}_{n,\alpha}^{\text{COLEP}}(\tilde{X}_{n+1})] > \mathbb{P}[Y_{n+1} \in \hat{C}_{n,\alpha}(\tilde{X}_{n+1})], \quad w.p.$$

$$1 - \max_{j \in [N_c]} \left\{ \exp\left\{ -2n_{\mathcal{D}_a}(0.5 - A(\hat{\pi}_j, \mathcal{D}_a))^2 \epsilon_{j,1,\mathcal{D}_a}^2 \right\} + n_{\mathcal{D}_b} \exp\left\{ -2n_{\mathcal{D}_b}\left((A(\hat{\pi}_j, \mathcal{D}_b) - 0.5)\sum_{c \in \{0,1\}} p_{jc}\epsilon_{j,c,\mathcal{D}_b}\right)^2 \right\} \right\} \tag{137}$$

where $p_{j0} = \mathbb{P}_{\mathcal{D}_b}[\mathbb{I}_{[Y \neq j]}]$ and $p_{j1} = \mathbb{P}_{\mathcal{D}_b}[\mathbb{I}_{[Y=j]}]$ are class probabilities on benign distribution.

*Proof of Thm. 4.* The following holds based on the definition of $\hat{C}_{n,\alpha}^{\text{COLEP}}(\cdot)$ in Equation (8).

$$\mathbb{P}\left[Y_{n+1} \notin \hat{C}_{n,\alpha}^{\text{COLEP}}(\tilde{X}_{n+1})\right] = \mathbb{P}\left[S_{\hat{\pi}_{Y_{n+1}}^{\text{COLEP}}}(\tilde{X}_{n+1}, 1) > Q_{1-\alpha}(\{S_{\hat{\pi}_{Y_{n+1}}^{\text{COLEP}}}(X_i, \mathbb{I}_{[Y_i = Y_{n+1}]})\}_{i \in \mathcal{I}_{\text{cal}}})\right] \tag{138}$$

Similarly, the following holds for $\hat{C}_{n,\alpha}(\cdot)$.

$$\mathbb{P}\left[Y_{n+1} \notin \hat{C}_{n,\alpha}(\tilde{X}_{n+1})\right] = \mathbb{P}\left[S_{\hat{\pi}_{Y_{n+1}}}(\tilde{X}_{n+1}, 1) > Q_{1-\alpha}(\{S_{\hat{\pi}_{Y_{n+1}}}(X_i, \mathbb{I}_{[Y_i = Y_{n+1}]})\}_{i \in \mathcal{I}_{\text{cal}}})\right] \tag{139}$$

We can transform the objective of comparison of marginal prediction coverage into a comparison of non-conformity scores as follows:

$$\mathbb{P}\left[\mathbb{P}[Y_{n+1} \in \hat{C}_{n,\alpha}^{\text{COLEP}}(\tilde{X}_{n+1})] > \mathbb{P}[Y_{n+1} \in \hat{C}_{n,\alpha}(\tilde{X}_{n+1})]\right] \tag{140}$$

$$= \mathbb{P}\left[\mathbb{P}\left[Y_{n+1} \notin \hat{C}_{n,\alpha}^{\text{COLEP}}(\tilde{X}_{n+1})\right] \leq \mathbb{P}\left[Y_{n+1} \notin \hat{C}_{n,\alpha}(\tilde{X}_{n+1})\right]\right] \tag{141}$$

$$\geq \mathbb{P}\left[S_{\hat{\pi}_{Y_{n+1}}^{\text{COLEP}}}(\tilde{X}_{n+1}, 1) \leq S_{\hat{\pi}_{Y_{n+1}}}(\tilde{X}_{n+1}, 1)\right] \tag{142}$$

$$\times \mathbb{P}\left[Q_{1-\alpha}(\{S_{\hat{\pi}_{Y_{n+1}}^{\text{COLEP}}}(X_i, \mathbb{I}_{[Y_i = Y_{n+1}]})\}_{i \in \mathcal{I}_{\text{cal}}}) \geq Q_{1-\alpha}(\{S_{\hat{\pi}_{Y_{n+1}}}(X_i, \mathbb{I}_{[Y_i = Y_{n+1}]})\}_{i \in \mathcal{I}_{\text{cal}}})\right] \tag{143}$$

Next, we individually bound the probability formulation in Equations (142) and (143).

**Case (1)**: to bound $\mathbb{P}\left[S_{\hat{\pi}_{Y_{n+1}}^{\text{COLEP}}}(\tilde{X}_{n+1}, 1) \leq S_{\hat{\pi}_{Y_{n+1}}}(\tilde{X}_{n+1}, 1)\right]$.

We first compute $\mathbb{E}\left[S_{\hat{\pi}_{Y_{n+1}}^{\text{COLEP}}}(\tilde{X}_{n+1}, 1) - S_{\hat{\pi}_{Y_{n+1}}}(\tilde{X}_{n+1}, 1)\right]$ and then leverage tail bounds to derive the lower bound of the probability formulation.

*Case (1a)*: $\hat{\pi}_{Y_{n+1}}(\cdot)$ correctly classifies the sample $\tilde{X}_{n+1} \sim \mathcal{D}_b$ (with probability $A(\hat{\pi}_{Y_{n+1}}(\cdot), \mathcal{D}_a)$). Since we assume $\epsilon_{j,c,\mathcal{D}} > 0$ for $j \in [N_c]$, $c \in \{0,1\}$, and $\mathcal{D} \in \{\mathcal{D}_a, \mathcal{D}_b\}$, $\hat{\pi}_{Y_{n+1}}^{\text{COLEP}}(\tilde{X}_{n+1})$ also correctly classifies the sample $\tilde{X}_{n+1} \sim \mathcal{D}_b$, and thus, we have the following:

$$\mathbb{E}\left[S_{\hat{\pi}_{Y_{n+1}}^{\text{COLEP}}}(\tilde{X}_{n+1}, 1) - S_{\hat{\pi}_{Y_{n+1}}}(\tilde{X}_{n+1}, 1)\right] = \frac{1}{2}\mathbb{E}\left[\hat{\pi}_{Y_{n+1}}^{\text{COLEP}}(\tilde{X}_{n+1}) - \hat{\pi}_{Y_{n+1}}(\tilde{X}_{n+1})\right] \tag{144}$$

*Case (1b)*: $\hat{\pi}_{Y_{n+1}}(\cdot)(\cdot)$ wrongly classifies the sample $\tilde{X}_{n+1}$ (with probability $1 - A(\hat{\pi}_{Y_{n+1}}(\cdot), \mathcal{D}_a)$). We have the following:

$$\mathbb{E}\Big[S_{\hat{\pi}_{Y_{n+1}}^{\text{COLEP}}}(\tilde{X}_{n+1}, 1) - S_{\hat{\pi}_{Y_{n+1}}}(\tilde{X}_{n+1}, 1)\Big] \leq \mathbb{E}\Big[1 - \frac{1}{2}\hat{\pi}_{Y_{n+1}}^{\text{COLEP}}(\tilde{X}_{n+1}) - (1 - \frac{1}{2}\hat{\pi}_{Y_{n+1}}(\tilde{X}_{n+1}))\Big] \tag{145}$$

Combining Equation (144) in case (1a) and Equation (145) in case (1b), we have:

$$\mathbb{E}\left[S_{\hat{\pi}_{Y_{n+1}}^{\text{COLEP}}}(\tilde{X}_{n+1}, 1) - S_{\hat{\pi}_{Y_{n+1}}}(\tilde{X}_{n+1}, 1)\right] \tag{146}$$

$$\leq (A(\hat{\pi}_{Y_{n+1}}(\cdot), \mathcal{D}_a) - 0.5)\mathbb{E}\Big[\hat{\pi}_{Y_{n+1}}^{\text{COLEP}}(\tilde{X}_{n+1}) - \hat{\pi}_{Y_{n+1}}(\tilde{X}_{n+1})\Big] \leq (A(\hat{\pi}_{Y_{n+1}}(\cdot), \mathcal{D}_a) - 0.5)\epsilon_{Y_{n+1}, 1, \mathcal{D}_a} \tag{147}$$

Then from Equation (147), the following holds by applying Hoeffding's inequality.

$$\mathbb{P}\left[S_{\hat{\pi}_{Y_{n+1}}^{\text{COLEP}}}(\tilde{X}_{n+1}, 1) - S_{\hat{\pi}_{Y_{n+1}}}(\tilde{X}_{n+1}, 1) > 0\right] \leq \exp\left\{-2n_{\mathcal{D}_a}(A(\hat{\pi}_{Y_{n+1}}(\cdot), \mathcal{D}_a) - 0.5)^2\epsilon_{Y_{n+1}, 1, \mathcal{D}_a}^2\right\} \tag{148}$$

Therefore, we have:

$$\mathbb{P}\left[S_{\hat{\pi}_{Y_{n+1}}^{\text{COLEP}}}(\tilde{X}_{n+1}, 1) \leq S_{\hat{\pi}_{Y_{n+1}}}(\tilde{X}_{n+1}, 1)\right] \geq 1 - \exp\left\{-2n_{\mathcal{D}_a}(A(\hat{\pi}_{Y_{n+1}}(\cdot), \mathcal{D}_a) - 0.5)^2\epsilon_{Y_{n+1}, 1, \mathcal{D}_a}^2\right\} \tag{149}$$

**Case (2)**: to bound $\mathbb{P}\left[Q_{1-\alpha}(\{S_{\hat{\pi}_{Y_{n+1}}^{\text{COLEP}}}(X_i, \mathbb{I}_{[Y_i=Y_{n+1}]})\}_{i \in \mathcal{I}_{\text{cal}}}) \geq Q_{1-\alpha}(\{S_{\hat{\pi}_{Y_{n+1}}}(X_i, \mathbb{I}_{[Y_i=Y_{n+1}]})\}_{i \in \mathcal{I}_{\text{cal}}})\right]$.

We first consider an individual sample $(X_i, Y_i)$ and then apply the union bound the quantile.

Similar to case (1), we first derive the bound for the expectation of the difference of non-conformity scores:

$$\mathbb{E}\left[S_{\hat{\pi}_{Y_{(n+1)}}^{\text{COLEP}}}(X_i, \mathbb{I}_{[Y_i=Y_{n+1}]}) - S_{\hat{\pi}_{Y_{n+1}}}(X_i, \mathbb{I}_{[Y_i=Y_{n+1}]}) \, | \, Y_i \neq Y_{n+1}\right] \geq (A(\hat{\pi}_{Y_{n+1}}(\cdot), \mathcal{D}_b) - 0.5)\epsilon_{Y_{n+1}, 0, \mathcal{D}_b} \tag{150}$$

$$\mathbb{E}\left[S_{\hat{\pi}_{Y_{(n+1)}}^{\text{COLEP}}}(X_i, \mathbb{I}_{[Y_i=Y_{n+1}]}) - S_{\hat{\pi}_{Y_{n+1}}}(X_i, \mathbb{I}_{[Y_i=Y_{n+1}]}) \, | \, Y_i = Y_{n+1}\right] \geq (A(\hat{\pi}_{Y_{n+1}}(\cdot), \mathcal{D}_b) - 0.5)\epsilon_{Y_{n+1}, 1, \mathcal{D}_b} \tag{151}$$

Then with Hoeffding's inequality and the union bound, we have:

$$\begin{aligned}
&\mathbb{P}\left[Q_{1-\alpha}(\{S_{\hat{\pi}_{Y_{n+1}}^{\text{COLEP}}}(X_i, \mathbb{I}_{[Y_i=Y_{n+1}]})\}_{i \in \mathcal{I}_{\text{cal}}}) \geq Q_{1-\alpha}(\{S_{\hat{\pi}_{Y_{n+1}}}(X_i, \mathbb{I}_{[Y_i=Y_{n+1}]})\}_{i \in \mathcal{I}_{\text{cal}}})\right] \\
&\geq 1 - n_{\mathcal{D}_b}\exp\left\{-2n_{\mathcal{D}_b}\big((A(\hat{\pi}_{Y_{n+1}}(\cdot), \mathcal{D}_b) - 0.5)\sum_{c \in \{0,1\}} p_{jc}\epsilon_{Y_{n+1}, c, \mathcal{D}_b}\big)^2\right\}
\end{aligned} \tag{152}$$

where $p_{j0} = \mathbb{P}_{\mathcal{D}_b}[\mathbb{I}_{[Y \neq j]}]$ and $p_{j1} = \mathbb{P}_{\mathcal{D}_b}[\mathbb{I}_{[Y = j]}]$ are class probabilities on benign distribution.

Combining Equations (149) and (152) and plugging them into Equations (142) and (143), we finally get:

$$\begin{aligned}
&\mathbb{P}\left[\mathbb{P}[Y_{n+1} \in \hat{C}_{n,\alpha}^{\text{COLEP}}(\tilde{X}_{n+1})] > \mathbb{P}[Y_{n+1} \in \hat{C}_{n,\alpha}(\tilde{X}_{n+1})]\right] \\
&\geq 1 - \exp\left\{-2n_{\mathcal{D}_a}(A(\hat{\pi}_{Y_{n+1}}(\cdot), \mathcal{D}_a) - 0.5)^2\epsilon_{Y_{n+1}, 1, \mathcal{D}_a}^2\right\} \\
&\quad - n_{\mathcal{D}_b}\exp\left\{-2n_{\mathcal{D}_b}\big((A(\hat{\pi}_{Y_{n+1}}(\cdot), \mathcal{D}_b) - 0.5)\sum_{c \in \{0,1\}} p_{jc}\epsilon_{Y_{n+1}, c, \mathcal{D}_b}\big)^2\right\} \\
&\geq 1 - \max_{j \in [N_c]}\left\{\exp\left\{-2n_{\mathcal{D}_a}(A(\hat{\pi}_{Y_{n+1}}(\cdot), \mathcal{D}_a) - 0.5)^2\epsilon_{Y_{n+1}, 1, \mathcal{D}_a}^2\right\}\right. \\
&\quad \left. + n_{\mathcal{D}_b}\exp\left\{-2n_{\mathcal{D}_b}\big((A(\hat{\pi}_{Y_{n+1}}(\cdot), \mathcal{D}_b) - 0.5)\sum_{c \in \{0,1\}} p_{jc}\epsilon_{Y_{n+1}, c, \mathcal{D}_b}\big)^2\right\}\right\}
\end{aligned} \tag{153}$$

$\square$

## H.3 PROOF OF THM. 5

**Theorem 5** (Recall). Suppose that we evaluate the expectation of prediction accuracy of $\hat{\pi}_j^{\text{COLEP}}(\cdot)$ and $\hat{\pi}_j(\cdot)$ on $n$ samples drawn from $\mathcal{D}_m$ and denote the prediction accuracy as $A(\hat{\pi}_j^{\text{COLEP}}(\cdot), \mathcal{D}_m)$ and $A(\hat{\pi}_j(\cdot), \mathcal{D}_m)$. Then we have:

$$A(\hat{\pi}_j^{\text{COLEP}}(\cdot), \mathcal{D}_m) \geq A(\hat{\pi}_j(\cdot), \mathcal{D}_m), \quad \text{w.p. } 1 - \sum_{\mathcal{D} \in \{\mathcal{D}_a, \mathcal{D}_b\}} p_{\mathcal{D}} \sum_{c \in \{0,1\}} \mathbb{P}_{\mathcal{D}}\left[Y = j\right] \exp\left\{-2n(\epsilon_{j,c,\mathcal{D}})^2\right\}. \tag{154}$$

*Proof of Thm. 5.* We can formulate the accuracy of $\hat{\pi}_j(\cdot)$ on $\mathcal{D}_m$ as:

$$A(\hat{\pi}_j(\cdot), \mathcal{D}_m) = \mathbb{P}[Y \neq j]\mathbb{P}[\hat{\pi}_j(X) < 0.5 | Y \neq j] + \mathbb{P}[Y = j]\mathbb{P}[\hat{\pi}_j(X) \geq 0.5 | Y = j] \tag{155}$$

Similarly, $A(\hat{\pi}_j^{\text{COLEP}}(\cdot), \mathcal{D}_m)$ can be formulated as:

$$A(\hat{\pi}_j^{\text{COLEP}}(\cdot), \mathcal{D}_m) = \mathbb{P}[Y \neq j]\mathbb{P}[\hat{\pi}_j^{\text{COLEP}}(X) < 0.5 | Y \neq j] + \mathbb{P}[Y = j]\mathbb{P}[\hat{\pi}_j^{\text{COLEP}}(X) \geq 0.5 | Y = j] \tag{156}$$

Let $\Delta_j(\cdot) := \hat{\pi}_j^{\text{COLEP}}(\cdot) - \hat{\pi}_j(\cdot)$. From the results in Lemma 5.1, we have $\mathbb{E}[\Delta_j(X) | Y \neq j] \leq -\epsilon_{j,0,\mathcal{D}}$ and $\mathbb{E}[\Delta_j(X) | Y = j] \geq \epsilon_{j,1,\mathcal{D}}$. Note that we have $\epsilon_{j,0,\mathcal{D}} > 0$ and $\epsilon_{j,1,\mathcal{D}} > 0$ by assumption.

Then by rearranging the terms and Hoeffding's inequality, we can derive as follows:

$$\mathbb{P}[\hat{\pi}_j^{\text{COLEP}}(X) - \hat{\pi}_j(X) > 0 | Y \neq j] = \mathbb{P}[\Delta_j(X) > 0 | Y \neq j] \tag{157}$$
$$= \mathbb{P}[\Delta_j(X) - \mathbb{E}[\Delta_j(X)] > -\mathbb{E}[\Delta_j(X)] | Y \neq j] \tag{158}$$
$$\leq \mathbb{P}[\Delta_j(X) - \mathbb{E}[\Delta_j(X)] > \epsilon_{j,0,\mathcal{D}} | Y \neq j] \tag{159}$$
$$\leq \exp\left\{-2n\epsilon_{j,0,\mathcal{D}}^2\right\} \tag{160}$$

Similarly, we consider the case conditioned on $Y = j$.

$$\mathbb{P}[\hat{\pi}_j^{\text{COLEP}}(X) - \hat{\pi}_j(X) < 0 | Y = j] = \mathbb{P}[\Delta_j(X) < 0 | Y = j] \tag{161}$$
$$= \mathbb{P}[\Delta_j(X) - \mathbb{E}[\Delta_j(X)] < -\mathbb{E}[\Delta_j(X)] | Y = j] \tag{162}$$
$$\leq \mathbb{P}[\Delta_j(X) - \mathbb{E}[\Delta_j(X)] < -\epsilon_{j,1,\mathcal{D}} | Y = j] \tag{163}$$
$$\leq \exp\left\{-2n\epsilon_{j,1,\mathcal{D}}^2\right\} \tag{164}$$

Note that from Equations (160) and (164), we have the following relation:

$$\mathbb{P}\left[A(\hat{\pi}_j^{\text{COLEP}}(\cdot), \mathcal{D}_m) - A(\hat{\pi}_j(\cdot), \mathcal{D}_m) \leq 0\right] \tag{165}$$
$$\leq \mathbb{P}[Y \neq j]\mathbb{P}[\hat{\pi}_j^{\text{COLEP}}(X) - \hat{\pi}_j(X) > 0 | Y \neq j] + \mathbb{P}[Y = j]\mathbb{P}[\hat{\pi}_j^{\text{COLEP}}(X) - \hat{\pi}_j(X) < 0 | Y = j] \tag{166}$$
$$\leq \mathbb{P}[Y \neq j]\exp\left\{-2n\epsilon_{j,0,\mathcal{D}}^2\right\} + \mathbb{P}[Y = j]\exp\left\{-2n\epsilon_{j,1,\mathcal{D}}^2\right\} \tag{167}$$

Finally, by considering both the adversarial distribution and benign distribution, we can conclude that:

$$\mathbb{P}[A(\hat{\pi}_j^{\text{COLEP}}(\cdot), \mathcal{D}_m) \geq A(\hat{\pi}_j(\cdot), \mathcal{D}_m)] \geq 1 - \sum_{\mathcal{D} \in \{\mathcal{D}_a, \mathcal{D}_b\}} p_{\mathcal{D}} \sum_{c \in \{0,1\}} \mathbb{P}_{\mathcal{D}}\left[Y = j\right] \exp\left\{-2n(\epsilon_{j,c,\mathcal{D}})^2\right\}. \tag{168}$$

$\square$

## I LOWER BOUND OF $A(\hat{\pi}_j^{\text{COLEP}}, \mathcal{D}_m)$

**Theorem 7** (Lower Bound of $A(\hat{\pi}_j^{\text{COLEP}}, \mathcal{D}_m)$). *Let $E_{j,d}^{COLEP} = \mathbb{E}_{\mathcal{D}_m}[\hat{\pi}_j^{COLEP}(X) | \mathbb{I}_{[Y=j]} = d]$ for $d \in \{0,1\}$. Assume that the class probability computed by the $r$-th PC $\hat{\pi}_{j,r}^{COLEP}$ is conditinally*

*independent of $\hat{\pi}_{j,r-1}^{COLEP}$ for $2 \leq r \leq R$ and $E_{j,0}^{COLEP} < 0.5 < E_{j,1}^{COLEP}$. Assume that we aggregate PCs by taking the average (i.e., $\beta_r = 1/R$). Then we have:*

$$A(\hat{\pi}_j^{COLEP}, \mathcal{D}_m) \geq 1 - \mathbb{P}[Y \neq j] \exp\left\{-3(0.5 - E_{j,0}^{COLEP})^2/\pi^2\right\} - \mathbb{P}[Y = j] \exp\left\{-3((E_{j,1}^{COLEP} - 0.5)^2/\pi^2\right\}.$$
(169)

*Proof sketch.* We construct a sequence $\{J_r\}_{r=1}^R$ in a specific way and prove that it is a super-martingale or a sub-martingale with bounded differences conditioned based on the true label. Then we use Azuma's inequality to bound the objective $A(\hat{\pi}_j^{COLEP}, \mathcal{D}_m)$ in different cases and combine them with union bound.

*Remarks.* Lemma 5.1 indicates that better utility of knowledge models and implication rules can indicate a smaller $E_{j,0}^{COLEP}$ and larger $E_{j,1}^{COLEP}$, which can further imply a tighter lower bound of prediction accuracy according to Thm. 7. Furthermore, according to the proof, we know that $E_{j,0}^{COLEP}$ negatively correlates with the number of PCs $R$ and $E_{j,1}^{COLEP}$ has a positive correlation with $R$, indicating that more PCs (i.e., knowledge rules) implies a better prediction accuracy.

*Proof of Thm. 7.* Consider $R$ PCs. We construct a sequence $\{J_r\}_{r=1}^R$ such that $J_r(X) = \frac{1}{r}\sum_{q\in[r]}\hat{\pi}_j^{(q)}(X)$, where $\hat{\pi}_j^{(q)}(\cdot)$ is defined as Equation (24).

We first consider the case conditioned on $Y \neq j$. Without loss of generality, assume that $\hat{\pi}_j^{(1)}(X) \geq \hat{\pi}_j^{(2)}(X) \geq \cdots \geq \hat{\pi}_j^{(R)}(X)$. Then we can prove $\{J_r\}_{r=1}^R$ is a super-martingale since we have the following:

$$J_{r+1} - J_r = \sum_{q=1}^{r+1}\frac{1}{r+1}\hat{\pi}_j^{(q)}(X) - \sum_{q=1}^{r}\frac{1}{r}\hat{\pi}_j^{(q)}(X)$$
(170)

$$= \frac{1}{r+1}\hat{\pi}_j^{(r+1)}(X) - \frac{1}{r(r+1)}\sum_{q=1}^{r}\hat{\pi}_j^{(q)}(X)$$
(171)

$$\leq \frac{1}{r+1}\hat{\pi}_j^{(r+1)}(X) - \frac{1}{r(r+1)}r\hat{\pi}_j^{(r+1)}(X)$$
(172)

$$\leq 0$$
(173)

$\{J_r\}_{r=1}^R$ also has bounded difference since we have the following:

$$J_r - J_{r+1} = \sum_{q=1}^{r}\frac{1}{r}\hat{\pi}_j^{(q)}(X) - \sum_{q=1}^{r+1}\frac{1}{r+1}\hat{\pi}_j^{(q)}(X)$$
(174)

$$= \frac{1}{r(r+1)}\sum_{q=1}^{r}\hat{\pi}_j^{(q)}(X) - \frac{1}{r+1}\hat{\pi}_j^{(r+1)}(X)$$
(175)

$$\leq \frac{1}{r+1} := c_{r+1}$$
(176)

Also note that we have:

$$\mathbb{E}\left[J_{r+1}(X)\,|\,J_1(X),\cdots,J_r(X)\right] = \frac{r}{r+1}J_r(X) + \frac{1}{r+1}\hat{\pi}_j^{(r+1)}(X)$$
(177)

Therefore, $\{J_r\}_{r=1}^R$ is conditionally independent. Then by using Azuma's inequality, we can derive as follows:

$$\mathbb{P}[J_R - \mathbb{E}[J_R] \geq 0.5 - \mathbb{E}[J_R]] \leq \exp\left\{-\frac{(0.5 - \mathbb{E}[J_R])^2}{2\sum_{q=1}^R c_q^2}\right\} \tag{178}$$

$$\leq \exp\left\{-\frac{(0.5 - \mathbb{E}[J_R])^2}{2\sum_{q=1}^R (\frac{1}{q})^2}\right\} \tag{179}$$

$$\overset{(a)}{\leq} \exp\left\{-\frac{(0.5 - \mathbb{E}[J_R])^2}{2\frac{\pi^2}{6}}\right\} \tag{180}$$

$$\leq \exp\left\{-\frac{3(0.5 - \mathbb{E}[J_R])^2}{\pi^2}\right\} \tag{181}$$

Inequality $(a)$ holds because the inequality $\sum_{q=1}^R \frac{1}{q^2} < \frac{\pi^2}{6}$ holds. Finally, the following holds.

$$\mathbb{P}\left[J_R(X) < 0.5 \,|\, Y \neq j\right] \geq 1 - \exp\left\{-\frac{3(0.5 - \mathbb{E}[J_R])^2}{\pi^2}\right\} \tag{182}$$

Then we consider the case conditioned on $Y = j$. Similarly, we construct a sequence $\{J_r'\}_{r=1}^R$ such that $J_r'(X) = \frac{1}{r}\sum_{q\in[r]} \hat{\pi}_j^{(q)}(X)$, where $\hat{\pi}_j^{(q)}(\cdot)$ is defined as Equation (24). Without loss of generality, assume that $\hat{\pi}_j^{(1)}(X) \leq \hat{\pi}_j^{(2)}(X) \leq \cdots \leq \hat{\pi}_j^{(R)}(X)$. Then we can prove $\{J_r'\}_{r=1}^R$ is a sub-martingale since we have the following:

$$J_{r+1}' - J_r' = \sum_{q=1}^{r+1} \frac{1}{r+1}\hat{\pi}_j^{(q)}(X) - \sum_{q=1}^{r} \frac{1}{r}\hat{\pi}_j^{(q)}(X) \tag{183}$$

$$= \frac{1}{r+1}\hat{\pi}_j^{(r+1)}(X) - \frac{1}{r(r+1)}\sum_{q=1}^r \hat{\pi}_j^{(q)}(X) \tag{184}$$

$$\geq \frac{1}{r+1}\hat{\pi}_j^{(r+1)}(X) - \frac{1}{r(r+1)}r\hat{\pi}_j^{(r+1)}(X) \tag{185}$$

$$\geq 0 \tag{186}$$

Again, we can show that $\{J_r'\}_{r=1}^R$ has bounded difference:

$$J_{r+1}' - J_r' = \sum_{q=1}^{r+1} \frac{1}{r+1}\hat{\pi}_j^{(q)}(X) - \sum_{q=1}^{r} \frac{1}{r}\hat{\pi}_j^{(q)}(X) \tag{187}$$

$$= \frac{1}{r+1}\hat{\pi}_j^{(r+1)}(X) - \frac{1}{r(r+1)}\sum_{q=1}^r \hat{\pi}_j^{(q)}(X) \tag{188}$$

$$\leq \frac{1}{r+1} := c_{r+1}' \tag{189}$$

Also note that we have:

$$\mathbb{E}\left[J_{r+1}'(X) \,|\, J_1'(X), \cdots, J_r'(X)\right] = \frac{r}{r+1}J_r'(X) + \frac{1}{r+1}\hat{\pi}_j^{(r+1)}(X) \tag{190}$$

Therefore, $\{J'_r\}_{r=1}^R$ is conditionally independent. Then by using Azuma's inequality, we can derive as follows:

$$\mathbb{P}[J'_R - \mathbb{E}[J'_R] \leq 0.5 - \mathbb{E}[J'_R]] \leq \exp\left\{-\frac{(\mathbb{E}[J'_R] - 0.5)^2}{2\sum_{q=1}^R (c'_q)^2}\right\} \tag{191}$$

$$\leq \exp\left\{-\frac{(\mathbb{E}[J'_R] - 0.5)^2}{2\sum_{q=1}^R (\frac{1}{q}))^2}\right\} \tag{192}$$

$$\leq \exp\left\{-\frac{(\mathbb{E}[J'_R] - 0.5)^2}{2\frac{\pi^2}{6}}\right\} \tag{193}$$

$$\leq \exp\left\{-\frac{3((\mathbb{E}[J'_R] - 0.5)^2}{\pi^2}\right\} \tag{194}$$

Finally, the following holds.

$$\mathbb{P}\left[J_R(X) > 0.5 \,|\, Y = j\right] \geq 1 - \exp\left\{-\frac{3(\mathbb{E}[J_R] - 0.5)^2}{\pi^2}\right\} \tag{195}$$

Note that we can formulate $A(\hat{\pi}_j^{\text{COLEP}}, \mathcal{D}_m)$ as:

$$A(\hat{\pi}_j^{\text{COLEP}}, \mathcal{D}_m) = \mathbb{P}[Y \neq j]\mathbb{P}[\hat{\pi}_j^{\text{COLEP}}(X) < 0.5 | Y \neq j] + \mathbb{P}[Y = j]\mathbb{P}[\hat{\pi}_j^{\text{COLEP}}(X) > 0.5 | Y = j] \tag{196}$$

Combining Equations (182) and (195), we have:

$$A(\hat{\pi}_j^{\text{COLEP}}, \mathcal{D}_m) \geq 1 - \mathbb{P}[Y \neq j]\exp\left\{-3(0.5 - E_{j,0}^{\text{COLEP}})^2/\pi^2\right\} - \mathbb{P}[Y = j]\exp\left\{-3((E_{j,1}^{\text{COLEP}} - 0.5)^2/\pi^2\right\}. \tag{197}$$

$\square$

## J EXPERIMENT

### J.1 IMPLEMENTATION DETAILS

**Dataset & Model.** We evaluate `COLEP` on three datasets, GTSRB, CIFAR-10, and AwA2. GTSRB consists of 12 types of German road signs, such as "Stop" and "Speed Limit 20". CIFAR-10 includes 10 categories of animals and transportation. AwA2 contains 50 animal categories. We use GTSRB-CNN architecture (Eykholt et al., 2018), ResNet-56 (He et al., 2016), and ResNet-50 (He et al., 2016) as the architecture of models on them, respectively. For fair comparisons, we use the same model architecture and training smoothing factors for models in `COLEP` and SOTA baselines CP and RSCP. We perform smoothed training (Cohen et al., 2019) with the same smoothing factor $\sigma$ for models. We use the official validation set of GTSRB including 973 samples, and randomly select 1000 samples from the test set of CIFAR-10 and AwA2 as the calibration sets for conformal prediction with the nominal level $1 - \alpha = 0.9$ across evaluations. The evaluation is performed on the test set of GTSRB and the remaining samples of the test set of CIFAR-10 and GTSRB. GTSRB and CIFAR-10 contain images with the size of $32 \times 32$. AwA2 contains images with the size of $224 \times 224$. GTSRB and CIFAR-10 are under the MIT license. AwA2 is under the Creative Commons Attribution-NonCommercial-ShareAlike (CC BY-NC-SA) license.

**Implementation details.** We run evaluations using three different random seeds and report the averaged results. We omit the standard deviation of the results since we observe they are of low variance due to a large sample size used for calibration and testing. All the evaluation is done on a single $A6000$ GPU. In the evaluation of certified coverage, we compute the smoothed score or prediction probability $100,000$ times for randomized smoothing and fix the ratio of perturbation bound and the smoothing factor during certification $\delta/\sigma_{cer}$ as 0.5 for both RSCP and `COLEP`. In the evaluation of marginal coverage and set size under PGD attack, we use the attack objective of

cross-entropy loss for RSCP. For `COLEP`, the objective is the same cross-entropy loss for the main model and the binary cross-entropy loss for the knowledge models.

**Knowledge Rules & Reasoning**. In summary, we have 3 PCs and 28 knowledge rules in GTSRB, 3 PCs and 30 knowledge rules in CIFAR-10, and 4 PCs and 187 knowledge rules in AwA2. Based on the domain knowledge in each dataset, we encode a set of implication rules into PCs. In each PC, we encode a type of implication knowledge with disjoint attributes. In GTSRB, we have a PC of shape knowledge (e.g.,"octagon", "square"), a PC of boundary color knowledge (e.g., "red boundary", "black boundary"), and a PC of content knowledge (e.g., "digit 50", "Turn Left"). In CIFAR-10, we have a PC of category knowledge (e.g., "animal", "transportation"), a PC of space knowledge (e.g., "in the sky", "in the water", "on the ground"), and a PC of feature knowledge (e.g., "has legs", "has wheels"). In AwA2, we have a PC of superclass knowledge (e.g., "procyonid", "big cat"), a PC of predation knowledge (e.g., "fish", "meat", "plant"), a PC of appearance (e.g., "big ears", "small eyes"), and a PC of fur color knowledge (e.g., "green", "gray", "white").

For example, in the PC of shape knowledge in GTSRB, we can encode the implication rules related to the shape knowledge such as IsStopSign $\implies$ IsOctagon (stop signs are octagon) with weight $w_1$, IsSpeedLimit $\implies$ IsSquare (speed limit signs are square) with weight $w_2$, and IsTurnLeft $\implies$ IsRound (turn left signs are round) with weight $w_3$. To implement the COLEP learning-reasoning framework, we follow the steps to encode the rules into PCs. (1) We learn the main model and knowledge models implemented with DNNs to estimate a preliminary probability for $N_c$ class labels and $L$ concept labels (e.g., $p(\text{IsSpeedLimit} = 1) = \hat{\pi}_{sp}, p(\text{IsSquare} = 1) = \hat{\pi}_{sq}$). (2) We compute the factor value $F(\mu)$ for any possible assignment $\mu \in \{0,1\}^{N_c+L}$ as $F(\mu) = \exp\{\sum_{h=1}^{H} w_h \mathbb{I}_{[\mu \sim K_h]}\}$ (e.g., $F([1,1,1,1,1,1]) = e^{w_1+w_2+w_3}$ since the assignment satisfy all three knowledge rules above). (3) We construct a three-layer PC as shown in Figure 1: (a) the bottom layer consists of leaf nodes representing Bernoulli distributions of all class and concept labels and the constant factor values of assignments (e.g., $B(\hat{\pi}_{sp}), B(1 - \hat{\pi}_{sp}), B(\hat{\pi}_{sq}), B(1 - \hat{\pi}_{sq}), ..., F([1,1,1,1,1,0]), F([1,1,1,1,1,1]), ...$), (b) the second layer consists of product nodes computing the likelihood of each assignment by multiplying the likelihood of the instantiations of Bernoulli variables and the correction factor values (e.g., $\hat{\pi}_{sp} \times \hat{\pi}_{sq} \times ... \times F([1,1,1,1,1,1]))$, and (c) the top layer is a sum node computing the summation of the products. (4) We linearly combine multiple PCs that encode different types of knowledge rules (e.g., 3 PCs in GTSRB: PCs of shape/color/content knowledge). We compute $\beta_k$ based on the utility of the $k$-th PC on the calibration set and manually set the weights of knowledge rules based on their utility.

Codes are available at `https://github.com/kangmintong/COLEP`.

## J.2 ADDITIONAL EVALUATION OF `COLEP`

Table 1: Comparison of the runtime (millisecond) of COLEP per forward time between a standard model and COLEP. The evaluation is done on RTX A6000 GPUs.

|  | GTSRB | CIFAR-10 | AwA2 |
|---|---|---|---|
| standard model | 13.2 | 15.5 | 35.2 |
| COLEP | 13.5 | 15.7 | 35.6 |

**Visualization of the contribution of different knowledge rules**. In GTSRB, we consider the contribution of the shape knowledge rules, color knowledge rules, and content knowledge rules. In CIFAR-10, we consider the contribution of the category knowledge rules, space knowledge rules, and feature knowledge rules. More details of the meanings and examples of knowledge rules are provided in Appendix J.1. We provide the visualization of contributions of different types of knowledge rules in Figure 5. The results show that the rules related to the color knowledge in GTSRB and space knowledge in CIFAR-10 benefit the certified coverage more, but collectively applying all the well-designed knowledge rules leads to even better coverage. The effectiveness of these types of knowledge attributes to a higher utility of the knowledge models (i.e., the accuracy of concept detection). Concretely, the feature of these concepts is relatively easy to learn and can be more accurately detected (e.g., color concepts are more easily learned than shape concepts). The correlation between the knowledge model utility and effectiveness of COLEP is also theoretically analyzed and observed in Section 5.

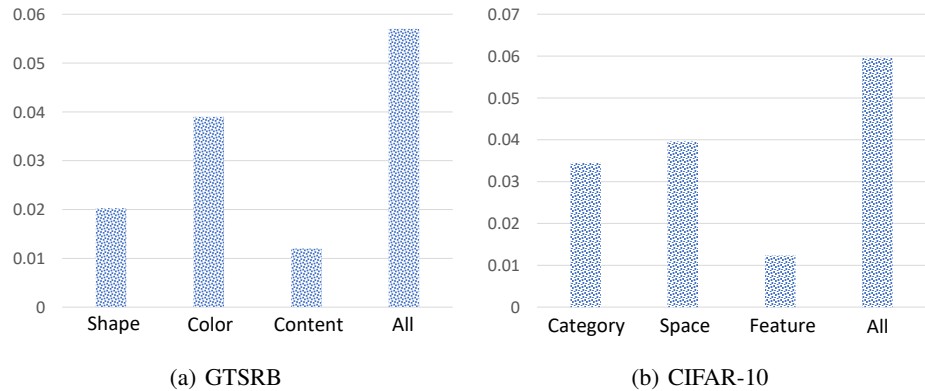

(a) GTSRB           (b) CIFAR-10

Figure 5: Certified coverage of COLEP under bounded perturbation $\delta = 0.25$ on GTSRB and CIFAR-10 with different types of knowledge rules.

**Runtime evaluation of COLEP**. Since the knowledge models in the learning component are simple neural networks and can perform forward passes in parallelism, the additional computational complexity of COLEP comes mainly from the overhead of the reasoning component (i.e., PC). Given a PC defined on the directed graph $(\mathcal{V}, \mathcal{E})$, the time complexity of probabilistic inference is $\mathcal{O}(|\mathcal{V}|)$, according to the details illustrated in Appendix C.1. Therefore, the additional time complexity of COLEP is $\mathcal{O}(|\mathcal{V}|)$, linear to the number of nodes in PC graph. We empirically evaluate the runtime of a forward pass of COLEP and compare it with a standard model. The results in Table 1 show that COLEP only achieves a little higher runtime compared with a single standard model.

Table 2: Marginal coverage / average set size of RSCP and COLEP under AutoAttack with $\ell_2$ bounded perturbation of $0.25$ on GTSRB, CIFAR-10, and AwA2. The nominal level $1 - \alpha$ is 0.9.

|  | GTSRB | CIFAR-10 | AwA2 |
|---|---|---|---|
| RSCP | 0.9682 / 2.36 | 0.9102 / 3.56 | 0.923 / 5.98 |
| COLEP | **0.9702 / 2.13** | **0.9232 / 2.55** | **0.952 / 4.57** |

Table 3: Marginal coverage / average set size of RSCP and COLEP under PGD Attack with multiple desired coverage levels $1 - \alpha$ with $\ell_2$ bounded perturbation of $0.25$ on GTSRB.

|  | $1 - \alpha = 0.85$ | $1 - \alpha = 0.9$ | $1 - \alpha = 0.95$ |
|---|---|---|---|
| RSCP | 0.9029 / 1.63 | 0.9682 / 2.36 | 0.9829 / 2.96 |
| COLEP | **0.9053 / 1.34** | **0.9702 / 2.13** | **0.9859 / 2.53** |

**Evaluations with different types of adversarial attacks**. To evaluate the robustness of COLEP under various types of attacks, we further consider the AutoAttack method Croce & Hein (2020), which uses an ensemble of four diverse attacks to reliably evaluate the robustness: (1) APGD-CE attack, (2) APGD-DLR attack, (3) FAB attack, and (4) square attack. APGD-CE and APGD-DLR attacks are step size-free versions of PGD attacks with cross-entropy loss and DLR loss, respectively. FAB attack optimizes the adversarial sample by minimizing the norm of the adversarial perturbations. Square attack is a query-efficient black-box attack. We evaluate the prediction coverage and averaged set size of COLEP and SOTA method RSCP with $\ell_2$ bounded perturbation of $0.25$ on GTSRB, CIFAR-10, and AwA2. The desired coverage is $1 - \alpha = 0.9$ in the evaluation. The results in Table 2 show that under diverse and stronger adversarial attacks, COLEP still achieves higher marginal coverage than the nominal level $0.9$ and also demonstrates a better prediction efficiency (i.e., smaller size of prediction sets) than RSCP.

**Evaluations with various nominal levels**. We further evaluate COLEP with multiple nominal levels of coverage $1 - \alpha$. We provide the results with multiple coverage levels $1 - \alpha$ in Table 3. The

results demonstrate that for different parameters of $\alpha$, (1) COLEP always achieves a higher marginal coverage than the nominal level, indicating the validity of the certification in COLEP, and (2) COLEP shows a better trade-off between the prediction coverage and prediction efficiency (i.e., average set size) than the SOTA baseline RSCP.

**Selection of weights of knowledge rules.** The weight $w$ of logical rules is specified based on the utility of the corresponding knowledge rule. In Section 6, we theoretically show that the robustness of COLEP correlates with the utility of knowledge rules regarding the uniqueness of the rule, defined in Equation (17). Specifically, a universal knowledge rule which generally holds for almost all class labels is of low quality and does not benefit the robustness and should be assigned with a low weight $w$. For example, if all the traffic signs in the dataset are of octagon shape, then the implication rule $IsStopSign \implies IsOctagon$ should be useless and assigned with a low weight $w$, which also aligns with the intuition. In the empirical evaluations, since we already designed knowledge rules of high quality (i.e., good uniqueness) as shown in Section 7, we do not need to spend great efforts selecting weights for each knowledge rule individually. We assume that all the knowledge rules have the same weight $w$ and select a proper weight $w$ for all the knowledge rules via grid searching. The results inspire us to fix $w$ as $1.5$, and the evaluations demonstrate that it is sufficient to achieve good robustness in different settings. One can also model the weight selection problem as a combinatorial optimization, which is non-trivial but can better unleash the power of COLEP. We leave a more intelligent and automatic approach of knowledge weight selection for future work.

## K  OVERVIEW OF NOTATIONS

In this section, we provide overviews of used notations for a better understanding of the analysis of COLEP. Specifically, we provide an overview of notations related to models in Table 4, notations related to conformal prediction in Table 5, notations related to certification in Section 4 in Table 6, and notations related to analysis in Section 5 in Table 4.

Table 4: Notations related to models in COLEP.

| Notation | Meaning |
|---|---|
| $N_c$ | number of class labels |
| $j$ | $j \in [N_c]$ to index the output of the main model |
| $L$ | number of concept labels, number of knowledge models |
| $l$ | $l \in [L]$ to index $L$ knowledge models/concept labels |
| $\hat{\pi}_j(x)$ | the output likelihood of the main model for the $j$-th class label ($j \in [N_c]$) given input $x$ |
| $\hat{\pi}^{(l)}(x)$ | the output likelihood of the $l$-th knowledge model ($l \in [L]$) given input $l$ |
| $\hat{\pi}$ | $\hat{\pi} := \left[ [\hat{\pi}_j]_{j \in [N_c]}, [\hat{\pi}^{(l)}]_{l \in [L]} \right] \in \mathbb{R}^{N_c + L}$, concatenated vector of class and concept probabilities |
| $j_\forall$ | $j_\forall \in [N_c + L]$ to index $\hat{\pi}$(the concatenation of the output of the main model and concept models) |
| $\hat{\pi}_{j_\forall}(x)$ | the output likelihood of the $j$-th class label ($j_\forall \in [N_c + L]$) **before the reasoning component** |
| $\hat{\pi}_j^{\text{COLEP}}(x)$ | the output likelihood of the $j$-th class label ($j \in [N_c]$) **after the reasoning component** |
| $\{K_h\}_{h=1}^H$ | $H$ propositional logic rules |
| $w_h$ | the weight of the $h$-th logic rule ($h \in [H]$) |
| $\mathbb{I}_{[\cdot]}$ | indicator function |
| $\mu$ | $\mu \in M := \{0,1\}^{N_c + L}$, the index variable in the feasible assignment set $M$ |
| $F(\mu)$ | $F(\mu) = \exp\{\sum_{h=1}^H w_h \mathbb{I}_{[\mu \sim K_h]}\}$, factor value of the assignment $\mu$ |
| $T(a,b)$ | $T(a,b) := \log(ab + (1-a)(1-b))$, introduced for ease of notation |
| $O_{\text{leaf}}, O_{\text{root}}$ | leaf node and root node in the PC |
| $R$ | number of probabilistic circuits |
| $r$ | $r \in [R]$ to index the $r$-th PC |
| $\beta_r$ | the coefficient of the $r$-th PC |

Table 5: Notations related to conformal predictions.

| Notation | Meaning |
|---|---|
| $n$ | number of calibration data samples |
| $(X_i, Y_i)$ | the $i$-th data sample with input $X_i$ and label $Y_i$ ($i \in [n]$) |
| $(X_{n+1}, Y_{n+1})$ | the test sample |
| $\mathcal{Y}$ | label space $[N_c] := \{1, 2, ..., N_c\}$ |
| $y$ | $y \in \mathcal{Y}$ to index labels in the label space $\mathcal{Y}$ |
| $u$ | random variable draw from uniform distribution $U(0,1)$ |
| $Q_{1-\alpha}$ | $1 - \alpha$ quantile value |
| $S_{\hat{\pi}}(x, y)$ | APS non-conformity score of sample $(x, y)$ given probability estimate $\hat{\pi}$ |
| $S_{\hat{\pi}_j^{\text{COLEP}}}(x, y)$ | APS score with $\hat{\pi}_j^{\text{COLEP}}$ (output likelihood of the $j$-th class label after the reasoning component) |
| $\hat{C}_{n,\alpha}(x)$ | $\hat{C}_{n,\alpha}(x) \subseteq \mathcal{Y}$ is the conformal prediction set of input $x$ **without COLEP** |
| $\hat{C}_{n,\alpha_j}^{\text{COLEP}_j}(x)$ | $\hat{C}_{n,\alpha_j}^{\text{COLEP}_j}(x) \subseteq \{0,1\}$ is the conformal prediction set of the $j$-th class label |
| $\hat{C}_{n,\alpha}^{\text{COLEP}}(x)$ | $\hat{C}_{n,\alpha}^{\text{COLEP}}(x) \subseteq \mathcal{Y}$ is the conformal prediction set of input $x$ **with COLEP** |
| $q^y$ | $q^y \in \{0,1\}$ to index in label space $\{0,1\}$ |

Table 6: Notations related to certification in Section 4.

| Notation | Meaning |
|---|---|
| $\overline{\hat{\pi}_{j_\forall}}(x)$ | upper bound of $\hat{\pi}_{j_\forall}(x)$ (likelihood **without reasoning component**) with bounded perturbation |
| $\underline{\hat{\pi}_{j_\forall}}(x)$ | lower bound of $\hat{\pi}_{j_\forall}(x)$ (likelihood **without reasoning component**) with bounded perturbation |
| $\mathbb{U}[\hat{\pi}_j^{\text{COLEP}}(x)]$ | upper bound of $\hat{\pi}_j^{\text{COLEP}}(x)$ (likelihood **with reasoning component**) with bounded perturbation |
| $\mathbb{L}[\hat{\pi}_j^{\text{COLEP}}(x)]$ | lower bound of $\hat{\pi}_j^{\text{COLEP}}(x)$ (likelihood **with reasoning component**) with bounded perturbation |
| $V_d^j$ | index set of conditional variables (left operand of rule $A \implies B$) in the PC except for $j$ |
| $V_s^j$ | index set of conditional variables (right operand of rule $A \implies B$) in the PC except for $j$ |
| $\mu_j$ | the $j$-th element of assignment $\mu$ |
| $\epsilon$ | adversarial perturbation |
| $\delta$ | bound of perturbations such that $\|\epsilon\|_2 \leq \delta$ |
| $\tilde{X}_{n+1}$ | $\tilde{X}_{n+1} = X_{n+1} + \epsilon$ adversarial test sample |
| $\hat{C}_{n,\alpha}^{\text{COLEP}_\delta}(\tilde{X}_{n+1})$ | certified prediction set with perturbation bound $\delta$ |
| $S_{\hat{\pi}_j^{\text{COLEP}_\delta}}$ | worst-case non-conformity score for the $j$-th class label with bounded perturbation $\delta$ |
| $\tau^{\text{COLEP}_{\text{cer}}}$ | lower bound of the coverage guarantee within bounded perturbation |
| $\tau_j^{\text{COLEP}_{\text{cer}}}$ | lower bound of the coverage guarantee of the $j$-th class label within bounded perturbation |

Table 7: Notations related to analysis in Section 5.

| | |
|---|---|
| $\mathcal{D}_b$ | benign data distribution |
| $\mathcal{D}_a$ | adversarial data distribution |
| $\mathcal{D}_m$ | mixed data distribution |
| $p_\mathcal{D}$ | portion of distribution $\mathcal{D}$ in $\mathcal{D}_m$ |
| $\mathcal{G}_r = (\mathcal{V}_r, \mathcal{E}_r)$ | undirected graph encoded with rules in the $r$-th PC |
| $\mathcal{A}_r(j_\forall)$ | node set in connected components of node $j_\forall$ in graph $\mathcal{G}_r$ |
| $\mathcal{A}_{r,d}(j_\forall)$ | node set of conditional variables in connected components of node $j_\forall$ in graph $\mathcal{G}_r$ |
| $\mathcal{A}_{r,s}(j_\forall)$ | node set of consequence variables in connected components of node $j_\forall$ in graph $\mathcal{G}_r$ |
| $\nu(x)$ | boolean vector of ground truth class and concepts given sample $x$ as input |
| $t_{j_\forall, \mathcal{D}}$ | the threshold of confidence for the $j_\forall$-th label |
| $z_{j_\forall, \mathcal{D}}$ | probability matching this threshold for the $j_\forall$-th label |
| $T_{j,\mathcal{D}}^{(r)}$ | multiplication of $t_{j_\forall, \mathcal{D}}$ in the $r$-th PC |
| $Z_{j,\mathcal{D}}^{(r)}$ | multiplication of $z_{j_\forall, \mathcal{D}}$ in the $r$-th PC |
| $\mathcal{P}_d(j), \mathcal{P}_s(j)$ | the set of PC index where $j$ appears as conditional variables and consequence variables |
| $U_j^{(r)}$ | utility of associated knowledge rules within the $r$-th PC for the $j$-th class |
| $\epsilon_{j,0}, \epsilon_{j,1}$ | characterization of probability correction by COLEP conditioned on ground truth label 0 and 1 |
| $\epsilon_{j,0,\mathcal{D}}$ | $\epsilon_{j,0,\mathcal{D}} = \mathbb{E}_{(X,Y)\sim\mathcal{D}}[\epsilon_{j,0}|\nu_j = 0]$, expectation of $\epsilon_{j,0}$ |
| $\epsilon_{j,1,\mathcal{D}}$ | $\epsilon_{j,1,\mathcal{D}} = \mathbb{E}_{(X,Y)\sim\mathcal{D}}[\epsilon_{j,1}|\nu_j = 1]$, expectation of $\epsilon_{j,1}$ |
| $A(\hat{\pi}, \mathcal{D})$ | accuracy of probability estimator $\hat{\pi}$ on data distribution $\mathcal{D}$ |

