# OpenReview forum: "COLEP: Certifiably Robust Learning-Reasoning Conformal Prediction via Probabilistic Circuits"
_ICLR.cc/2024/Conference — ICLR 2024 poster_

### Official Review · Reviewer_nQJc · 2023-10-31

**Soundness:** 3 good
**Presentation:** 2 fair
**Contribution:** 2 fair
**Rating:** 8
**Confidence:** 5

**Summary:**

The authors propose conformal prediction sets that are robust to adversarial perturbations -- they maintain the nominal coverage for any perturbed input in the threat model. The main idea is to leverage a learning-reasoning component which can improve the worst-case bounds on the predicted conditional class probabilities. Constructing the prediction sets using the worst-case scores derived from the worst-case (corrected) class probabilities yields adversarially robust coverage.

**Strengths:**

In terms of originality, the approach can be seen as the clever combination of two main ideas:
- improved worst-case bounds on the class probabilities using a reasoning component
- using worst-case bounds on the conformity scores to get robust sets

Both ideas have been explored in the literature, but their combination leads to improved results. Specifically, more efficient (smaller) sets compared to RSCP while having the same guarantee, and larger worst-case lower bound.

The technical and theorethical components of the paper are sufficiently detailed and rigorous. Theorem 3 can be seen as a generalization of Theorem 2 by Gendler et al. The analysis in Section 6, while making stronger assumptions, is interesting and shows when we can expect improvement.

The exprimental results are expected since the derived bounds are tighter.

The contribution is significant, as is the improvement over the previous SOTA.

**Weaknesses:**

While the authors do consider the finite-sample error induced by the finite size of the calibration set, they fail to account for the finite-sample error due to the Monte-Carlo sampling (they use 10000 samples) when estimating the expectations under the randomized smoothing framework. Therefore, the resulting sets are only asymptotically valid.

As pointed out by a concurent ICLR submission (https://openreview.net/forum?id=BWAhEjXjeG) RSCP also suffers from the same issue and it is non-trivial to correct for this. The same issue is also discussed among the reviewers and the RSCP authors on their respective openreview page (https://openreview.net/forum?id=9L1BsI4wP1H). Naively, one would have to apply a union bound over all examples in the calibration set and the test example such that each of the randomized smoothing expectations hold simultaneously. There is also a subtler alternative solution. When correcting RSCP for this error the resulting sets end up returning all labels, i.e. they are useless. It is not clear to which degree COLEP suffers from the same issue. In any case, the "fix" proposed by the concurent submission is orthogonal and can be applied to COLEP as well. Still, this finite-sample issue needs to be addressed in the paper, especially given that the goal is to produce a sound certificate.

The second weakness is in the PGD attack. The authors state "For fair comparisons, we apply PGD attack (Madry et al., 2018) with the same parameters on CP, RSCP, and COLEP". However, it is unclear whether the PGD attack is adaptive, i.e. it takes the learning-reasoning component into account. An adaptive attacker can in fact know that a reasoning component is used, and thus can try to perturb the input such that both the class and the concept probabilities are suitably changed as to fool the entire pipeline. While this is not critical, not using adaptive attacks makes the conclusions drawn from Figure 3 less reliable.

**Questions:**

1. How does COLEP perform when accounting for finite-sample errors when estimating randomized smoothing expectations?
2. In section 3 you have "assume that the data samples are drawn i.i.d. (thereby exchangeably)". Is the i.i.d. assumptions necessary for the learning-reasonig component, or can this be relaxed to just exchangeability as with vanilla CP?
3. Is the PGD attack adaptive, i.e. takes the reasoning component into account?

---

> ### Author Response · Authors · 2023-11-18
> **Author Response to Reviewer nQJc [1/2]**
>
> > Q1: Consideration of finite-sample error induced by Monte-Carlo sampling in randomized smoothing. How does COLEP perform when accounting for finite-sample errors when estimating randomized smoothing expectations?
>
> We thank you for bringing up this interesting point! We added the following discussions as well as theoretical and empirical results considering the finite-sample errors of randomized smoothing in the revised manuscript.
>
> RSCP+ [1] points out the neglect of finite-sample errors of Monte-Carlo sampling by randomized smoothing in RSCP [2]. A naive solution is to apply union bound over all calibration samples, but it leads to a loose prediction set as shown in RSCP+.  Therefore, RSCP+ proposes a subtle solution by **viewing the empirical mean $\hat{S}\_{\text{RS}}$ as the non-conformity scores** and thus only need to bound the finite-sample error of the empirical mean $\hat{S}\_{\text{RS}}$ on the test sample $(x\_{n+1},y\_{n+1})$ (Corollary 2 in [1]). Specifically, they apply Hoeffding’s inequality and Empirical Bernstein’s inequality to bound the finite-sample error of empirical mean $\hat{S}\_{\text{RS}}$ compared to the true mean $S\_{\text{RS}}$ of the test sample $(x\_{n+1},y\_{n+1})$ (Theorem C.6 in [1]). In this way, RSCP+ does not need to apply union over calibration samples and one concentration bound over the test sample is sufficient.
>
> Since COLEP applies randomized smoothing to the model output in the learning component instead of the end-to-end non-conformity scores, we need to deal with the finite-sample error in a slightly different way, but the general spirit aligns with RSCP+.
> COLEP applies randomized smoothing to get the bound of output probabilities of the main model and knowledge models, and then propagates the bound to the probability bound after the reasoning component (Theorem 1) and the bound of non-conformity scores (Theorem 2).
> Therefore, to consider the finite-sample errors, we need to define a **$\beta$-confidence worst-case non-conformity score** based on Equation (12) in Theorem 2. Concretely, in Equation (12), we replace the original probability upper bound $\mathbb{U}\_{\delta}[\cdot]$ with a $\beta$-confidence version $\mathbb{U}\_{\beta,\delta}[\cdot]$, denoting the upper bound of non-conformity scores with confidence $1-\beta$, and similarly for the lower bound. In this way, during the proof of Theorem 2, we need to apply the union bound in the derivation from Equation (53) to Equation (54) by considering the event probability $\mathbb{P}[\hat{\pi}\_j^{\text{COLEP}}(\tilde{X}\_{n+1}) \le \mathbb{U}\_{\beta,\delta}[\hat{\pi}\_j^{\text{COLEP}}(X\_{n+1})] ] \ge 1 - 2\beta$. Similarly, we consider the event related to the lower bound in the derivation from Equation (57) to Equation (58). Finally, to counter the effect of finite-sample error and maintain a $1-\alpha$ coverage, we need to compute the $1-\alpha+2\beta$ quantile in Equation (11) in Theorem 2. We provided the rigorous theorem statement and proofs in **Theorem 6 in Appendix F.3**. We separated the theoretical result with finite-sample errors since Theorem 2 also allows for deterministic certification methods such as bound propagation methods which do not lead to finite-sample errors. We added related discussions and references in the remarks of Theorem 2.
>
> We also empirically considered the finite-sample errors induced by randomized smoothing expectation approximation and provided the updated results in Figure 2 and Figure 3 in the revised manuscript. We consider the finite-sample errors of RSCP following RSCP+ (Corollary 2 in [1]) and the finite-sample errors of COLEP following Theorem 6 in Appendix F.3. We performed 100k Monte-Carlo sampling for a smaller error in practice and provided all the details in the experiment section of the revised manuscript. We generally observe an error of $[0.002,0.005]$ induced by finite samples of randomized smoothing. One can further reduce the finite-sample errors by increasing the Monte-Carlo sample sizes or applying better concentration bounds. From the updated results in Figure 2 and Figure 3, we can still conclude that COLEP with the reasoning component demonstrates better robustness and prediction efficiency than SOTA baseline RSCP.
>
> *[1] Provably Robust Conformal Prediction with Improved Efficiency, ICLR 2024 submission (https://openreview.net/forum?id=BWAhEjXjeG)*
>
> *[2] Gendler, Asaf, et al. "Adversarially robust conformal prediction." ICLR 2021.*

---

> ### Author Response · Authors · 2023-11-18
> **Author Response to Reviewer nQJc [2/2]**
>
> > Q2: Clarification of adaptive PGD attack against the learning-reasoning pipeline.
>
> We thank you for the valuable comment. We added a clear clarification that we consider an adaptive PGD attack against the complete learning-reasoning pipeline in Section 7. Formally, the attack process can be formulated as the optimization problem: $\max\_{\|\eta\|\_2 \le \delta} \ell\_\text{CE}(\hat{\pi}^{\text{COLEP}}(x),y)$, where $\ell\_\text{CE}$ is the cross-entropy loss and $\hat{\pi}^{\text{COLEP}}(x)$ is the class probabilities given sample $x$ computed by Equation (6) which takes the reasoning component into consideration.
>
> > Q3: Are the i.i.d. assumptions necessary for the learning-reasoning component, or can this be relaxed to just exchangeability as with vanilla CP?
>
> We thank you for the valuable questions. In Section 3 of the revised manuscript, we made it clear that we **only need the exchangeability assumptions** as vanilla CP. Specifically, we only assume the clean test sample is drawn **exchangeably** to the calibration samples and the adversarial sample is within bounded perturbations from the clean test sample.

---

> ### Comment · Reviewer_nQJc · 2023-11-21
> **Reply**
>
> Thank you for the comprehensive reply!
>
> I have one follow question. If I understood correctly Figure 3 in the updated version considers the finite-sample errors for COLEP similar to RSCP+. However, for the second bar do you show RSCP (without error correction) or do you show RSCP+ (with error correction)? I'm asking since [1] reported very high set size for RSCP+.
>
> It might be also interesting to show COLEP and COLEP+ (or however the corrected version is called) side by side.

---

> > ### Author Response · Authors · 2023-11-22
> > **Follow-up Discussions with Reviewer nQJc**
> >
> > We thank you for the reply! In the updated version, we showed the results of RSCP+ (with error correction). [1] reports a very high set size of baseline RSCP+ because they use a small Monte-Carlo sample size $N_{\text{MC}}=256$, according to the paragraph of hyperparameters in Section 5 of [1]. However, we use a large Monte-Carlo sample size of 100K in the updated evaluations. [1] also shows the sensitivity of the selection of Monte-Carlo sample sizes to the certification results in Table 3 of [1]. We selected a large Monte-Carlo sample size to reduce the influence of finite-sample error and focused more on the effectiveness of the reasoning component. Following the suggestion, we provided results of comparisons between COLEP (without finite-sample error as Theorem 2) and COLEP+ (with finite-sample errors as Theorem 6) in Table 5 in Appendix J.3. The results show that with the sample size of 100K, we generally observe a small gap ($0.002\sim0.005$) induced by finite-sample errors.
> >
> > *[1] Provably Robust Conformal Prediction with Improved Efficiency, ICLR 2024 submission (https://openreview.net/forum?id=BWAhEjXjeG)*
> >
> > *[2] Gendler, Asaf, et al. "Adversarially robust conformal prediction." ICLR 2021.*

---

> > > ### Comment · Reviewer_nQJc · 2023-11-22
> > > **Follow up**
> > >
> > > Thank you, that clears things up. I'm assuming the results in Table 5 are for certified coverage. What value of $\delta$?

---

> ### Author Response · Authors · 2023-11-22
> **Follow-up Discussions with Reviewer nQJc**
>
> Thank you for the comment! We evaluate the marginal coverage and average set size under PGD attack with perturbation bound $\delta=0.25$ (same setting as Figure 3). We added the specification in the caption of Table 5 in the updated version.

---

> > ### Comment · Reviewer_nQJc · 2023-11-22
> > **Follow-up**
> >
> > Thank you for the clarification. How does the average set size look like for the *certifed* sets with $\delta=0.25$?

---

> > > ### Author Response · Authors · 2023-11-22
> > > **Follow-up Discussions with Reviewer nQJc**
> > >
> > > Thank you for the question! We included both the marginal coverage and average set size in **Figure 3 and Table 5** under PGD attack ($\delta=0.25$) with the **certified prediction set given by Theorem 2**. In contrast, we evaluate the **certified coverage given by Theorem 3** in **Figure 2**. The certified coverage is the theoretical lower bound of the guaranteed coverage with bounded input perturbations, and thus, the empirical average set size cannot be evaluated in the setting.
> > >
> > > Comparing the average set size of the certified sets between COLEP and COLEP+ in Table 5, we also only observe a marginal gap induced by the finite-sample errors of Monte-Carlo sampling with 100k samples.

---

> > > > ### Comment · Reviewer_nQJc · 2023-11-22
> > > > **Follow up**
> > > >
> > > > Does that mean for the result in table 5 you start with a clean x, then find a perturbed x' with PGD, and then compute the worst case sets for a ball of size $\delta$ around x' either with theorem 2 or theorem 6?

---

> > > > > ### Author Response · Authors · 2023-11-22
> > > > > **Follow-up Discussions with Reviewer nQJc**
> > > > >
> > > > > Thank you for the question! The procedure is generally correct for the inference time. One clarification is that the worst case set has a valid coverage guarantee for the $\delta$-ball around clean $x$ (instead of perturbed $x'$).
> > > > > In practice, we can compute the quantile value used for the construction of the worst case set in the calibration time (before the inference time).
> > > > >
> > > > > The complete procedure including both the calibration and inference is as follows:
> > > > >
> > > > > 1.  **calibration**: given a perturbation bound $\delta$, we compute the worst-case scores for all samples in the calibration set following Equation (12) in Theorem 2 and the $(1-\alpha)$ quantile value $Q_{1-\alpha}$;
> > > > >
> > > > > 2.  **inference**: given a clean $x$, we find a perturbed $x'$ with PGD, and then compute the certified set according to Equation (11) in Theorem 2 by plugging into quantile value $Q_{1-\alpha}$ and the perturbed $x'$.
> > > > >
> > > > > The quantile value $Q_{1-\alpha}$ is a function of perturbation bound $\delta$ and can lead to a certified set with valid coverage guarantee as long as the perturbed test point $x'$ is within $\delta$-ball of $x$. The procedure is similar for Theorem 6 with the consideration of finite-sample error.

---

> > > > > > ### Comment · Reviewer_nQJc · 2023-11-22
> > > > > > **Follow-up**
> > > > > >
> > > > > > Thank you very much for the clarification. It makes sense.
> > > > > >
> > > > > > One final question. If I understand correctly:
> > > > > > - Your certificate plugs in the standard score for for test example and the quantile is computed on the worst-case scores for each calibration point
> > > > > > - While the certificate of [1] and implicitly of RSCP plug in the worst-case score for the test example and the standard score for each callibration point
> > > > > >
> > > > > > This seems to be an interesting difference in where the worst-case bound is being used. Can you in principle also plug in your (tighter) worst-case bound for the test point and use standard scores for the calibration point, resulting in a different certificate?
> > > > > >
> > > > > > [1] Provably Robust Conformal Prediction with Improved Efficiency, ICLR 2024 submission (https://openreview.net/forum?id=BWAhEjXjeG)

---

> > > > > > > ### Author Response · Authors · 2023-11-22
> > > > > > > **Follow-up Discussions with Reviewer nQJc**
> > > > > > >
> > > > > > > Thank you so much for the valuable and detailed comments!
> > > > > > >
> > > > > > > The non-conformity score used in the calibration process in [1] and RSCP is the smoothed score $\tilde{S}(x,y)$ as Equation (9) in [1]. The worst-case score is $\tilde{S}(x,y)+\dfrac{\epsilon}{\delta}$, as Equation (10) in [1]. According to the presentation, [1] and RSCP **(1)** compute smoothed score $\tilde{S}(x,y)$ on all calibration samples and compute the quantile value, and **(2)**  increase the quantile value of smoothed scores by adding $\dfrac{\epsilon}{\delta}$ to get the worst-case score and use it for set construction.
> > > > > > >
> > > > > > > The process is **equivalent to** **(1)** using the worst case score $\tilde{S}(x,y)+\dfrac{\epsilon}{\delta}$ for all calibration samples, and **(2)** compute the quantile value, and use it for set construction. The reason is that the additional term $\dfrac{\epsilon}{\delta}$ is fixed parameters and does not influence the computation of quantile value.
> > > > > > >
> > > > > > > Therefore, [1] and RSCP essentially leverage the worst-case scores ($\tilde{S}(x,y)+\dfrac{\epsilon}{\delta}$) for calibration and use the standard score (smoothed score $\tilde{S}(x,y)$) for inference. We deem that the confusion stems from the presentation of results, but the fundamental proof and evaluations in RSCP and [1] follow the first pattern (worst-case score for calibration, standard score for inference).
> > > > > > >
> > > > > > > Generally, we **should consider the first pattern** (worst-case score for calibration, standard score for inference) in this line of certification. The reason is that the worst-case score cannot be generally computed on the test sample since the computation of the worst-case score requires the clean input $x$ and perturbation bound $\delta$, but the clean input $x$ is not given during test time. Therefore, we can only apply the worst-case score to consider the perturbation in the future during calibration time when the clean samples can be accessed.
> > > > > > >
> > > > > > > The major difference in terms of certification is that [1] and RSCP apply randomized smoothing for end-to-end scores, but COLEP only applies randomized smoothing for models in the learning components and propagates the worst-case bounds in the reasoning component to get the final bounds for the scores.
> > > > > > >
> > > > > > > *[1] Provably Robust Conformal Prediction with Improved Efficiency, ICLR 2024 submission (https://openreview.net/forum?id=BWAhEjXjeG)*

---

> ### Comment · Reviewer_nQJc · 2023-11-22
> **Thank you**
>
> Thank you for the engaging discussion and the detailed reply.
>
> I agree with your comment. One minor point is that since $\frac{\epsilon}{\delta}$ is fixed it can be interepreted either as the first line or the second line of certification.
>
> While the first line might be indeed more reasonable. I think that the second line also make sense. The reason is that if the perturbation ball is symmetric, then a ball around the perturbed $x'$ must contain the clean $x$ (of course the ball around the clean $x$ is different, but the two balls overlap).
>
> Now when certifying classification, if everything inside the ball has the same prediction then the prediction for $x'$ coincides with the prediction for $x$. We can conclude this without knowing the clean $x$. The same applies for the conformal set if we use the worst-case score, since the worst-case score in a ball around x' is by definition larger than the score of the clean x.

---

> ### Author Response · Authors · 2023-11-22
> **Follow-up Discussions with Reviewer nQJc**
>
> Thank you for the quite valuable response!
>
> Providing general certification for the second line based on the symmetry between $x$ and $x'$ is indeed an interesting point! Specifically, given perturbed $x'$ during test time, we can compute the bounds of output logits in the $\delta$-ball, which will also cover the clean $x$. Since the logit bounds also hold for clean $x$, we can compute the worst-case scores based on the bounds and finally construct the prediction set with the clean quantile value.
>
> We agree that the two lines of certification are sound and essentially parallel. It is fundamentally due to the exchangeability of clean samples, which leads to consideration of worst-case bounds either during calibration or inference. One practical advantage of the first line of certification might be that the computation of worst-case bounds requires more runtime, and thus, computing it during the offline calibration process might be more desirable than during the online inference process. In terms of the tightness of the certification, we currently do not have a straightforward intuition of which one is dominantly better. Since we mainly focus on demonstrating the effectiveness of logical reasoning in this paper, we would like to leave the interesting exploration to future work.
>
> We added discussions of alternative certification methods in the revised version in Appendix A. Thank you again for the valuable comments and points!

---

> > ### Comment · Reviewer_nQJc · 2023-11-22
> > **Increased score**
> >
> > Thank you for the interesting discussion. I've increased my score to 8.

---

### Official Review · Reviewer_qYpF · 2023-11-01

**Soundness:** 3 good
**Presentation:** 2 fair
**Contribution:** 2 fair
**Rating:** 6
**Confidence:** 1

**Summary:**

This paper introduces COLEP a certifiably robust learning-reasoning conformal prediction framework. Authors leverage probabilistic circuits for efficient reasoning and provide robustness certification. They also provide marginal coverage guarantees using conformal prediction methods. Finally, by performing several experiments, they highlight the relevance of their proposal.

**Strengths:**

First, I would like to make it clear that this paper is not in my area of expertise. That is why I had some difficulty understanding it. However,

1) the contributions of the paper seem to be significant.

2) experience suggests that COLEP is competitive with previous work.

**Weaknesses:**

The paper is very dense and difficult to read.

Citations are not appropriate. For example, in the related work on conformal prediction, the first sentence mentions articles from 2021, and the seminal articles, notably by Vovk and others, are not cited.

There is no "limitation" in the conclusion. This section should be added.

Minor:

"with the guarantee of marginal prediction coverage:..." the inclusion should be a $\in$.

**Questions:**

Why is COLEP better suited to conformal prediction than already existing methods?

What does "certified conformal prediction" mean? Is this different from saying that we have a marginal coverage guarantee?

The paper claims that it achieved "a certified coverage of COLEP" but, for example in Figure 2, the coverage obtained with COLEP is well below 0.9. Can you explain this in more detail?

---

> ### Author Response · Authors · 2023-11-18
> **Author Response to Reviewer qYpF [1/2]**
>
> > Q1: Density notations and difficulty of reading.
>
> Upon your feedback, we made the following improvements to the readability of the notations and analysis.
>
> 1.  We include more comprehensive and structured notation review tables in Appendix K for a better understanding of the analysis of COLEP. Specifically, we provide an overview of notations related to model definitions in Table 5, notations related to conformal prediction in Table 6, notations related to certification of Section 5 in Table 7, and notations related to analysis of Section 6 in Table 8. All the used notations in COLEP are currently provided in the notation tables in a structured manner, which is emphasized at the beginning of Section 4. In this way, readers can look up the meanings of notations easily and quickly via the tables.
>
> 2. We also improved the overall certification flow in Figure 4 in Appendix E. We restructured the flow and included more illustrations to improve the readability. We also added a brief overview description of the certification flow for a better understanding. The certification setting is that the inference time adversaries can add bounded perturbations to the test sample and violate the data exchangeability assumption, compromising the guaranteed coverage. The certification generally achieves the following three goals. (1) We can preserve the guaranteed coverage using a prediction set that takes the perturbation bound into account (in Theorem 2), achieved by computing the probability bound of models before the reasoning component (by randomized smoothing) and the bound after the reasoning component (by Theorem 1). (2) We prove the worst-case coverage (a lower bound) if we use the standard prediction set as before (in Theorem 3). (3) We theoretically show that COLEP can achieve better prediction coverage (in Theorem 4) and prediction accuracy (in Theorem 5) than a data-driven model without the reasoning component.
>
> 3. We follow the suggestion to separate important definitions from the paragraphs for ease of backtracking the original definitions. Concretely, we separate the definitions of $\hat{\pi}_j(x)$ and $\hat{\pi}^{(l)}(x)$ in Section 4.1 and the definition of $F(\mu)$ in Section 4.2.
>
> We believe that with the improvements, the readers can follow the presented analysis more easily.
>
>
> > Q2: Citation issues.
>
> We thank you for the valuable comments. We added the citations of seminal articles in the related work section.
>
> > Q3: Limitations section.
>
> We included the discussions on limitations in Appendix A and added references to it in Section 8. To summarize, a possible limitation of COLEP may lie in the computational costs induced by pretraining knowledge models. However, the problem can be partially relieved by training knowledge models with parallelism. Moreover, we only need to pretrain them once and then can encode different types of knowledge rules based on domain knowledge. The training cost is valuable since we theoretically and empirically show that more knowledge models benefit the robustness of COLEP a lot for conformal prediction.
> Another possible limitation may lie in the access to knowledge rules.
> For the datasets that release hierarchical information (e.g., ImageNet, CIFAR-100, Visual Genome), we can easily adapt the information to implication rules used in COLEP. We can also seek knowledge graphs with related label semantics for rule design. These approaches may be effective in assisting practitioners in designing knowledge rules, but the process is not fully automatic, and human efforts are still needed. Overall, for any knowledge system, one naturally needs domain experts to design the knowledge rules specific to that application. There is probably no universal strategy on how to aggregate knowledge for any arbitrary application, and rather application-specific constructions are needed. This is where the power as well as limitation of our framework comes from.
>
> > Q4: Notation fix in the preliminaries.
>
> We thank you for pointing it out. We fixed the notation in Section 3 in the revised manuscript.

---

> ### Author Response · Authors · 2023-11-18
> **Author Response to Reviewer qYpF [2/2]**
>
> > Q5: Why is COLEP better suited to conformal prediction than already existing methods?
>
> To the best of our knowledge, RSCP [1] is the most relevant existing method proposed for the problem of adversarially robust conformal prediction.
> RSCP provides the robustness certification of conformal prediction for a single model where they treat the mapping from input to the non-conformity score as a black-box function and apply randomized smoothing to achieve the certification. COLEP differs from RSCP [1] in all these aspects. Namely,
>
> At the model level: RSCP considers a standard single model, while COLEP follows a two-stage learning-reasoning pipeline where the learning component employs several models, and then integrates a separate reasoning component with PCs to it. We study this complex pipeline and theoretically prove that its robustness is better than a robust single model (RSCP) in terms of prediction coverage and prediction accuracy.
>
> At the certification level: the direct application of randomized smoothing as in RSCP does not yield a tight end-to-end robustness certification for COLEP. As such, we
>
>  a) use robustness certification approaches to derive bounds of the learning component,
>
>  b) consider the structural properties of PCs and provide efficient robustness certification of the PC component and
>
>  c) translate the bound of output probabilities of COLEP to the bound of non-conformity scores to achieve the end-to-end certification.
>
> *[1] Gendler, Asaf, et al. "Adversarially robust conformal prediction." International Conference on Learning Representations. 2021.*
>
>
> > Q6: What does "certified conformal prediction" mean? Is this different from saying that we have a marginal coverage guarantee?
>
> Certified conformal prediction presents the worst-case coverage guarantee under bounded input perturbations. Marginal coverage guarantee presents the lower bound of prediction coverage with the assumption of exchangeability, indicating that the test samples are clean and drawn from the same distribution as the calibration samples. In contrast, certified conformal prediction presents the lower bound of prediction coverage **with bounded input perturbations on the clean test sample**. The additional input perturbations can violate the exchangeability assumption and break the original marginal coverage guarantee. Based on the problem, COLEP presents a certification of the worst-case coverage guarantees considering the adversarial perturbations during the inference time and improves the robustness with the power of logical reasoning. We made it more clear in the revised manuscript.
>
> > Q7: Explanations of why the certified coverage of COLEP is below $0.9$ in Figure 2.
>
> We evaluate the certified coverage (i.e., the lower bound of prediction coverage under bounded perturbations) in Figure 2. Since $0.9$ is the nominal coverage level without perturbations during the test time, the certified coverage considering perturbations should be lower than the nominal coverage level $0.9$ and can be rigorously computed by our certification, as shown in Figure 2. In Figure 2, the comparison of the certified coverage to the baseline (RSCP) demonstrates the robustness of COLEP with logical reasoning and the tightness of our certification in Theorem 3.
>
> Furthermore, we can also take the test-time perturbations into consideration during the calibration process to ensure the certified coverage remains at the specified level $0.9$. We evaluate the results in this scenario in Figure 3. The results show that the coverage of COLEP is always above the nominal coverage $0.9$ and achieves better efficiency than RSCP.

---

> > ### Comment · Reviewer_qYpF · 2023-11-22
> >
> > I would like to thank the authors for their detailed response and maintain my score in favor of accepting the article.

---

### Official Review · Reviewer_GJ8Z · 2023-11-01

**Soundness:** 3 good
**Presentation:** 2 fair
**Contribution:** 3 good
**Rating:** 6
**Confidence:** 3

**Summary:**

The paper offers a new construction for a conformal prediction pipeline that offers guaranteed coverage both in the exchangeable and in the adversarial perturbations case, and generally should offer improved adaptivity and set size whenever domain-specific learning-reasoning rules are provided. The main new component is the integration of probabilistic circuits into the construction of a nonconformity score of the APS (Romano et al) type, where an underlying model's class probabilities output is postprocessed according to a collection of reasoning rules, which help correct each class's predicted probability before plugging it into the APS score. The benefits of the reasoning components are investigated theoretically, by means of a theorem stating that under some assumptions marginal coverage will be strictly better with than without these components, as well as experimentally over three datasets and compared to two benchmark conformal method (one with adversarially robust guarantees).

**Strengths:**

This paper essentially constructs a novel type of score for conformal prediction; rather than the simple softmax layer-based constructions of some recent nonconformity scores like APS and RAPS, the idea here is to take such a softmax output from e.g. a neural net and, using domain specific learning-reasoning rules, postprocess each individual class's predicted probability to become appropriately corrected, and only then use them in an APS score-like fashion.

These corrections imply both the robustness properties (at least to l2 perturbations), as well as offer (subject to mild assumptions) theoretically guaranteed improvement over just using the underlying model's softmax outputs.  It is also more generally easy (as the authors show) to propagate estimation uncertainty through the learning-reasoning pipeline. This type of provable guarantees is, to my knowledge, novel and has not come up in the conformal literature before.

Moreover, the proposed technique is more flexible than standard conformal prediction, in that it can provide guarantees not just over all classes, but also for each class separately (at its own coverage threshold).

These above technical innovations, as well as the practical potential of learning-reasoning components to improve domain-specific performance of conformal prediction, lead me to conclude that the proposed paper would be a novel and interesting addition to the conformal literature.

**Weaknesses:**

1. Empirical evaluation is quite limited, which is quite unfortunate given that including more settings beyond datasets as simple as CIFAR-10, as well as more benchmarks, would likely let the guarantees afforded by the learning-reasoning setup shine even more: both in terms of the ability to construct rules specific to each considered setting, as well as the resulting adaptivity compared to other conformal scores/methods.

Relevantly, I was quite confused when the authors repeatedly call the APS score of Romano et al (2020) 'SOTA'. It is certainly one of the existing reasonable conformal approaches to use, but by no means state of the art when it comes to adaptivity and especially set size. For instance, its generalization and improvement, the RAPS nonconformity score introduced in "Uncertainty Sets for Image Classifiers using Conformal Prediction", may still be considered close to the SOTA frontier for many domains --- and it would be very pertinent to include it, along with further improved conformal methods developed thereafter. Also, CIFAR 10 not quite sounding like the natural domain to apply potentially highly complex learning-reasoning boosts, it'd be wise to show the performance of the framework on Imagenet; as a byproduct, it would help me convince myself of the tractability of useful learning-reasoning in more challenging settings. In any case, having just three relatively simple datasets and only two comparison methods, Romano et al and Gendler et al, appears insufficient to fully explore the relative advantages of learning-reasoning.

2. The main other weakness to me is the overbearing nature of the notation used, and the overall presentation. It took me many hours to internalize the type of the guarantees that learning-reasoning components lead to, and even the notation itself. Moreover, especially the multi-subscript-superscript notation involved in both theorem statements and proofs related to the learning-reasoning component was very hard to parse and even read. The notational review in the last appendix was a nice touch, but only marginally helpful as it only listed some of the relevant notation; Diagram in Figure 4, describing the overall certification flow, also felt confusing and didn't really help, making me resort to understanding text only. Instead of structured definitions of the main notations and concepts (using appropriate latex environments), everything was crammed into the paragraphs, thus diluting the structure of the presentation and making it hard to backtrack to the context in which a piece of notation was originally defined.

**Questions:**

1. The experimental section needs strengthening; please see above for suggestions/guidelines on directions.
2. In discussing experimental results, it is of especial interest to discuss which learning-reasoning components were used in each domain. Other details are quite standard and common in the conformal literature, but this one is novel and thus should be propagated to the main part rather than left in the appendix. The current running example is the stop sign example, but it would be very useful to have a walk-through of how the rules were implemented in actual experiments.
3. In the same vein, right now the experimental part only features plots of standard metrics (coverage, set size, ..) Meanwhile, I would like to see a visualization of what the learning-reasoning components do to the predictions of the underlying model --- e.g. which rules turned out to correct the initial estimates by the largest amount, etc.
4. Notation needs to be significantly improved and clarified to achieve readability.

---

> ### Author Response · Authors · 2023-11-18
> **Author Response to Reviewer GJ8Z [1/3]**
>
> > Q1: Limited empirical evaluations.
>
> We thank you for your valuable comments and suggestions on the evaluation part. We made the following updates to clarify and strengthen the evaluation of COLEP.
>
> 1. We avoided the confusion induced by improper namings in Section 7. APS score is indeed not the SOTA method, and thus, we called it a baseline of conformal prediction and only called RSCP [1] as the SOTA approach in adversarially robust conformal prediction in the revised version.
>
> 2. We followed the suggestion to compare with more baselines of various conformal prediction scores to show the robustness of COLEP. Specifically, we further evaluate the RAPS score [2], which improves the APS score by adding a regularization term regarding the ranking of the ground truth label. We also evaluate the recent clustered conformal prediction (CCP) [3], which computes the quantile and performs calibration at the cluster level. We compare COLEP with these baselines on AwA2 under PGD attack. The results in the following Table 1 show that (1) conformal prediction with vanilla conformity scores APS, RAPS, and CCP leads to a broken coverage in the adversary setting, while adversarially robust conformal prediction method RSCP and COLEP still maintains the valid coverage, and (2) compared to RSCP, COLEP demonstrates a better tradeoff between the coverage and prediction efficiency.
>
> 3. We added the following clarifications and more evaluations to demonstrate the scalability of COLEP. Besides the relatively small-scale but standard benchmark CIFAR-10 and GTSRB, we also conducted evaluations on AwA2 [4] in different settings in Figure 2 and Figure 3 in Section 7. AwA2 dataset contains 37,322 images of 50 different animal classes, which are extracted from the ImageNet dataset and with a high resolution of $256 \times 256$. The dataset also has 85 attribute labels (concept labels) for each animal class and is suitable for evaluations of knowledge-intensive tasks. In total, we construct 187 implication rules in total with 4 types of knowledge encoded in 4 PCs separately. Evaluations on AwA2 are already the largest scale in the related literature on learning-reasoning [5] and neuro-symbolic [6]. The scalability of COLEP comes from the usage of probabilistic circuits with linear inference time with respect to the graph size instead of the intractable Markov logic network with NP-complete time complexity used by prior works [5,6]. To consolidate the scalability to AwA2, we further added more evaluations with multiple perturbation bounds in the following Table 1. The results together with results in Figure 2 and Figure 3 on AwA2 demonstrate the scalability and robustness of COLEP to achieve an impressive prediction coverage and efficiency in the adversary setting.
>
> We also admit that it is challenging for evaluations on the full ImageNet with 1000 classes due to the expensive costs of training thousands of concept models and the lack of knowledge rules at the current stage, which is also discussed as one limitation in Appendix A. However, we deem that the principles and analysis in COLEP shed light on the effectiveness of the reasoning component to achieve a more robust conformal prediction. It is also interesting for future work to leverage the CLIP model for concept detection and the large language models for the design of knowledge rules, which can fully unleash the power of COLEP on more large-scale evaluations.
>
> Table 1. Marginal coverage / average set size of multiple conformal prediction baselines, SOTA method RSCP, and COLEP under PGD Attack and AutoAttack with $\ell_2$ bounded perturbations $0.25$ and $0.50$ on AwA2. The nominal level of coverage is $0.9$.
>
> |  | APS | RAPS | CCP | RSCP | COLEP |
> | --- | ----------- | ----------- | ----------- | ----------- | ----------- |
> | $\delta=0.25$ | 0.8325 / 3.22 | 0.8363 / 3.20 | 0.8427 / 3.53 | 0.945 / 6.41 | 0.962 / 4.74 |
> | $\delta=0.50$ | 0.7829 / 2.83 | 0.7821 / 2.89 | 0.7894 / 2.93 | 0.932 / 5.83 | 0.942 / 4.42 |
>
> *[1] Gendler, Asaf, et al. "Adversarially robust conformal prediction." ICLR 2021.*
>
> *[2] Angelopoulos, Anastasios, et al. "Uncertainty sets for image classifiers using conformal prediction." ICLR 2021.*
>
> *[3] Ding, Tiffany, et al. "Class-Conditional Conformal Prediction With Many Classes." NeurIPS 2023.*
>
> *[4] Xian, Yongqin, et al. "Zero-shot learning—a comprehensive evaluation of the good, the bad and the ugly." TPAMI 2018.*
>
> *[5] Yang, Zhuolin. Improving certified robustness via statistical learning with logical reasoning. NeurIPS 2022.*
>
> *[6] Wu, Tailin, et al. "Zeroc: A neuro-symbolic model for zero-shot concept recognition and acquisition at inference time." NeurIPS 2022.*

---

> ### Author Response · Authors · 2023-11-18
> **Author Response to Reviewer GJ8Z [2/3]**
>
> > Q2: The notations are complex for readers.
>
> We thank you for your valuable comments and suggestions. We make the following updates to enhance the reading of the paper.
>
> 1.  We include more comprehensive and structured notation review tables in Appendix K for a better understanding of the analysis of COLEP. Specifically, we provide an overview of notations related to model definitions in Table 5, notations related to conformal prediction in Table 6, notations related to certification of Section 5 in Table 7, and notations related to analysis of Section 6 in Table 8. All the used notations in COLEP are currently provided in the notation tables in a structured manner, which is emphasized at the beginning of Section 4. In this way, readers can look up the meanings of notations easily and quickly via the tables.
>
> 2. We also improved the overall certification flow in Figure 4 in Appendix E. We restructured the flow and included more illustrations to improve the readability. We also added a brief overview description of the certification flow for a better understanding. The certification setting is that the inference time adversaries can add bounded perturbations to the test sample and violate the data exchangeability assumption, compromising the guaranteed coverage. The certification generally achieves the following three goals. (1) We can preserve the guaranteed coverage using a prediction set that takes the perturbation bound into account (in Theorem 2), achieved by computing the probability bound of models before the reasoning component (by randomized smoothing) and the bound after the reasoning component (by Theorem 1). (2) We prove the worst-case coverage (a lower bound) if we use the standard prediction set as before (in Theorem 3). (3) We theoretically show that COLEP can achieve better prediction coverage (in Theorem 4) and prediction accuracy (in Theorem 5) than a data-driven model without the reasoning component.
>
> 3. We follow the suggestion to separate important definitions from the paragraphs for ease of backtracking the original definitions. Concretely, we separate the definitions of $\hat{\pi}_j(x)$ and $\hat{\pi}^{(l)}(x)$ in Section 4.1 and the definition of $F(\mu)$ in Section 4.2.
>
> We believe that these enhancements will make it simpler for readers to understand the provided analysis.

---

> ### Author Response · Authors · 2023-11-18
> **Author Response to Reviewer GJ8Z [3/3]**
>
> > Q3: More details of reasoning component construction in the experiment part in the main text.
>
> We thank you for the valuable suggestion, In summary, we have 3 PCs and 28 knowledge rules in GTSRB, 3 PCs and 30 knowledge rules in CIFAR-10, and 4 PCs and 187 knowledge rules in AwA2. Based on the domain knowledge in each dataset, we encode a set of implication rules into PCs. In each PC, we encode a type of implication knowledge with disjoint attributes. In GTSRB, we have a PC of shape knowledge (e.g., “octagon'', “square''), a PC of boundary color knowledge (e.g., “red boundary'', “black boundary''), and a PC of content knowledge (e.g., “digit 50'', “Turn Left''). In CIFAR-10, we have a PC of category knowledge (e.g., “animal'', “transportation''), a PC of space knowledge (e.g., “in the sky'', “in the water'', “on the ground''), and a PC of feature knowledge (e.g., “has legs'', “has wheels''). In AwA2, we have a PC of superclass knowledge (e.g., “procyonid'', “big cat''), a PC of predation knowledge (e.g., “fish'', “meat'', “plant''), a PC of appearance (e.g., “big ears'', “small eyes''), and a PC of fur color knowledge (e.g., “green'', “gray'', “white'').
>
> For example, in the PC of shape knowledge in GTSRB, we can encode the implication rules related to the shape knowledge such as $\text{IsStopSign} \implies \text{IsOctagon}$ (stop signs are octagon) with weight $w_1$, $\text{IsSpeedLimit} \implies \text{IsSquare}$ (speed limit signs are square) with weight $w_2$, and $\text{IsTurnLeft} \implies \text{IsRound}$ (turn left signs are round) with weight $w_3$. To implement the COLEP learning-reasoning framework, we follow the steps to encode the rules into PCs.
>
> - (1) We learn the main model and knowledge models implemented with DNNs to estimate a preliminary probability for $N_c$ class labels and $L$ concept labels (e.g., $p(\\text{IsSpeedLimit}=1)=\\hat{\\pi}\_{sp}$, $p\(\text{IsSquare}=1)=\\hat{\\pi}\_{sq}$).
>
> - (2) We compute the factor value $F(\mu)$ for any possible assignment $\mu \in \\{0,1\\}^{N_c+L}$ as $F(\mu) = \exp \{\sum\_{h=1}^H w\_h \mathbb{I}\_{[\mu \sim K_h]}\}$ (e.g.,$ F([1,1,1,1,1,1])=e^{w_1+w_2+w_3}$ since the assignment satisfy all three knowledge rules above).
>
> - (3) We construct a three-layer PC as shown in Figure 1: (a) the bottom layer consists of leaf nodes representing Bernoulli distributions of all class and concept labels parameterized by the preliminary class probabilities, and the constant factor values of assignments (e.g., $B(\hat{\pi}\_{sp}),B(1-\hat{\pi}\_{sp}),B(\hat{\pi}\_{sq}),B(1-\hat{\pi}\_{sq}),...,F([1,1,1,1,1,0]),F([1,1,1,1,1,1]),...$), (b) the second layer consists of product nodes computing the likelihood of each assignment by multiplying the likelihood of the instantiations of Bernoulli variables and the correction factor values (e.g., $\hat{\pi}\_{sp} \times \hat{\pi}\_{sq} \times … \times F([1,1,1,1,1,1])$), and (c) the top layer is a sum node computing the summation of the products.
>
> - (4) We linearly combine multiple PCs that encode different types of knowledge rules (e.g., 3 PCs in GTSRB: PCs of shape/color/content knowledge).
>
> We included the statistics of knowledge rules and details of PC implementation in Section 7 and Appendix J.1 in the revised manuscript.
>
> > Q4: Visualization of the functionality of the reasoning component.
>
> We thank you for the suggestion of adding visualization of the contribution of different knowledge rules. Specifically in GTSRB, we consider the contribution of the shape knowledge rules, color knowledge rules, and content knowledge rules. In CIFAR-10, we consider the contribution of the category knowledge rules, space knowledge rules, and feature knowledge rules. More details of the meanings and examples of knowledge rules are provided in response to Q3 and Appendix J.1. We provided the visualization of contributions of different types of knowledge rules in Figure 5 in Appendix J.2. The results show that the rules related to the color knowledge in GTSRB and space knowledge in CIFAR-10 benefits the certified coverage more, but collectively applying all the well-designed knowledge rules lead to even better coverage. The effectiveness of these types of knowledge is attributed to a higher utility of the knowledge models (i.e., the accuracy of concept detection). Concretely, the feature of these more effective concepts is relatively easy to learn and can be more accurately detected (e.g., color concepts are more easily learned than shape concepts). The correlation between the knowledge model utility and effectiveness of COLEP is also theoretically analyzed and observed in Section 6.

---

> > ### Comment · Reviewer_GJ8Z · 2023-11-22
> > **Response to the Authors**
> >
> > Thank you for a very detailed answer to my questions, as well as for making extensive edits to the paper with the aim of making empirical evaluation more extensive and --- importantly --- enhancing readability. Thanks to the above explanations of the mechanics of knowledge rules in the provided examples, I now think I better understand how this new type of conformal prediction can really incorporate side/prior knowledge about a task. Thank you for the clarifications here, as well as for improving the exposition in the Appendices.
> >
> > On the one hand, new experiments (both those addressing my questions on comparisons to true SOTA as well as on more challenging datasets, as well as those in response to other reviewers' requests to test the method's robustness in the presence of additional adversarial attacks) appear to show that COLEP achieves both competitive efficiency and coverage as compared to well-performing prior conformal methods, as well as good robustness in several settings. That is good. Also, I agree with the authors' point that it becomes more challenging to construct knowledge rules etc. for these larger and more unwieldy datasets, and that CLIP and the like may be of good help there --- I am perfectly happy with this being a nice direction for future research, and will not insist on including such further explorations in the paper.
> >
> > On the other hand, readability of the manuscript has also improved due to extensive efforts on the authors' part (adding intuition behind main theorems/results, doing a more extensive recap of all notation, as well as longer/somewhat clearer explanations of the circuitry that is being used). That being said, the text and the formulas are still incredibly densely presented, and the manuscript in my opinion remains very challenging to read for a conformal prediction researcher. I would highly recommend to seriously contemplate ways in which the amount of notation could be significantly reduced (!). As well as, the paper still does not use proper Definition environments etc. to introduce all necessary notions/concepts. Of course, I understand the challenges of needing a lot of notation to rigorously express results that amalgamate several components (i.e. conformal, PCs, etc.) each with their own traditional notation, as well as having limited space to properly lay out all concepts in the intro sections, but I believe that making such further readability improvements will significantly expand the audience for the paper.
> >
> > In sum, having considered both other reviews and the responses, I appreciate the directions in which the paper has been updated, and once again thank the authors for their significant effort to improve the manuscript. I will thus keep my positive score and evaluation of the paper --- but encourage the authors to keep improving exposition (e.g. along the above lines), which still prevents me from raising my score.

---

> ### Author Response · Authors · 2023-11-22
> **Follow-up discussion with Reviewer GJ8Z**
>
> We thank the reviewer for the valuable suggestions and positive feedback!! We will definitely follow the reviewer’s suggestion to further improve our paper further and add additional results and related analysis based on our discussion. Please let us know if you have other suggestions, and we hope our paper will lead to an interesting new direction to integrate data-driven learning with knowledge-enabled logical reasoning for the community.

---

### Official Review · Reviewer_MM5a · 2023-11-03

**Soundness:** 3 good
**Presentation:** 2 fair
**Contribution:** 2 fair
**Rating:** 6
**Confidence:** 3

**Summary:**

The authors applied conformal prediction to a conformity score incorporating logic reasoning based on prior knowledge graphs and demonstrated that the resulting prediction intervals achieved desired coverage while being narrower compared to RSCP on three real data sets under l2 norm perturbations.

**Strengths:**

The authors adapted conformal prediction to a new scenario where a reliable knowledge graph is available and showed empirically that incorporating this side information can improve the efficiency of conformal predictions.

**Weaknesses:**

See questions.

**Questions:**

1. Is the F(mu) prefixed function with H rules? Is mu observed only for the training? (the knowledge graph encodes the relationship between, e.g.,  "stop" and shape info, but the goal is to predict if it is a stop sign).  I am confused about how pi_j(x) is calculated when mu is unknown.

2. It seems that the gain of COLEP originates from utilizing the prior knowledge graph through F(u) and u: The knowledge graphs incorporated are side information with both the graph relationship and u remained true against attacks. How robust does the method perform against contaminated graphs/u?

3. Is it easy to calculate max|η|2≤δ ˆ πj (x + η) and min|η|2≤δ ˆ πj (x + η) for general probability assignment function pi_j(.)? The two main Theorems are from a brute-forth search of the worst-case scenario under perturbations, which are intuitively correct but it seems more important to show the feasibility of achieving this for general functions.

4. How are the weights w chosen in F(mu)?

---

> ### Author Response · Authors · 2023-11-18
> **Author Response to Reviewer MM5a [1/2]**
>
> > Q1: More clarifications about assignment $\mu$, factor function value $F(\mu)$, and the computation of $\pi^{\text{COLEP}}_j(x)$ accordingly.
>
> We included the following clarification of definitions in Section 4.2. $F(\mu)$ is indeed a fixed factor function once $H$ knowledge rules and weights are determined. Note that $\mu$ is an **index variable** of the summation process in Equation (6) (instead of observation), and thus we do not need to observe the values of $\mu$. For a particular index vector $\mu$ in the summation process, $F(\mu)$ computes the summation of the weights of satisfied knowledge rules, denoting how well the index vector $\mu$ aligns with the specified knowledge rules.
>
> Let’s consider the illustrative example of stop sign classification model with an auxiliary knowledge model of octagon shape classification in Section 4.2. The index variable $\mu \in \\{0,1\\}^2$ denotes a possible assignment of Bernoulli random variables $\text{IsStopSign}$ and $\text{IsOctagon}$.
> Therefore, we have that $F(\mu=[0,0])=F(\mu=[0,1])=F(\mu=[1,1])=e^w$ and $F(\mu=[1,0])=e^0$ since only the assignment $\mu=[1,0]$ (i.e., $\text{IsStopSign}=1, \text{IsOctagon}=0$) violates the knowledge rule $\text{IsStopSign} \rightarrow \text{IsOctagon}$ with weight $w$. Then according to Equation (6), we can compute $\pi_j^{\text{COLEP}}(x)$ (i.e., $p(\text{IsStopSign}=1)$) by marginalizing the Bernoulli variable $\text{IsOctagon}$ as follows:
> $$ \pi_j^{\text{COLEP}}(x) = \dfrac{p(\text{IsStopSign}=1)p(\text{IsOctagon}=0)F(\mu=[1,0]) +  p(\text{IsStopSign}=1)p(\text{IsOctagon}=1)F(\mu=[1,1])}{p(\text{IsStopSign}=1)p(\text{IsOctagon}=0)F(\mu=[1,0]) + p(\text{IsStopSign}=1)p(\text{IsOctagon}=1)F(\mu=[1,1]) + p(\text{IsStopSign}=0)p(\text{IsOctagon}=0)F(\mu=[0,0]) + p(\text{IsStopSign}=0)p(\text{IsOctagon}=1)F(\mu=[0,1])} $$, where $p(\cdot)$ is the likelihood estimate of the main model and knowledge models without the reasoning component.
>
>
> > Q2: Discussion of the robustness of COLEP with contaminated knowledge graph.
>
> We thank you for the interesting question! It is interesting to consider cases where the knowledge graph is contaminated by adversarial attackers or contains noises and misinformation by the neglect of designers. According to the response to Q1, $\mu$ is an index variable instead of observations, and thus, the robustness of COLEP mainly originates from the knowledge rules via factor function $F(\cdot)$ denoting how well each assignment conforms to the specified knowledge rules. Since the knowledge rules and weights are specified by practitioners and fixed during inferences, the PCs encoding the knowledge rules together with the main model and knowledge models (learning + reasoning component in Figure 1) can be viewed as a complete model. According to the adversarial attack settings in the literature[1,2], the model weights can be well protected and not manipulated by attackers in practical cases, and thus, the PC graph (i.e., source of side information) which is a part of the complete model can not be easily manipulated by adversarial attackers.
>
> On the other hand, it is an interesting question to discuss the case when the knowledge rules may be contaminated and contain misinformation due to the neglect of designers. We can view the problem of checking knowledge rules and correcting misinformation as a pre-processing step before the employment of COLEP. The parallel module of knowledge checking and correction is explored by a line of research [3,4,5], which basically detects and corrects false logical relations via semantic embeddings and consistency-checking techniques. It is also interesting to leverage more advanced large language models to check and correct human-designed knowledge rules. We deem that the process of knowledge-checking is independent and parallel to COLEP, which may take additional efforts before the deployment of COLEP but significantly benefits in achieving a much more robust system based on our analysis and evaluations. We added the related discussions in Appendix A.
>
>
> *[1] Goodfellow, Ian J., Jonathon Shlens, and Christian Szegedy. "Explaining and harnessing adversarial examples." arXiv preprint arXiv:1412.6572 (2014).*
>
> *[2] Gendler, Asaf, et al. "Adversarially robust conformal prediction." International Conference on Learning Representations. 2021.*
>
> *[3] Melo, André, and Heiko Paulheim. "An approach to correction of erroneous links in knowledge graphs." CEUR Workshop Proceedings. Vol. 2065. RWTH Aachen, 2017.*
>
> *[4] Caminhas, Daniel D. "Detecting and correcting typing errors in open-domain knowledge graphs using semantic representation of entities." (2019).*
>
> *[5] Chen, Jiaoyan, et al. "Correcting knowledge base assertions." Proceedings of the Web Conference 2020.*

---

> > ### Comment · Reviewer_MM5a · 2023-11-22
> >
> > [Q1] Thank you for the explanation-- that makes sense.
> >
> > [Q2] Related, another possibility is that a relationship is partially true, e.g., IsStopSign->IsOctogan is true with a probability of 70%. This is not a good example, but it can happen in many settings. Discussion of such a knowledge graph that might not be universally true could be interesting and related to the choice of w.
> >
> > Minor comments: (1) I suggest changing the example after e.g., on page 3, to match [IsStopSign, IsOctagon] the figure and the probability vector after it.

---

> ### Author Response · Authors · 2023-11-18
> **Author Response to Reviewer MM5a [2/2]**
>
> > Q3: Computation of the bound of class probabilities for more general functions.
>
> As illustrated in Section 5.1, the computation of the bound of class probabilities consists of two steps:
> - (1) bound computation of class probabilities after the learning component (i.e., $\min\_{\| \eta \|_2 \le \delta} \pi\_j(x+\eta)$ and $\max\_{\| \eta \|_2 \le \delta} \pi\_j(x+\eta)$)
> - (2) bound computation of class probabilities after the reasoning component (i.e., $\min\_{\|\eta\|\_2 \le \delta} \pi\_j^{\text{COLEP}}(x+\eta)$ and $\max\_{\|\eta\|\_2 \le \delta} \pi\_j^{\text{COLEP}}(x+\eta)$).
>
> We can compute the bound in step (1) for any general functions $\pi_j$ with randomized smoothing. We can compute the bound in step (2) based on the results of step (1) according to Theorem 1, which holds for general probabilistic circuits that can encode arbitrary logic rules. Therefore, COLEP allows for bound computation for general deep neural networks in the learning component and general probabilistic circuits in the reasoning component.
>
> > Q4: Selection of weight $w$ in $F(\mu)$.
>
> We thank you for the interesting question. The weight $w$ of logical rules is specified based on the utility of the corresponding knowledge rule. In Section 6, we theoretically show that the robustness of COLEP correlates with the utility of knowledge rules regarding the uniqueness of the rule, defined in Equation (17). Specifically, a universal knowledge rule which generally holds for almost all class labels is of low quality and does not benefit the robustness and should be assigned with a low weight $w$. For example, if all the traffic signs in the dataset are of octagon shape, then the implication rule $\text{IsStopSign} \implies \text{IsOctagon}$ should be useless and assigned with a low weight $w$, which also aligns with the intuition.
>
> In the empirical evaluations, since we already designed knowledge rules of high quality (i.e., good uniqueness) as shown in Section 7, we do not need to spend great efforts selecting weights for each knowledge rule individually. We assume that all the knowledge rules have the same weight $w$ and select a proper weight $w$ for all the knowledge rules via grid search. The results in the following Table 1 inspire us to fix $w$ as $1.5$, and the evaluations demonstrate that it is sufficient to achieve good robustness in different settings. One can also model the weight selection problem as a combinatorial optimization, which is non-trivial but can better unleash the power of COLEP. We leave a more intelligent and automatic approach of knowledge weight selection for future work. We included the discussions in Section 7 and Appendix J.
>
> Table 1. Certified coverage of COLEP on GTSRB with different weights $w$ of knowledge rules under perturbation bound $\delta=0.25$.
>
> | w | 0.0 | 0.5 | 1.0 | 1.5 | 2.0 | 3.0 | 5.0 | 10.0 |
> | --- | ----------- | ----------- | ----------- | ----------- | ----------- | ----------- | ----------- | ----------- |
> | Certified Coverage | 0.816 | 0.832 | 0.837 | **0.842**  | 0.840 | 0.823 | 0.829 | 0.827 |

---

> > ### Comment · Reviewer_MM5a · 2023-11-22
> >
> > Q3. I think my question is the computation of the min and max values because it requires looping through eta. Could the authors explain more about what you have done in their experiments to guarantee, e.g., the min_{|\eta|_2 < \delta}\pi_j(x+\eta), has been achieved?
> >
> > Q4. I would suggest the authors have some guidelines for how practitioners choose w themselves without double dipping their test samples. It seems important to automatically address the choice of w as a future work.

---

> ### Author Response · Authors · 2023-11-21
> **Follow-up Discussions**
>
> We thank you again for the valuable suggestions and comments! Since it is close to the end of the discussion period, we wonder whether our responses address your concerns. We are happy to take any further questions or concerns.

---

> ### Author Response · Authors · 2023-11-22
> **Follow-up Discussions with Reviewer MM5a**
>
> We thank you for the valuable feedback!
>
> > Q2 & Q4: More discussions about partially true knowledge rules and the guidelines of selections of rule weights.
>
> We thank you for the interesting and valuable question! If a knowledge rule is true with a lower probability, then the associated weight rule should be smaller, and vice versa. In the example, the probability is with respect to the data distribution, which means that $70\\%$ of data satisfies the rule “IsStopSign->IsOctogan”. Besides, other distribution-dependent factors such as the portions of class labels also affect the importance of corresponding rules. If one class label is pretty rare in the data distribution, then the knowledge rules associated with it clearly should be assigned a small weight, and vice versa. In summary, the importance of knowledge rules depends on the data distribution, no matter whether we select it via grid search or optimize it in a more subtle way. Therefore, we deem that a general guideline for an effective and efficient selection of rule weights is to perform a grid search on a held-out validation set, which is also essential for standard model training. However, we do not think the effectiveness of COLEP is quite sensitive to the selection of weights. From the theoretical analysis and evaluations, we can expect benefits from the knowledge rules as long as we have a positive weight $w$ for the rules.  We included related discussions in Appendix A in the revision.
>
> > Minor comments: Changing the example on page 3.
>
> We changed the example on Page 3 for better alignment in the revised version.
>
> > Q3. More details about how to compute the probability bound (e.g., $\min\_{\\|\\eta\\|\_2 < \delta}\pi_j(x+\eta)$).
>
> We directly apply randomized smoothing [1] to compute the bound of model output in the learning component (i.e., $\min\_{ \\|\\eta\\|_2 < \delta}\\pi\_j(x+\\eta)$, $\max\_{ \\|\\eta\\|_2 < \delta}\\pi\_j(x+\\eta)$) in experiments. Following the standard procedure in [1], we (a) randomly sample Gaussian noises $\eta$ from $\mathcal{N}(0,\sigma^2)$ for 100k times, (b) compute the empirical mean of the output of the smoothed model $\hat{\mathbb{E}}[\\pi\_j(x+\\eta)]$, (c) compute the lower bound and upper bound as $\min\_{ \\|\\eta\\|_2 < \delta}\\pi\_j(x+\\eta) = \Phi(\Phi^{-1}(\hat{\mathbb{E}}[\\pi\_j(x+\\eta)]) - \delta/\sigma)$, $\max\_{ \\|\\eta\\|_2 < \delta}\\pi\_j(x+\\eta)=\Phi(\Phi^{-1}(\hat{\mathbb{E}}[\\pi\_j(x+\\eta)]) + \delta/\sigma)$, where $\Phi(\cdot)$ is the standard Gaussian CDF and $\delta$ is the perturbation bound. We also provided related details in Appendix C.3 in our paper.
>
> *[1] Cohen, Jeremy, Elan Rosenfeld, and Zico Kolter. "Certified adversarial robustness via randomized smoothing." ICML 2019.*

---

> > ### Comment · Reviewer_MM5a · 2023-11-22
> >
> > Thank you for clarifying Q3, I was not aware of this line of work previously. I have raised my score.

---

### Official Review · Reviewer_TP2m · 2023-11-06

**Soundness:** 3 good
**Presentation:** 2 fair
**Contribution:** 3 good
**Rating:** 6
**Confidence:** 3

**Summary:**

COLEP is a learning-reasoning framework for conformal prediction that provides improved certified prediction coverages under adversarial perturbations. The framework is composed of a learning/sensing component followed by a knowledge-enabled logical reasoning component. The learning component consists of several deep learning models while the reasoning component consists of one or more probabilistic circuits. Alongside the main classification task, the deep learning models are used to predict and estimate the probabilities of other concepts in the input e.g. shape, color etc. The PCs in the reasoning component encode domain knowledge specified as propositional rules over the class and concept variables and helps to ensure robustness against $\ell_2$ bounded adversarial perturbations to the input variables. The experiments show that COLEP achieves higher prediction coverages as well as smaller prediction set sizes compared to other SOTA methods.

**Strengths:**

The paper presents a novel idea to improve certified robustness of conformal prediction using deep learning models. To the best of my knowledge, this is the first work on using tractable probabilistic circuits to improve the coverage of conformal prediction.

A positive side of the work is its theoretical analyses. The authors have derived certified coverage for COLEP under both $ell_2$ and finite calibration sample size. And then show that COLEP received higher certified coverage compared to a single model.

Experimental results are also promising.

**Weaknesses:**

The manuscript is strong overall, but there are areas where it could benefit from further elaboration and additional empirical support.

1. The reasoning part seems to be missing significant details to understand how the PCs are working together to reduce the error produced by the main classifier. The paper makes a strong assumption that the reader will be familiar with semantics of PC structures. There should be some brief introduction to these models.

2. The leaf weights (factor weights) are assumed to be prespecified and then there are also the Bernoulli parameters estimated by the neural networks.  It is unclear how these estimated parameters, predictions (classes and concepts) and user defined knowledge rules are combined together to make robust decisions. The coefficients of the component PCs $\beta_k$ should be more clearly specified in the main manuscript.

3. The theorems in the main paper need a more detailed discussion. Having only a short paragraph for each theorem doesn't fully explain the complexities and nuances involved, which doesn't give the paper's theoretical contributions their due.

4. To make the empirical evaluation stronger, the authors could consider using more datasets or testing against different types of attacks. This will help confirm that COLEP is robust and works well in various situations.

**Questions:**

1. How many PCs were considered in each of the experiments? Can the authors give an idea on the complexity of the PCs?

2. Can the scope of the evaluation be extended to include the performance of COLEP against  other forms of adversarial attacks?

3. Why wasn't the effect of varying the parameter $\alpha$ and $\rho$ on COLEP's performance analyzed?

---

> ### Author Response · Authors · 2023-11-18
> **Author Response to Reviewer TP2m [1/3]**
>
> > Q1: Provide more details about how the PCs can help reduce the error of the main classifier. Specifically, add clarifications of how the prespecified factor weights and estimated likelihood (classes and concepts) and user-defined knowledge rules are combined together to make robust decisions.
>
> We thank you for the valuable comment. We included the following details about the PC structure and its benefits to the robustness in Section 4.2 in the revised manuscript.
>
> PC defines a joint distribution over a set of random variables (e.g., $\text{IsStopSign}$, $\text{IsOctagon}$) and computes the joint probability of them in the graph efficiently by performing forwarding passes. We can encode the knowledge rules with a 3-layer PC as shown in Figure 1: (1) the bottom layer consists of leaf nodes representing Bernoulli variables; (2) the second layer consists of product nodes computing the joint probability of an assignment by multiplying the likelihood of individual Bernoulli variables and the factor values (e.g., $\hat{\pi}_{s} \times \hat{\pi}^{o} \times F([1,1])$); and (c) the top layer is a sum node computing the summation of the products. With the PC structure, one can efficiently compute the marginal probabilities in Equation (6) with two forwarding passes. The details and proofs are shown in Appendix C.1.
>
> The reasoning component can correct the output probability with encoded knowledge rules and improve the robustness of the prediction. Consider the following concrete example. Suppose that the adversarial example of a speed limit sign misleads the main classifier to detect it as a stop sign with a large probability (e.g., $\hat{\pi}_s=0.9$), but the octagon classifier is not misled and correctly predicts that the speed limit sign does not have an octagon shape (e.g., $\hat{\pi}_o=0.0$). Then according the computation as Equation (6), the corrected probability of detecting a stop sign given the attacked speed limit image is $\hat{\pi}^{\text{COLEP}}_s=0.9/(0.1e^w+0.9)$, which is down-weighted from the original wrong prediction $\hat{\pi}_s=0.9$ as we have a positive logical rule weight $w>0$. Therefore, the reasoning component can correct the wrong output probabilities with the knowledge rules and lead to a more robust prediction framework. Besides the intuition reflected by the simple example, we also theoretically show that the reasoning component benefits in achieving better prediction coverage and accuracy compared to a single DNN without the reasoning component.
>
>
> > Q2: Add clear specification of PC coefficient $\beta_r$ in the main manuscript.
>
> Thank you for the suggestion! We incorporated more specifications for PC coefficients $\beta_r$ in the main manuscript in Section 4.2. Concretely, we can formulate the corrected conditional class probability $\pi_j^{\text{COLEP}}$ with multiple PCs as $\pi_j^{\text{COLEP}}(x) = \sum_{r \in [R]} \beta_{r} \pi_j^{(r)}(x)$, where $\pi_j^{(r)}$ is the output class probabilities of the $r$-th PC. The core of this formulation is the mixture model involving a latent variable $r_{\text{PC}}$ representing the PCs. In short, we can write $\pi_j^{(r)}(x)$ as $\pi_j^{(r)}(x)=\mathbb{P}[Y=y|X=x, r_{\text{PC}}=r]$, and $\pi_j(x)$ as the marginalized probability over the latent variable $r_{\text{PC}}$ as $\pi_j(x)=\sum_{r \in [R]} \mathbb{P}[r_{\text{PC}}=r] \cdot \pi_j^{(r)}(x)$. Hence, the coefficient $\beta_{r}$ for the $r$-th PC are determined by $\mathbb{P}[r_{\text{PC}}=r]$. Although we lack direct knowledge of this probability, we can estimate it using the data by examining how frequently each PC $r$ correctly predicts the outcome across the given examples, similarly as in the estimation of prior class probabilities for Naive Bayes classifiers.

---

> ### Author Response · Authors · 2023-11-18
> **Author Response to Reviewer TP2m [2/3]**
>
> > Q3: Add more discussions about the theorems in the main paper.
>
> We thank you for the valuable suggestion. We added more details in the remarks of theorems in the revised manuscript. Concretely, we made the following additions:
>
> - updated details of how we can get the probability bounds of the learning component and how we leverage them for computations of the bounds after the reasoning component in remarks of Theorem 1
>
> - details of the meanings of different terms such as certified prediction set and worst-case non-conformity score and the corresponding references to the equations in the remark of Theorem 2
>
> - illustrations of the technical difference between Theorem 3 and Theorem 2 in the remark of Theorem 3 for a better understanding
>
> - discussions of the physical meanings of the quantities affecting the bounds such as the utility of models $T_{j,\mathcal{D}}^{(r)},Z_{j,\mathcal{D}}^{(r)}$ and the utility of knowledge rules $U_j^{(r)}$ in the remark of Theorem 4 and the practical insights implied by the results
>
> - discussions of the influence of the utility of models $T_{j,\mathcal{D}}^{(r)},Z_{j,\mathcal{D}}^{(r)}$ and the utility of knowledge rules $U_j^{(r)}$ on the prediction accuracy of COLEP and the practical implications in the remark of Theorem 5
>
> > Q4: Add more empirical evaluations of COLEP. Specifically, evaluate the performance of COLEP against other forms of adversarial attacks.
>
> We thank you for the valuable suggestions. To evaluate the robustness of COLEP under various types of attacks, we further consider AutoAttack [1], which uses an ensemble of four diverse attacks to reliably evaluate the robustness: (1) APGD-CE attack, (2) APGD-DLR attack, (3) FAB attack, and (4) square attack. APGD-CE and APGD-DLR attacks are step-size-free versions of PGD attacks with cross-entropy loss and DLR loss, respectively. FAB attack optimizes the adversarial sample by minimizing the norm of the adversarial perturbations. Square attack is a query-efficient black-box attack. We evaluate the prediction coverage and averaged set size of COLEP and SOTA method RSCP with $\ell_2$ bounded perturbation of $0.25$ on GTSRB, CIFAR-10, and AwA2. The desired coverage is $1-\alpha=0.9$ in the evaluation. The results in the following Table 1 show that under diverse and stronger adversarial attacks, COLEP still achieves higher marginal coverage than the nominal level $0.9$ and also demonstrates a better prediction efficiency (i.e., smaller size of prediction sets) than RSCP. We incorporated the results in Appendix J.2 and referred to it in Section 7 in the main text.
>
> Table 1. Marginal coverage / average set size of RSCP and COLEP under AutoAttack [1] with $\ell_2$ bounded perturbation of $0.25$ on GTSRB, CIFAR-10, and AwA2.
>
> | | GTSRB | CIFAR-10 | AwA2 |
> | --- | ----------- | ----------- | -- |
> | RSCP | 0.9682 / 2.36 | 0.9102 / 3.56 | 0.9230 / 5.98  |
> | COLEP | **0.9702** / **2.13** | **0.9232** / **2.55** | **0.9520** / **4.57** |
>
> *[1] Croce, Francesco, and Matthias Hein. "Reliable evaluation of adversarial robustness with an ensemble of diverse parameter-free attacks." International conference on machine learning. PMLR, 2020.*

---

> ### Author Response · Authors · 2023-11-18
> **Author Response to Reviewer TP2m [3/3]**
>
> > Q5: Include more details about the number and complexity of PCs in the experiment part.
>
> We thank you for the valuable question. In summary, we have 3 PCs and 28 knowledge rules in GTSRB, 3 PCs and 30 knowledge rules in CIFAR-10, and 4 PCs and 187 knowledge rules in AwA2. Based on the domain knowledge in each dataset, we encode a set of implication rules into PCs. In each PC, we encode a type of implication knowledge with disjoint attributes. In GTSRB, we have a PC of shape knowledge (e.g., “octagon'', “square''), a PC of boundary color knowledge (e.g., “red boundary'', “black boundary''), and a PC of content knowledge (e.g., “digit 50'', “Turn Left''). In CIFAR-10, we have a PC of category knowledge (e.g., “animal'', “transportation''), a PC of space knowledge (e.g., “in the sky'', “in the water'', “on the ground''), and a PC of feature knowledge (e.g., “has legs'', “has wheels''). In AwA2, we have a PC of superclass knowledge (e.g., “procyonid'', “big cat''), a PC of predation knowledge (e.g., “fish'', “meat'', “plant''), a PC of appearance (e.g., “big ears'', “small eyes''), and a PC of fur color knowledge (e.g., “green'', “gray'', “white''). For example, in the PC of shape knowledge in GTSRB, we can encode the implication rules related to the shape knowledge such as $\text{IsStopSign} \implies \text{IsOctagon}$ (stop signs are octagon), $\text{IsSpeedLimit} \implies \text{IsSquare}$ (speed limit signs are square), and  $\text{IsTurnLeft} \implies \text{IsRound}$ (turn left signs are round). We included the statistics of PCs and knowledge rules in Section 7 and more details in Appendix J.1 in the revised manuscript.
>
> > Q6: Add evaluations with variations of parameters $\alpha$ and $\delta$.
>
> We thank you for the valuable suggestions. We further evaluate COLEP with multiple nominal levels of coverage $1-\alpha$ and with various perturbation bounds $\delta$. We provide the results with multiple coverage levels $1-\alpha$ in the following Table 2, and the results with multiple perturbation bounds $\delta$ in Table 3. The results demonstrate that for different parameters of $\alpha$ and $\delta$, (1) COLEP always achieves a higher marginal coverage than the nominal level, indicating the validity of the certification in COLEP, and (2) COLEP shows a better trade-off between the prediction coverage and prediction efficiency (i.e., average set size) than the SOTA baseline RSCP. We incorporated the results in Appendix J.2 and referred to it in Section 7 in the main text.
>
> Table 2. Marginal coverage / average set size of RSCP and COLEP under PGD Attack with multiple desired coverage levels $1-\alpha$ with $\ell_2$ bounded perturbation of $0.25$ on GTSRB.
>
> |  | $1-\alpha=0.85$ | $1-\alpha=0.9$ | $1-\alpha=0.95$ |
> | --- | ----------- | ----------- | -- |
> | RSCP | 0.9029 / 1.63 | 0.9682 / 2.36 | 0.9829 / 2.96  |
> | COLEP | **0.9053** / **1.34** | **0.9702** / **2.13** | **0.9859** / **2.53** |
>
> Table 3. Marginal coverage / average set size of RSCP and COLEP under PGD Attack with multiple $\ell_2$ bounded perturbations on GTSRB. The nominal level of coverage is $0.9$.
>
> | | $\delta=0.125$ | $\delta=0.25$ | $\delta=0.5$ |
> | --- | ----------- | ----------- | -- |
> | RSCP | 0.9320 / *1.34* | 0.9682 / 2.36 | 0.9712 / 2.78  |
> | COLEP | **0.9891** / 1.47 | **0.9702** / **2.13** | **0.9780** / **2.69** |

---

> ### Author Response · Authors · 2023-11-21
> **Follow-up Discussions**
>
> We thank you again for the valuable suggestions and comments! Since it is close to the end of the discussion period, we wonder whether our responses address your concerns. We are happy to take any further questions or concerns.

---

> > ### Comment · Reviewer_TP2m · 2023-11-22
> >
> > I thank the authors for their detailed responses and I have no further questions. At this point, I would like to increase the score for the paper.

---

### Author Response · Authors · 2023-11-18
**Revision Summary**

We thank all the reviewers for their valuable comments and feedback. We are glad that the reviewers find our work novel with sound theoretical and empirical results. Based on the reviews, we have made the following updates to further improve our work. **All the updates were already incorporated into the revised manuscript with a highlighted blue color.**

1. We provided more details of PC structure and reasoning with illustrative examples in Section 4.2 and more details of PC construction in experiments in Section 7, following the suggestions of Reviewer TP2m and GJ8Z.

2. We added more illustrations of PC coefficients in the main text in Section 4.2 and more detailed remarks for theorems in Sections 5 and 6, following the suggestions of Reviewer TP2m.

3. We strengthened the evaluations of COLEP by adding the following experiments in Section 7 and Appendix J: (1) evaluations under diverse attacks, (2) evaluations with different coverage level $1-\alpha$ and perturbation bound $\delta$, (3) comparisons with more conformal prediction baselines, and (4) visualizations of contributions of different knowledge rules, following the suggestions of Reviewer TP2m and GJ8Z.

4. We added more clarifications on the functionality of the reasoning component in Section 4.2 and the generalizability of COLEP certification in Section 5.2, following the suggestions of Reviewer MM5a.

5. We added more discussions of COLEP with contaminated knowledge graph in Appendix A and the selection of knowledge weights in Section 7 and Appendix J.2, following the suggestions of Reviewer MM5a.

6. To make the presentation of COLEP more clear, we (1) included more comprehensive and structured notation review tables in Appendix K, (2) restructured and illustrated the overall certification flow in Figure 4 in Appendix E, and (3) separated important definitions from the paragraphs in Section 4, following the suggestions of Reviewer GJ8Z and qYpF.

7. We added references to the limitation part in Section 8 and fixed the notation and citations, following the suggestions of Reviewer qYpF.

8. We added a more comprehensive certification version considering the finite-sample errors induced by randomized smoothing mean approximation in Theorem 6 in Appendix F.3 and updated the corresponding evaluation results, following the suggestions of Reviewer nQJc.

9. We added clarifications of the assumption in Section 3 and experiment settings in Section 7, following the suggestions of Reviewer nQJc.

---

### Meta-Review · Area_Chair_Hi79 · 2023-12-10

**Metareview:**

The paper proposes to use probabilistic circuits to enable the tractable computation of conformal predictions that are robust to adversarial attacks. This also enables to build a neuro-symbolic pipeline where symbolic knowledge is embedded in the neural pipeline via circuits. The authors provide theoretical probabilistic guarantees where the robustness is certified. The proposed method, COLEP, is then tested on datasets for on certified conformal prediction under adversarial attacks.

Reviewers appreciated the theoretical contribution and direction of COLEP. Several concerns were raised w.r.t. the experimental setting in which COLEP is evaluated and w.r.t. the dense presentation of the paper (missing details and construction on which circuits are used and how they are built). The authors addressed some of these points in the rebuttal by running more experiments as suggested by reviewers and clarifying missing details. This significantly improved the quality of the paper.

The paper is accepted and authors are asked to discuss in the camera ready the connections with the applications of circuits for neuro-symbolic learning and how COLEP poses itself in that landscape.

**Justification For Why Not Higher Score:**

The current presentation of the manuscript is hard to parse and it is even harder to pose it in the rich literature of probabilistic circuits for neuro-symbolic AI.

**Justification For Why Not Lower Score:**

This constitutes a nice contribution not only for the circuit community, the proposed bounds can be of interest for research on approximate inference with circuits.

---

### Decision · Program_Chairs · 2024-01-16

Accept (poster)